# Overcoming data scarcity in biomedical imaging with a foundational multi-task model

Raphael Schäfer [1], Till Nicke[1], Henning Höfener [1], Annkristin Lange[1], Dorit Merhof[1,2], Friedrich Feuerhake[3,4], Volkmar Schulz[1,5], Johannes Lotz [1,6] & Fabian Kiessling [1,5,6]

Foundational models, pretrained on a large scale, have demonstrated substantial success across non-medical domains. However, training these models typically requires large, comprehensive datasets, which contrasts with the smaller and more specialized datasets common in biomedical imaging. Here we propose a multi-task learning strategy that decouples the number of training tasks from memory requirements. We trained a universal biomedical pretrained model (UMedPT) on a multi-task database including tomographic, microscopic and X-ray images, with various labeling strategies such as classification, segmentation and object detection. The UMedPT foundational model outperformed ImageNet pretraining and previous state-of-the-art models. For classification tasks related to the pretraining database, it maintained its performance with only 1% of the original training data and without fine-tuning. For out-of-domain tasks it required only 50% of the original training data. In an external independent validation, imaging features extracted using UMedPT proved to set a new standard for cross-center transferability.

Deep learning has started to revolutionize biomedical image analysis due to its ability to learn and extract useful image representations. A widely adopted approach for enabling deep learning in biomedical image analysis involves pretraining models on extensive natural image datasets, such as ImageNet-1K[1] or LAION[2], and subsequently either fine-tuning them or utilizing pretrained features directly for specific target tasks[3–5]. Fine-tuning leverages the pretrained model's weights as the initial foundation, enabling accelerated training and enhanced performance even in situations with limited data. Alternately, such foundational models can be kept frozen, with their features directly applied to biomedical downstream tasks. Despite requiring more computation time and data, fine-tuning has firmly established itself

as standard practice across a diverse range of downstream computer vision tasks, encompassing object detection and semantic segmentation, among others[6,7].

Driven by the recent trend of increasingly large pretraining datasets, the need for foundational models in biomedical imaging has become clear[8,9]. However, effective pretraining of deep neural networks requires large amounts of annotated training data, which are often scarce in biomedical imaging[10]. Although many public small- and medium-sized datasets exist in the biomedical domain, there is no single pretraining dataset comparable to ImageNet or LAION.

Several methods have been proposed to address the data scarcity problem. One approach is to use self-supervised learning, which learns

[1]Fraunhofer Institute for Digital Medicine MEVIS, Bremen, Germany. [2]Institute of Image Analysis and Computer Vision, Faculty of Informatics and Data Science, University of Regensburg, Regensburg, Germany. [3]Institute for Pathology, Hannover Medical School, Hanover, Germany. [4]Institute for Neuropathology, Medical Center, University of Freiburg, Freiburg, Germany. [5]Institute for Experimental Molecular Imaging, RWTH Aachen University, Aachen, Germany. [6]These authors contributed equally: Johannes Lotz, Fabian Kiessling. ✉e-mail: johannes.lotz@mevis.fraunhofer.de; fkiessling@ukaachen.de

visual representations from unlabeled data by solving pretext tasks. However, clear performance gaps exist between self-supervised and label-supervised pretraining methods[11].

Another approach is to use domain-specific supervised pretraining. For example, Zhou and colleagues[11] used a large text-labeled chest X-ray dataset to train universal representations for chest X-rays. They evaluated their approach on unseen datasets and found that their chest X-ray encoder outperforms ImageNet pretraining by up to 10% accuracy when applied to other chest X-ray analysis tasks. Nonetheless, supervised pretraining can only be applied to domains where large amounts of training data are available, such as radiographs.

Mei and colleagues[5] proposed to combine multiple medical classification datasets into one and use it for pretraining deep networks for radiology tasks, often outperforming ImageNet. However, the approach relies solely on classification labels, which may not capture all relevant information in medical images, and it requires the network to predict unrelated or not meaningful classes in the combined dataset.

Multi-task learning (MTL) promises to provide a solution to data scarcity by enabling simultaneous training of a single model that generalizes across multiple tasks[12]. It takes advantage of the many small- and medium-sized datasets in biomedical imaging, efficiently utilizing different label types and data sources to pretrain image representations that are applicable to all tasks, enabling deep learning for domains with sparse data. MTL has been applied to biomedical image analysis in various ways, such as training on multiple small- and medium-sized datasets from distinct tasks, specifically limited to classification[13] or segmentation[14]. Additionally, MTL has been used with multiple label types for individual images, demonstrating that sharing features across label types enhances task performance[15].

In this Article, to combine multiple datasets with different label types for large-scale pretraining, we introduce a multi-task training strategy and a corresponding model architecture specifically designed to address the data scarcity problem in biomedical imaging by learning versatile representations across diverse modalities, diseases and label types. To cope with the memory constraints encountered in large-scale multi-task learning, our approach employs a gradient accumulation-based training loop, the scaling of which is almost unconstrained by the number of training tasks. Based on this, we trained a fully supervised foundational model for biomedical imaging named UMedPT, using 17 tasks and their original annotations. Each task consisted of training and test sets with its label type, for example, classification, segmentation or object detection. A study overview is presented in Fig. 1.

UMedPT consistently matched or outperformed the pretrained ImageNet network in in- and out-of-domain tasks, while maintaining a strong performance with fewer training data in both the direct application of image representations (frozen) and fine-tuning settings. We also compared our model with external reference results and demonstrated the robustness of UMedPT through external validation.

Serving as a basis for future advancements in data-scarce domains, UMedPT opens perspectives to extend the application of deep learning in medical fields where collecting large cohorts is particularly challenging, such as rare diseases and pediatric imaging.

## Results

We evaluated our models according to three benchmarks. The first, the 'in-domain benchmark', aimed to determine the performance of UMedPT on tasks closely related to its pretraining database. The second, the 'out-of-domain benchmark', aimed to assess how well UMedPT adapted to new tasks outside its immediate training domain. The third, the MedMNIST benchmark[16], was used to evaluate the proposed multi-task training strategy on a separate training database and, independently, to test UMedPT. We then compared our findings with previously published results from the in-domain, out-of-domain and MedMNIST target tasks considered to be the state of the art.

Where applicable, we evaluated the performance of UMedPT in data-scarce scenarios by training it with varying amounts of original training data, ranging from 1% to 100%, and report the average results gathered from five repeated runs for each experimental set-up. All networks were trained with a frozen encoder and subsequently in a fine-tuning setting with the same training scheme and hyperparameters.

For our in-domain and out-of-domain clinical benchmarks, we conducted ablation studies for UMedPT to investigate the effects of the variable input image size of UMedPT compared to the fixed input image size of 224 × 224 with UMedPT-fixed, and whether to include trainable parameters in the layer normalizations within its architecture with UMedPT-affine, as detailed in Supplementary Section 1. We found that a variable input size was beneficial for the performance of UMedPT, and UMedPT-affine had a minor impact on the results. In addition, we compared the performance of UMedPT with a variant that was trained only with the classification tasks UMedPT-clf, as described in Supplementary Section 2. This showed a great benefit of including segmentation and object detection tasks, especially for other similar tasks.

### Model overview

Figure 2 shows the architecture of our neural network, which consists of shared blocks, including an encoder, a segmentation decoder and a localization decoder, along with task-specific heads. The shared blocks were trained to be applicable to all pretraining tasks, facilitating the extraction of universal features, and the task-specific heads handled label-specific loss computation and predictions. Our tasks included three supervised label types: object detection, segmentation and classification. Classification tasks, for instance, can model binary biomarkers, segmentation tasks can extract spatial information, and object detection tasks can be used, for example, to train biomarkers based on cell quantities.

### Comparison of UMedPT with ImageNet for in-domain tasks

We compared UMedPT to results obtained with weights pretrained on ImageNet-1K. In both classification tasks, UMedPT was able to match the best performance of the ImageNet baseline over all configurations using only 1% of the original training data. Notably, our model achieved higher performance with a frozen encoder compared to our model with fine-tuning, as shown in Fig. 3.

In our colorectal cancer (CRC) tissue classification (CRC-WSI), we classified CRC tissue from microscopic whole slide images (WSIs) into nine different classes, including adipose tissue, normal colon mucosa and colorectal adenocarcinoma epithelium[17]. For CRC-WSI, ImageNet achieved an average F1 score of 95.2% on the unseen test set using all of the training data and fine-tuning. UMedPT achieved a comparable performance with 1% of the training data and a frozen encoder (95.4% F1 score; Fig. 3). When the training dataset size was increased to 50% and 100% and the models were fine-tuned, the results converged to approximately the same F1 score across all methods (Supplementary Table 3). Surprisingly, for UMedPT, increasing the training data beyond 1% did not enhance the model's performance and sometimes tended to degrade it. Notably, it did not matter which 1% were picked, as the final performance had a low variance. We further investigated whether this could be due to catastrophic forgetting of the well-generalizing pretrained features or overfitting to the training data, and found the phenomenon to be dataset-specific (Supplementary Section 6).

In our Pneumo-CXR investigation, we focused on diagnosing pediatric pneumonia[18]. Here, UMedPT outperformed ImageNet across all dataset sizes. The best performance of UMedPT was achieved using 5% of the data (~250 images) and frozen features, resulting in an F1 score of 93.5%. The best ImageNet performance (90.3% F1 score, 100% of the data) was matched with the smallest split (1% of the data, ~50 images; Supplementary Table 3).

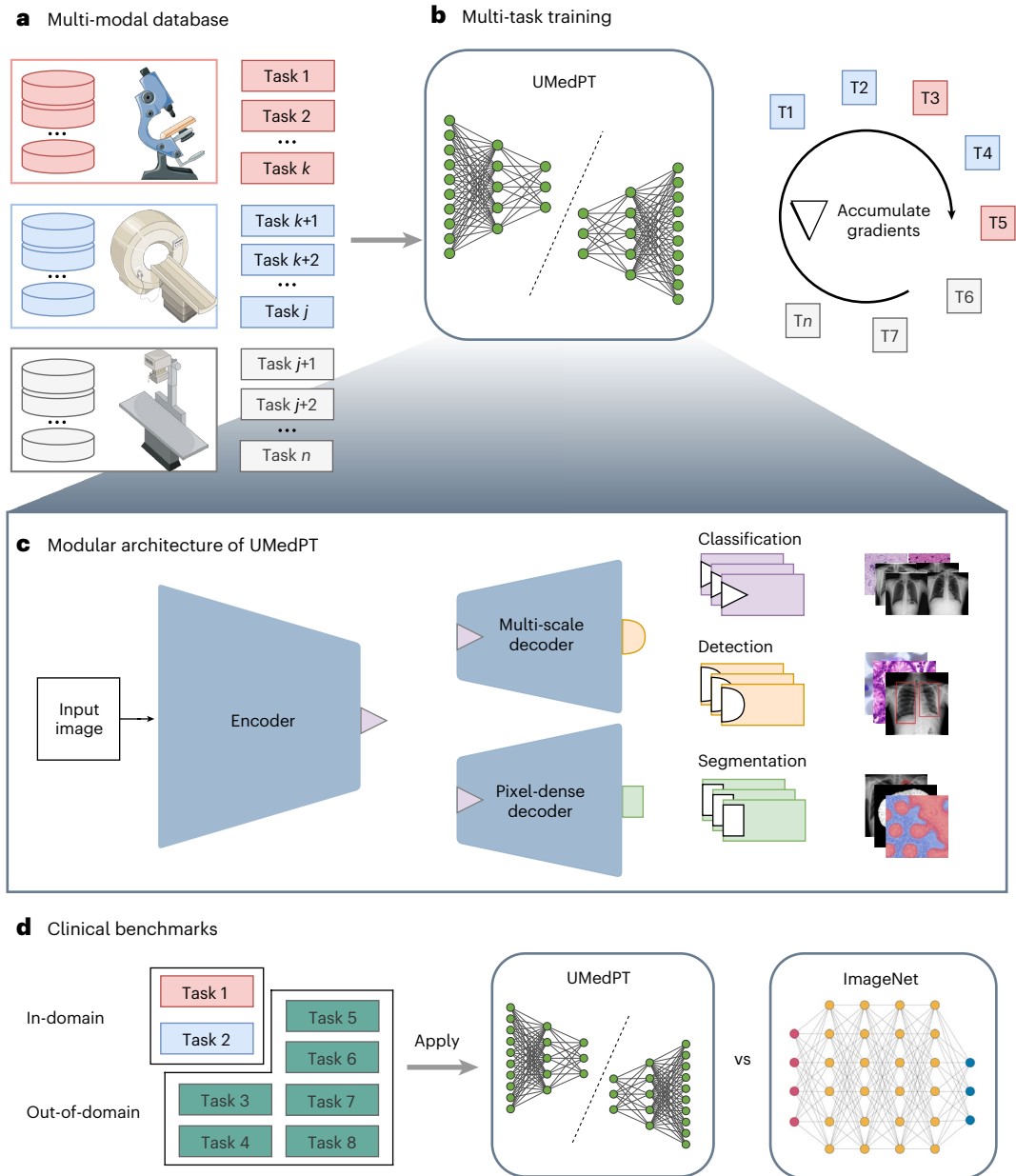

**Fig. 1 | Study overview.** Illustration of the organization of the study and the multi-task learning approach. **a**, Heterogeneous data sources from histology (red), tomographic imaging (blue) and X-rays (gray) form the generalized training database. **b**, The pretraining process trains generally applicable neural network components (shared blocks). Our training loop, based on gradient accumulation and detailed in Algorithm 1, allows for joint training incorporating all pretraining tasks at each optimization step despite different label types. **c**, Depending on the label type of the current task being processed, our modular architecture is assembled from a set of three shared blocks. More information about each shared block is provided in Fig. 2. Icon sources: CXR[54], WSI[55] and MRI[56].

**d**, The applicability of the pretrained neural components to medical imaging was evaluated using a diverse benchmark consisting of several clinically relevant small datasets categorized as in-domain or out-of-domain. The performance of UMedPT was compared with that of ImageNet-1K pretraining. Image credits: **a**, device images, Servier Medical Art under a Creative Commons license CC BY 4.0; **b**,**d**, neural network created with BioRender.com. Panel **c** reproduced with permission from: middle back, ref. 57, Cancer Imaging Archive; middle middle, bottom front, ref. 55 courtesy of the authors; top middle, top front, middle front, bottom back, ref. 58, Kaggle; bottom middle, ref. 56, IEEE.

We used the NucleiDet-WSI dataset[19] to detect nuclei in ten different cancer types from WSI. The best ImageNet performance was achieved using 100% of the data together with fine-tuning, resulting in a mean average precision (mAP) of 0.71. UMedPT was able to replicate this performance with 50% of the training data and no fine-tuning. However, fine-tuning tended to improve the results for both models. Interestingly, compared to ImageNet, UMedPT showed superior performance across all data fractions with both fine-tuning and a frozen pretrained model. This resulted in a

maximum performance of 0.792 mAP when using the full training dataset and fine-tuning.

**Comparison of UMedPT with ImageNet for out-of-domain tasks**

In the out-of-domain benchmark, UMedPT compensated a data reduction of 50% or more across all classification datasets in the benchmark when the encoder was frozen, as detailed in Supplementary Table 4. With fine-tuning, ImageNet's performance consistently improved if more

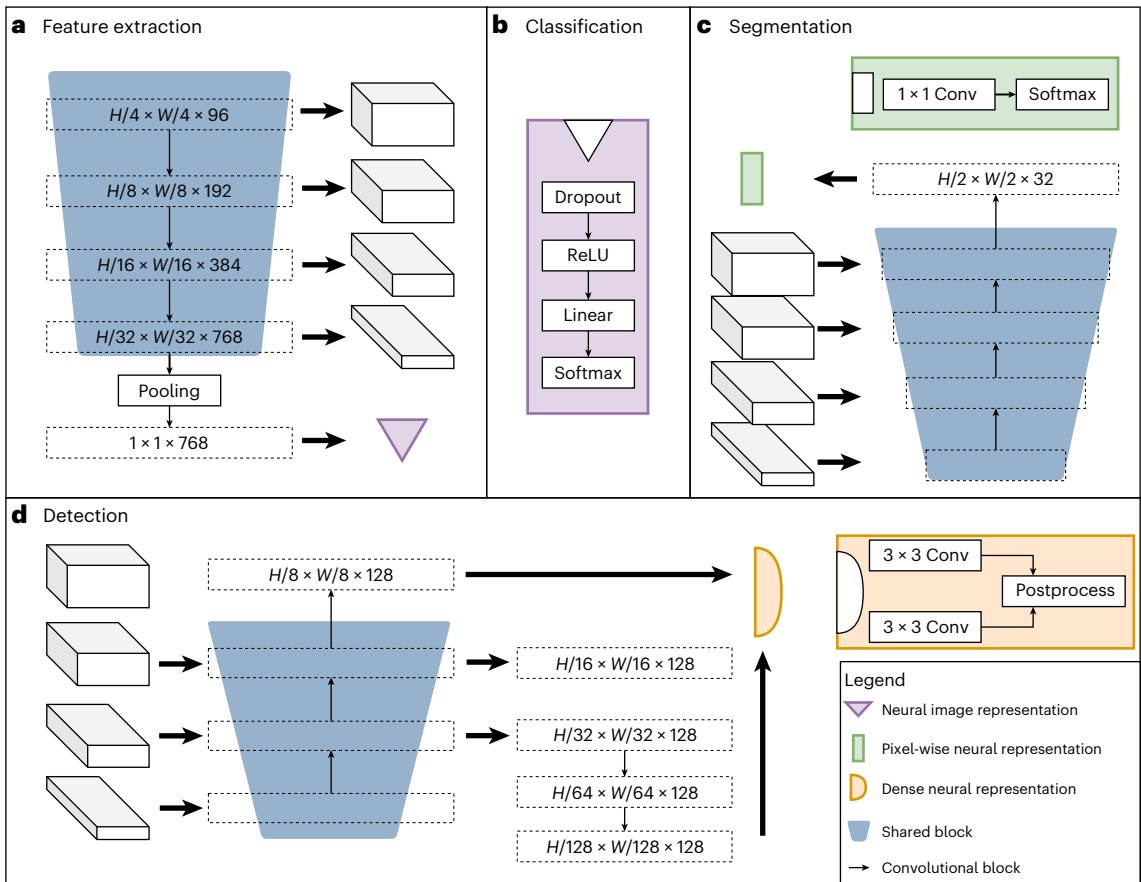

**Fig. 2 | Architecture of UMedPT. a**, Features are extracted from an image of size $H \times W$ through a shared encoder. **b**, Classification heads take the neural image representation of an image and apply a single linear layer. **c**, A shared pixel-wise decoder processes the multi-scale feature maps and returns an embedding per pixel. Segmentation heads employing the popular U-Net spatial decoding strategy for handling segmentation features generate the prediction. **d**, The multi-scale decoder uses the feature maps and transforms them into features for box regression and box classification. The FCOS-based detection head generates the final prediction.

data were used. Here, for two out of five datasets, UMedPT was able to match the performance of ImageNet using only 50% or less of the data, even when fine-tuning was applied (Fig. 4 and Supplementary Table 5). In the following we highlight key findings in the out-of-domain tasks.

In the task of diagnosing tuberculosis from chest X-rays (Tuber-CXR)[20], UMedPT delivered the highest average result. This was achieved by fine-tuning the model and using just 10% of the data, resulting in an F1 score of 96.3%. Adding more training data did not further improve the score for our model (Fig. 4). To match the overall best average result of ImageNet, UMedPT required 5% of the data with fine-tuning and 50% of the data with a frozen encoder.

We used the CNS-MRI[21] dataset to train our system to diagnose central nervous system (CNS) neoplasms from magnetic resonance imaging (MRI) scans. ImageNet with frozen features achieved an F1 score of 89.0%. UMedPT was able to match this score using 5% of the training data. With the full training set and fine-tuning, UMedPT achieved the top F1 score of 99.3%.

The BC-Bach-WSI dataset was used for breast cancer classification in WSIs. Using the frozen encoder, ImageNet achieved an F1 score of 72.9%. UMedPT obtained this score with 50% of the data, resulting in an F1 score of 78.0%. Here, the best results were achieved using fine-tuning.

The BC-BHis-MIC dataset was used for breast tumor classification into benign and malignant in microscopic (MIC) images. The best mean ImageNet result was achieved with 100% of the data and fine-tuning, resulting in an F1 score of 98.4%. UMedPT also achieved an F1 score of 98.4%. When using a frozen encoder, ImageNet achieved an F1 score of 82.3%. UMedPT was able to match this score using 50% of the data.

The PolypSeg-RGB dataset focused on the segmentation of polyps in coloscopy images. When using the entire dataset for fine-tuning, ImageNet achieved its best average result, demonstrating a mean intersection over union (mIoU) of 0.905. Here, UMedPT achieved an mIoU of 0.911. The model pretrained with ImageNet showed better results when the encoder was frozen, as presented in Extended Data Fig. 1c. The best performance across all fractions was achieved by UMedPT with fine-tuning. In addition, although UMedPT with fine-tuning outperformed ImageNet for all fractions, the strongest advantage occurred with 1% of the data (0.797 ± 0.09, compared to 0.683 ± 0.144 for ImageNet).

**Comparison of UMedPT with external reference results**

We next compared the performance of UMedPT to outcomes reported in the literature. When using the frozen encoder configuration, UMedPT surpassed the external reference results in the majority of tasks. In this setting, it also outperformed the average area under the curve (AUC) in the MedMNIST database[16]. Notably, the tasks where the frozen application of UMedPT did not outperform the reference result were out-of-domain (breast cancer classification BC-Bach-WSI and CNS neoplasia diagnosis CNS-MRI). With fine-tuning, pretraining with UMedPT exceeded the external reference results across all tasks (Table 1).

**Robust image representations across clinical centers**

To further assess the robustness of UMedPT's image representations across diverse settings, it was applied in the classification task of the SemiCOL challenge[22]. This task required the classification of CRC histopathology images into tumor and healthy. It enabled the evaluation of

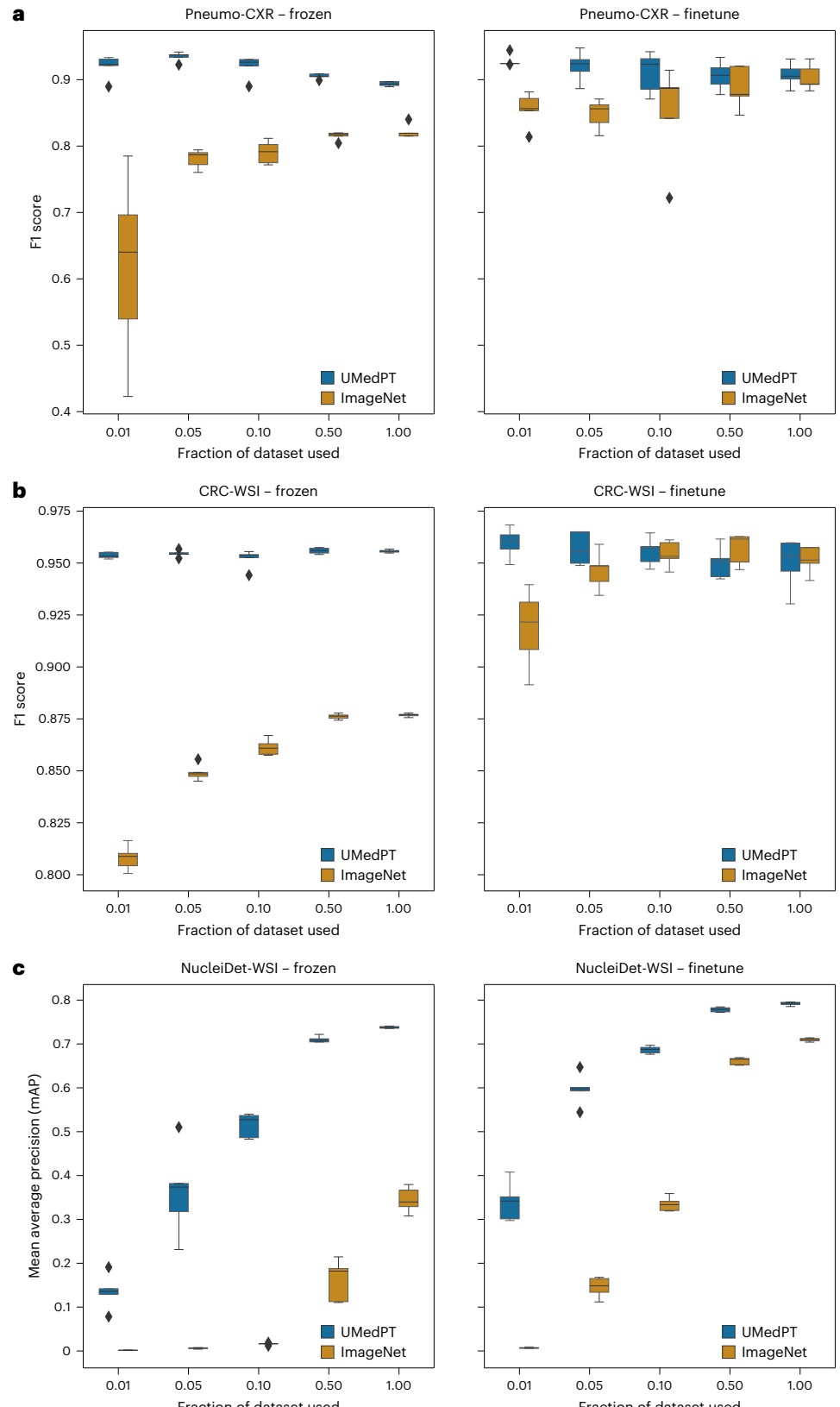

**Fig. 3 | Results for in-domain tasks. a**, In the diagnosis of pneumonia (Pneumo-CXR), UMedPT matched the full fine-tuned performance of ImageNet, even with a frozen encoder and a reduced dataset size (1%). **b**, CRC-WSI was the only target task where the training dataset was also part of the pretraining. Here, performance was stable across dataset fractions with a frozen encoder. When the encoder was fine-tuned, performance decreased to the result obtained with ImageNet pretraining. **c**, For NucleiDet-WSI, an object detection task for counting nuclei in WSIs, UMedPT outperformed ImageNet across all training settings. Best performance was achieved with 100% of the training data and fine-tuning. In each setting, five independent trainings were derived for each training subset and method. The middle line of the boxes represents the median, the boundaries are the Q1 and Q3 quartiles, the whiskers extend to 1.5 times the interquartile range (IQR), and outliers beyond are shown as single points.

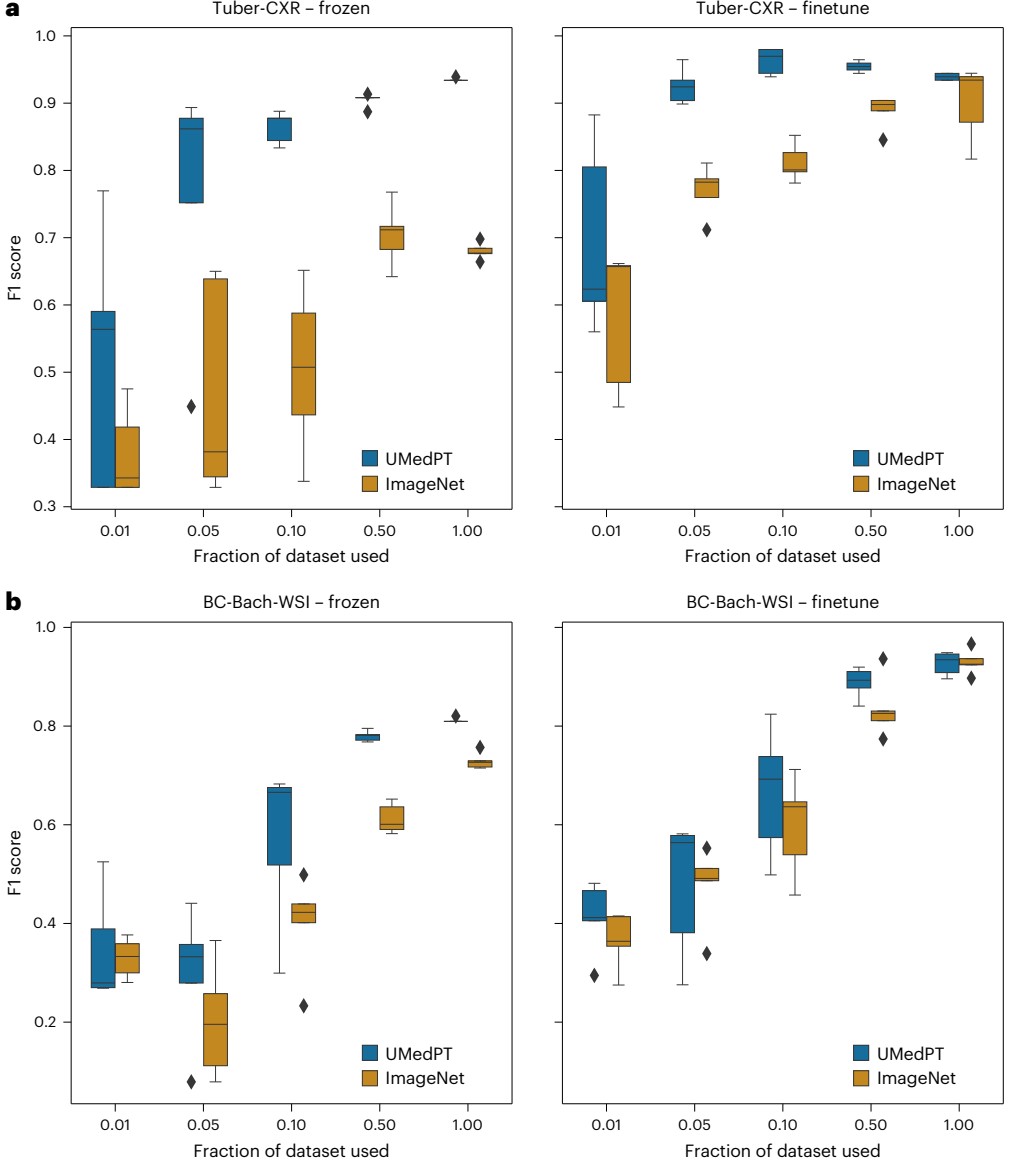

**Fig. 4 | Results for out-of-domain tasks. a**, For tuberculosis diagnosis (Tuber-CXR), UMedPT enables the full ImageNet performance to be matched, even with frozen encoder and a reduced amount of data (50%). **b**, In BC-Bach-WSI, fine-tuning was necessary for high performance. In the frozen setting, half of the training data were required to match the ImageNet frozen result. In each setting, five independent trainings were derived for each training subset and method. The middle lines of the boxes represent the median, the boundaries are the Q1 and Q3 quartiles, the whiskers extend to 1.5 IQR, and outliers beyond are shown as single points.

the model's robustness across multiple clinical centers, most of which were not part of the training data.

Across all participating clinical centers, our UMedPT-based model achieved an AUC of 99.7%. Furthermore, by exploiting the frozen features of UMedPT and the compact size of the training dataset (consisting of 499 images), we were able to train our model for CRC classification in less than 10 min.

**Comparison of 2D multi-task and 3D single-task models**
Independently of the above training database, we evaluated the training strategy on MedMNIST. MedMNIST contains a variety of standardized and downscaled biomedical image datasets, including both two-dimensional (2D) and 3D images. To assess the impact of transforming inherently 3D medical image data into 2D slices for pre-training, we evaluated single-task learning with 2D data, single-task learning with original 3D data[16] and multi-task learning with 2D data. For our multi-task and single-task trainings, we converted the 3D data

into 2D slices by slicing through the last dimension. We then applied a multiple-instance learning classification task that was based on a weighted averaging operation over the neural representations of the slices. When only the 3D tasks were considered, the average accuracies were as follows:

- 83.22 ± 1.61% for single-task learning pretrained using ImageNet;
- 83.76% for the single-task MedMNIST 3D convolutional neural network (CNN)[16];
- 86.46 ± 1.13% for multi-task learning.

The results show that single-task 3D CNNs perform better than single-task 2D CNNs pretrained using ImageNet. However, multi-task 2D networks outperform the single-task 3D CNNs. The details are presented in Table 2.

In this context, we also investigated the performance difference between CNN and transformer architectures for our multi-task learning

**Table 1 | Data required to match the state of the art**

| Task | Reference results | UMedPT (%) | |
|---|---|---|---|
| | | Frozen | Fine-tuning |
| CRC-WSI | >94% accuracy[17] | 1 | 1 |
| Pneumo-CXR | 92.8% accuracy[18] | 5 | 1 |
| Tuber-CXR[a] | 82% accuracy[74] | 10 | 5 |
| Tuber-CXR[a] | 98.0% AUC[11] | 50 | 10 |
| CNS-MRI[a] | >96% accuracy[21] | – | 50 |
| BC-Bach-WSI[a] | 87% accuracy[66] | – | 50 |
| BC-BHis-MIC[a] | 88% F1 score[52] | 100 | 10 |
| PolypSeg-RGB[a] | 0.778 mIoU[68] | 50 | 1 |
| PolypSeg-RGB[a] | 0.9051 mIoU[53] | – | 100 |
| MedMNIST mean AUC | See Supplementary Fig. 3[16] | 100 | 10 |
| MedMNIST mean ACC | See Supplementary Fig. 3[16] | – | 10 |

The table shows the amount of data required by UMedPT to match state-of-the-art performance on tasks from different imaging domains. [a]Datasets compared across different test splits. Unless otherwise stated, all results in the main text were obtained with UMedPT.

strategy, as presented in Supplementary Section 4, and aspects of the training algorithm (Supplementary Section 5). We found that the Swin transformer architecture has a minimal positive impact, as shown in Extended Data Fig. 2a. Regarding training schemes, we found that without gradient accumulation, both convergence and performance were worse with our training strategy (Extended Data Fig. 2b).

In addition to 3D classification, we also investigated the direct applicability of our training method to 3D lung nodule segmentation in computed tomography (CT) scans[23]. The experiments showed that a large 2D pretraining and the larger 2D spatial context it enables can compensate for the loss of 3D context, as detailed in Supplementary Section 7.

## Discussion

We have demonstrated that UMedPT's knowledge can be effectively transferred to unseen target tasks. Similarly, previous studies have reported that pretraining on a large dataset[11] or multi-task pretraining[13] can improve the performance of models on small, unseen datasets. In the field of medicine, this is especially important for rare and pediatric diseases, where the quantity of available images is often too small to effectively train deep neural networks. However, the performance advantage of UMedPT in in-domain tasks compared to out-of-domain tasks indicates that it is not yet entirely universal for all biomedical imaging applications, necessitating a broader scale of training. The extent to which such pretraining should be scaled remains an open question, as we did not observe any saturation with respect to the data, the variety of tasks or the number of different label types that can be included into a single supervised multi-task training.

Alternately, self-supervised pretraining can be used to improve the data efficiency for target tasks, as demonstrated with RAD-DINO[24]. However, recent literature suggests that label-supervised pretraining for imaging typically outperforms self-supervised pretraining, empirically[11,25] and theoretically[26]. Nonetheless, it offers value in regularizing models and might help in further reducing the required volume of labeled data. Our approach can be extended to include an arbitrary number of self-supervised tasks into the pretraining, which may further enhance the generalizability of UMedPT, especially in domains where abundant data are available, but labeling is difficult or costly.

Training artificial intelligence (AI) models from scratch can be computationally intensive. Here, foundational models such as UMedPT in a frozen configuration may enable efficient feature extraction without additional training. Frozen features from pretrained networks have solved in-domain classification in pathology, radiology and natural images[13,27–29]. However, the performance declined when frozen models were used for out-of-domain evaluations. Our in-domain benchmark results are consistent with these findings, suggesting that frozen features should be the primary consideration for in-domain tasks. Moreover, we demonstrate that a single network can effectively extract features across multiple domains, extending the applicability of frozen features within the in-domain context.

We show that for out-of-domain tasks, fine-tuning can outperform the frozen configuration if there are enough data or significant differences between the target task data distribution and the pretraining distributions. Other multi-task studies[13] have observed that fine-tuning multi-task models or pretrained ImageNet models yield comparable performance. However, even with fine-tuning, ImageNet did not outperform UMedPT in any of the medical applications evaluated. UMedPT showed advantages in the full data scenario with fine-tuning in three out of five out-of-domain tasks. This could potentially be due to either the larger scale of UMedPT's training, which resulted in a well-generalizing base for fine-tuning, or the possibility that these out-of-domain datasets were not sufficiently large at full size, making them ideal use cases for UMedPT.

In some cases we observed that the performance of UMedPT decreased as the size of the training dataset increased. We investigated both catastrophic forgetting[30] of the well-generalizing features learned during pretraining, as well as overfitting to the training set due to using all data for training instead of a validation set for model selection (Supplementary Section 6). The inconsistency of the results raises questions about the best practices for using foundational models in tasks with varying data sizes and varying degrees of similarity to the pretraining database. There were tasks that performed best with model selection using a validation set, and tasks that performed best with all the data used for training. Similarly, some tasks performed best with the frozen training setting, and others with fine-tuning of all pretrained parameters of UMedPT. Recently, more sophisticated fine-tuning strategies have been proposed for foundational models in natural language processing, such as BitFit[31] or LoRA[32], where only a specific subset of parameters is fine-tuned. A training configuration targeted specifically to foundational vision models could combine the strengths of the different training configurations.

Medical images vary in size, challenging deep learning methods, which typically require uniform sizes. Homogeneous downsampling, however, resulted in reduced performance when comparing UMedPT and UMedPT-fixed, which is consistent with previous findings[14]. For tasks that benefit from a large image size, our training method allows flexibility in choosing the optimal image size for each task, thus avoiding the problem of predefining it.

In addition to the ability to handle arbitrary image sizes, for the UMedPT encoder we needed a general base architecture capable of generating multi-scale feature maps, a feature found in both convolutional neural networks (CNNs) and Swin transformers[33]. Our experiments with MedMNIST showed a minimal difference between a CNN and a Swin transformer, but slightly in favor of the latter. This suggests that the proposed pretraining strategy can be implemented with both convolutional and transformer-based encoders, with the literature showing that CNNs can also work well with large datasets of full-sized images[34].

Differences in imaging modalities, scanners, annotations or patient populations can make models fail when applied to the data of different clinical centers[35]. Foundational models should be robust to multi-center variances, thereby improving their ability to generalize. We tested this using the SemiCOL challenge[22], which includes data from several research centers, most of which were not included in the original training dataset. UMedPT outperformed all other teams in the task of classifying CRC histopathology images into tumor and healthy, achieving an AUC of 99.7%, surpassing the next best models with AUCs of 97.3% and 93.6%. Thus, our pretraining method makes models based on frozen encoders viable competitors. This is particularly beneficial

**Table 2 | MedMNIST test performance**

| CNN | ACC MTL (%) | AUC MTL (%) | ACC ST (%) | AUC ST (%) | ACC ref. (%) | AUC ref. (%) |
|---|---|---|---|---|---|---|
| BloodMNIST | 94.31±0.41 | 99.64±0.02 | 94.49±0.41 | 99.70±0.06 | 95.60 | 99.70 |
| BreastMNIST | 87.18±0.41 | 86.81±1.58 | 81.54±1.59 | 83.86±1.49 | 81.20 | 85.70 |
| ChestMNIST | 94.76±0.01 | 73.34±0.30 | 93.42±0.29 | 66.95±0.30 | 94.70 | 76.90 |
| DermaMNIST | 76.55±0.67 | 91.97±0.20 | 76.81±0.42 | 92.00±0.33 | 73.50 | 91.30 |
| OCTMNIST | 73.32±1.47 | 95.53±0.31 | 75.16±2.01 | 95.03±0.42 | 76.20 | 95.20 |
| PathMNIST | 85.97±1.94 | 98.58±0.26 | 87.92±1.03 | 98.78±0.03 | 91.10 | 99.00 |
| PneumoniaMNIST | 91.76±0.93 | 97.91±0.25 | 90.26±1.24 | 97.97±0.18 | 85.40 | 94.80 |
| AdrenalMNIST3D | 83.89±0.88 | 86.85±1.95 | 80.87±3.40 | 76.96±17.78 | 74.50 | 82.80 |
| NoduleMNIST3D | 85.94±0.97 | 85.95±1.45 | 86.13±0.54 | 90.87±0.71 | 84.70 | 87.50 |
| OrganMNIST3D | 87.11±1.25 | 99.00±0.25 | 86.82±2.74 | 98.69±0.25 | 88.30 | 99.40 |
| SynapseMNIST3D | 81.08±1.50 | 81.81±1.32 | 73.52±0.55 | 65.62±6.14 | 79.50 | 85.10 |
| VesselMNIST3D | 94.29±0.45 | 95.64±0.75 | 88.74±0.00 | 69.60±2.80% | 91.80 | 90.70 |
| **Transformer** | | | | | | |
| BloodMNIST | 95.96±0.11 | 99.77±0.01 | 95.70±0.30 | 99.76±0.04 | – | – |
| BreastMNIST | 86.92±1.04 | 86.26±1.38 | 86.41±1.24 | 86.81±0.78 | – | – |
| ChestMNIST | 94.75±0.01 | 75.19±0.35 | 93.69±0.10 | 68.04±0.42 | – | – |
| DermaMNIST | 79.09±0.45 | 93.04±0.30 | 79.20±0.57 | 93.34±0.31 | – | – |
| OCTMNIST | 74.08±1.31 | 95.43±0.41 | 73.62±0.92 | 93.55±0.60 | – | – |
| PathMNIST | 90.60±0.48 | 99.15±0.06 | 92.11±0.98 | 99.34±0.13 | – | – |
| PneumoniaMNIST | 91.54±0.48 | 98.22±0.37 | 88.85±1.62 | 97.69±1.01 | – | – |
| AdrenalMNIST3D | 82.01±1.11 | 85.28±0.77 | 76.85±0.00 | 66.54±8.03 | – | – |
| NoduleMNIST3D | 83.81±1.74 | 86.39±2.37 | 83.29±2.56 | 84.52±5.25 | – | – |
| OrganMNIST3D | 84.89±1.17 | 98.81±0.07 | 90.52±2.06 | 99.28±0.12 | – | – |
| SynapseMNIST3D | 83.81±0.31 | 85.75±0.89 | 73.01±0.00 | 57.11±4.79 | – | – |
| VesselMNIST3D | 93.66±0.31 | 93.94±1.40 | 88.74±0.00 | 66.98±5.30 | – | – |

Multi-task learning (MTL) networks were trained using all tasks, including both 2D and 3D tasks. Single-task (ST) networks were trained independently. Metrics are reported as mean±s.d. as a percentage. The reference accuracy (ACC ref.) and AUC (AUC ref.) are taken from the publication associated with the MedMNIST database[16].

for complex data types such as WSIs, where fine-tuning deep neural networks can be challenging due to memory and data constraints. Additionally, based on UMedPT, our challenge model could be trained in less than 10 min, which is particularly advantageous for interactive training tools. However, the evaluation of the challenge was limited to a single histological imaging task, and multi-center robustness is a major obstacle for deep learning systems in tomographic and X-ray imaging[35]. A systematic assessment of UMedPT for multi-center robustness, including the use of MRI and X-ray data as well as training speed, poses a task for future studies.

## Methods

In deep learning, training of models is performed by optimizing an objective function (loss function) that measures the difference between ground-truth labels and the result of the current models' iteration. The gradient of the loss function determines the extent of the adjustments needed for each model parameter.

In the presented multi-task learning framework, the overall loss of the model was defined as the sum of the losses of all tasks that were evaluated simultaneously.

Every label type required the definition of a task-specific architecture component and an objective function computing its loss. For UMedPT, we combined these different components into a single model to solve many pretraining tasks simultaneously. These pretraining tasks integrated a large variety of the publicly available biomedical image data, including their annotations, into a single foundational training. This training resulted in a shared model compatible with all of the pretraining tasks.

We addressed challenges such as memory constraints, varying input sizes and label types to integrate a diverse set of small and medium-sized datasets for training UMedPT. The model's design accommodated a range of task types, including classification and dense prediction tasks like segmentation, and allowed each task to operate using its optimal patch size and resolution, respectively.

### Multi-task training strategy

A limiting factor in scaling currently established multi-task learning approaches is the increasing memory demand, which is proportional to the number of tasks. This memory requirement is caused by processing all tasks in parallel during a single network pass. To address this challenge, we developed a training strategy for UMedPT that mostly decouples the number of training tasks from the memory requirements.

We used PyTorch[36] to create an independent architecture, or 'computational graph', for each task. This graph was dynamically constructed so that each label type could be solved by a different architecture, but still shares almost all model parameters. For example, in the case of UMedPT, a U-Net[37] for segmentation labels was assembled by combining the shared Swin transformer encoder with the shared pixel-dense decoder and a small task-specific part. To combine the individual graphs, we used gradient accumulation (GA) before the optimization step, as described in Algorithm 1. GA allowed us to establish a training scheme wherein a single update step could consist of heterogeneous tasks in any order. This allowed the training strategy to use an adaptive architecture, where each type of label can be solved by a specialized combination of model components, such as a U-Net for segmentation labels[37].

GA is a common method for incorporating more data into a single optimization step. In the case of our multi-task learning strategy, unlike traditional deep multi-task learning, GA allows the weights and gradients of the shared part of the model to be stored only once, rather than duplicated for each task. Additionally, only the activations for one task are kept in memory at a time. They are discarded after the backward pass of each task, rather than being stored for all tasks simultaneously. Therefore, the only memory requirements that increase with the number of tasks are related to the gradients of each task's head. As the shared section of the model represents the majority of the total model size, this approach allows for multi-task learning across many tasks, even on hardware with limited computational power.

Unlike traditional training schemes, which merge multiple tasks in a batch of samples, GA enables flexible task scheduling. Each optimization step can therefore consist of multiple tasks and even multiple instances of the same task, enhancing the versatility of the proposed training approach.

When training a model on many tasks, the size of the respective dataset should not implicitly influence the weight of the task in the overall model. To accommodate datasets of different sizes, we implemented an 'infinite task sampler', which yielded one training batch of each task for every optimization step. Problematically, for 3D volumes, the number of image instances (individual 2D images that can be used directly for pretraining) used for training could depend on the randomly chosen axis of slicing, while for gigapixel images, the random zoom level could influence the number of image instances used for training. Our task sampler independently restarted the data loading for a task once all of its data points had been used. As a result, no information on the dataset's length was needed beforehand, allowing each epoch to have a different length depending on data augmentation. When training a multi-task model with GA, the model parameters were updated according to the sum of the model's gradients. Because summation is commutative, the order of tasks within an optimization step does not affect the outcome.

We uniformly used the AdamW optimizer[38] for all parameters. Following the training settings of ref. 33 for ImageNet-1K training, we used a learning rate of 0.001 and a weight decay of 0.05 for all transformer-based models.

**Algorithm 1.** The training loop processed tasks and their associated batches in the order given by the task sampler. For each step, a gradient was computed by evaluating the objective function of one task with respect to one of its batches. These gradients were accumulated until the task sampler initiated an optimization step. At this point, the model parameters were updated considering all tasks since the last update, utilizing the accumulated gradient. After this, the gradient buffer was cleared for the next cycle.

```
 1: procedure Train (stepsPerEpoch, sharedblocks, tasksampler,
        optim)
 2:   prepareSharedBlocks()                    ▷norm&task-specific
                                                  modules
 3:   optim.clearGradients()
 4:   for step ← 0 to stepsPerEpoch do
 5:     (batch, task) ← tasksampler.next()
 6:     loss ← task.computeLoss(batch, sharedblocks)
 7:     loss.backward()                         ▷Accumulate
                                                  gradients
 8:     if isUpdateStep(step) then              ▷ E.g. after processing
                                                  each task
 9:       optim.updateParams()
10:       optim.clearGradients()
11:     end if
12:   end for
13: end procedure
```

## Architecture

Our tasks vary in their label types, each requiring problem-specific architectures. We thus used a fixed-size embedding for classification tasks, designed to encapsulate features that are useful across all tasks. For segmentation tasks we implemented a U-Net-like encoder–decoder scheme to learn multi-scale features. Additionally, object detection using the fully convolutional one-stage (FCOS)[39] detector required features produced by a feature pyramid network[40].

To avoid wasting resources, as not all features are required for every task, we created a modular architecture. We hypothesized that parameters should be largely shared across tasks. To address this, we placed most parameters into a shared encoder. To compute the necessary types of feature, we then developed a pixel-dense decoder and a multi-scale decoder.

Our architecture supported encoders with configurable settings for image embedding dimensionality, stride (modulating the spatial range of feature pyramid levels) and feature pyramid depths. Given that these settings are common in computer vision, our framework was able to integrate open-source encoder architectures. For segmentation tasks, we used a pixel-dense decoder that upsampled the feature pyramids in a U-Net style to generate pixel features. For object detection, a multi-scale decoder was used to create feature maps from every pyramid level.

For the classification tasks, we directly used the image embedding from the encoder. This was computed using global average pooling from the lowest level of the feature pyramid, allowing us to handle variable input sizes. This approach is consistent with the ImageNet baseline. Segmentation tasks employed the shared pixel-dense decoder, whereas object detection tasks processed the encoder's output via a shared feature pyramid network. The output for each label type was computed using a single linear layer, or a single convolutional layer for dense prediction tasks.

In general, our proposed training loop can be used with any neural network, and UMedPT's decoders are compatible with any encoder that can generate multi-scale feature maps. We chose Swin transformers as the encoder for UMedPT[33], which introduced a shifted windowing scheme that improved the efficiency of vision transformers with respect to the image input size. We also investigated the compatibility of CNNs with our multi-task training loop and include an additional comparison in Supplementary Section 4.

Regardless of the chosen architecture, normalization has been shown to be essential for accelerating the training process[41,42]. Batch normalization[41] is a widely used normalization technique and is the default normalization layer within the ResNet CNN that we used for the MedMNIST benchmark[16], UMedPT's segmentation decoder[37] and UMedPT's object detection decoder[40]. However, in our experiments, batch normalization led to poor performance. One assumption when using batch normalization is that all input batches follow the same distribution. When combining different tasks and datasets, this assumption no longer holds.

Although we observed that normalization layers improved the training speed, we believed that they would underperform similarly in layer normalization due to the ineffectiveness of trainable parameters in batch normalization. Consequently, we took advantage of the tree-like property of PyTorch neural networks and recursively replaced the original normalization layers in all shared blocks with layer normalization[42], which by design do not require inter-task computation, and excluded trainable parameters. This modification enabled us to integrate state-of-the-art models to concurrently generalize across multiple training tasks despite possibly incompatible normalization layers. Notably, the Swin transformer encoder used in UMedPT already employed layer normalization, which comes by default with trainable parameters.

As a result, the default configuration of UMedPT excludes trainable parameters in its normalization layers. First, previous studies have

shown that trainable parameters such as bias and gain within layer normalization layers increase the risk of overfitting and generally do not contribute to improved performance[43]. Second, given the ineffectiveness of trainable parameters in batch normalization, we hypothesized that they might similarly underperform with layer normalization.

To empirically assess the impact of excluding such trainable parameters in UMedPT, we compared it to a variant of our model UMedPT-affine that included trainable bias and scaling layer normalization parameters, which is the default for the Swin transformer, the UMedPT encoder. Thus, UMedPT used layernorms without parameters in the form $y = \frac{x-\mu}{\sigma}$, where $\sigma = \sqrt{x + \epsilon}$, and $x$ was the input. The mean $\mu$ and standard deviation $\sigma$ were computed over all channels of an input, but not over the batch dimension. The UMedPT-affine variant again contained trainable parameters, including a bias $\beta$ and a scaling factor $\gamma$ in the form $y = \gamma \frac{x-\mu}{\sigma} + \beta$ for each channel. Similar to UMedPT-fixed, UMedPT-affine was only trained and evaluated with images downscaled to 224 × 224 pixels.

## Data loading from diverse sources

To evaluate the ability of the model to learn multi-modal representations, we integrated a diverse range of biomedical imaging data types—including microscopic pyramid gigapixel 2D images, standard 2D images (both grayscale and color) and 3D tomographic images—into a single network. Each data type requires unique pre-processing and domain-specific augmentations for a universally applicable deep learning solution. To accommodate these different data types, the encoder of UMedPT used a standardized 2D image input format. This required the conversion of each data type into a 2D format.

To ensure compatibility between different data types, we normalized all pixel intensities to a range between 0 and 1. For images with values ranging from 0 to 255, we divided them by 255. For 3D images, we normalized the maximum in the volume to 1 and the minimum to 0 for CT, and for MRI the intensity quantiles (2.5%, 97.5%) were used.

Smaller 2D images from modalities such as X-ray imaging were resized to an edge length of around 512 pixels. For larger images from histopathology, we extracted patches of similar size. Three-dimensional volumes were cut into slices to adapt them to the 2D format with task-specific patch sizes ranging from 224 to 512. Images did not need to be square for our training strategy.

In an ablation study, we evaluated how a uniform image size affected the performance of our model. We trained a version called UMedPT-fixed and downsized all image instances to 224 × 224 pixels. This contrasts with our standard UMedPT, where the gradient accumulation technique allows for dynamic image sizes to suit the requirements of each task. Besides this, the preparation of the 2D image inputs for UMedPT-fixed followed the same process as for the original UMedPT. The results are presented in Supplementary Section 1.

We used a caching component to load WSIs and 3D volumes. It is common practice to pre-extract image instances to disk or memory to minimize loading times, but this requires a lot of memory and loses the ability to perform augmentation on the original data. The proposed caching component eliminated the need for the pre-extraction of images, thereby enhancing data diversity. 3D augmentations were applied during every initial loading, and patch augmentations occurred when a patch was retrieved from the cache.

**Pretraining tasks.** We selected 15 publicly available datasets for pretraining[1,5,17,44–59], and extracted 17 tasks from them. Several criteria guided the selection of datasets:

- *Availability*. All datasets should be publicly available.
- *Clinical relevance*. Datasets should include imaging modalities that are widely used in radiology and pathology. For that reason, we included tasks from histopathology, X-ray and tomography.

- *Diversity of label types*. Where possible, we included tasks with a classification, segmentation and detection label type for each category.
- *Performance*. We prioritized datasets that demonstrated satisfactory performance when trained individually. We defined satisfactory performance as either aligning with the metrics reported by the dataset creators where available, or passing a plausibility check conducted by a medical expert.

We included four auxiliary datasets for the purpose of meta-learning. These datasets were not intended to directly improve a specific clinical application, but rather to enhance the model's general image understanding capabilities, drawing inspiration from the strong foundational capabilities of ImageNet pretrained models. Detailed statistics on the pretraining database are reported in Supplementary Table 2.

To further understand the importance of pretraining diversity, we conducted an ablation study focusing only on classification tasks. We trained an ablation UMedPT-clf using only the classification pretraining tasks. We evaluated UMedPT-clf on one representative task from classification (Pneumo-CXR), segmentation (PolypSeg-RGB) and object detection (NucleiDet-WSI), and compared it to the full model UMedPT.

## Data augmentation

For 3D tomographic images we applied standard 3D augmentations using the MONAI library[60] (3D rotations, scale and crop), followed by slicing and our set of standard 2D augmentations. We augmented the orientation of the volume if the maximum edge length was less than two times the shortest edge length as proposed in ref. 61.

For 2D images, we used standard augmentations using the Albumentations library[62] (CLAHE, Sharpen, Emboss, RandomBrightnessContrast, RandomGamma, Gaussnoise and HueSaturationValue, ShiftScaleRotate). For X-ray images, we added image inversion with a probability of 30%. For histological images, flipping and mirroring were applied to improve orientation invariance, and channel shuffling to improve the model's robustness to stain and color variations.

## Task types

We have included classification, segmentation and detection tasks. These have different loss functions with different magnitudes. We normalized the respective loss functions for each task type such that the observed value for random inputs for reinitialized weights was close to one. This strategy prevented the loss of one task from dominating the combined loss. In addition, this allows model selection based on the average loss.

**Classification.** In the classification task, we handled data where a single input was associated with a single classification label from a set of $C$ classes. We used the latent representation computed by the encoder and processed it through a fully connected layer to obtain classification scores.

For classification tasks, we employed categorical cross-entropy $\mathcal{L}_{\mathrm{CCE}}$ to compute the loss as implemented in PyTorch[36]. Typically, the loss magnitude would increase with a larger number of classes $C$. To prevent a bias towards tasks with more classes, we added a normalization term $\log C$ to compute the final loss as

$$\mathcal{L}_{\mathrm{multiclass}}(\hat{y}, y) = \mathcal{L}_{\mathrm{CCE}}(\hat{y}, y) \times \log C \qquad (1)$$

where $\hat{y} \in \mathbf{R}^C$ and $y \in \{0, 1, ., C\}$ are the true and predicted labels, respectively.

We did not utilize label smoothing or class probabilities in the classification task.

For multi-label classification tasks, we considered inputs that each had multiple binary classification targets $y$. In these cases, we chose

the binary cross-entropy loss $\mathcal{L}_{BCE}$ from ref. 36. To normalize the loss to 1, we multiplied by a constant factor $\log_2(\exp(1)) \approx 1.44269$:

$$\mathcal{L}_{multilabel}(\hat{y}, y) = \mathcal{L}_{BCE} \times 1.44269 \qquad (2)$$

**Semantic segmentation task.** The U-Net architecture consisted of an encoder and a decoder, with the decoder producing dense pixel-level embeddings as output. To generate the final output, skip connections were established between the encoder's feature maps and the decoder's upsampling layers. These skip connections were implemented by concatenating the corresponding feature maps with the decoder's upsampling outputs. For training, we adopted the U-Net decoder from ref. 63, which is an implementation of the original U-Net formulation proposed by ref. 37.

The semantic segmentation task aimed to assign a class label to each pixel within an image, with the targets consisting of classes ranging from 1 to $C$. For UMedPT, we configured its encoder to yield feature maps with strides of 4, 8, 16 and 32.

We employed an equally weighted combination of Dice loss and Focal loss for all segmentation tasks in UMedPT. This strategy was chosen because Dice loss has been shown to perform well for mildly skewed datasets[64], whereas Focal loss is particularly effective for highly imbalanced datasets[65]. Thus, this combination allowed us to address challenges associated with both balanced and imbalanced datasets, such as the presence of large background regions, without the need for hyperparameter tuning.

**Object detection task.** FCOS[39] is an anchor-free object detection method, which makes it an ideal candidate for our multi-task learning approach due to its similarity with our segmentation and classification methods, enabling efficient feature reuse.

Instead of traditional anchor-box-based detection, FCOS employs pixel-dense predictions and a box-postprocessing technique. The architecture incorporated a shared encoder and a detection-specific decoder that produced two branches: one for classification and another for bounding box regression. The classification branch managed multi-class classification and centerness per pixel, and the bounding box regression branch predicted rectangle parameters, specifically the distances from each pixel to the edges of the bounding box. Although centerness is not essential to the algorithm, it helps suppress low-quality bounding boxes. The final score for a box was computed by multiplying the predicted centerness with the corresponding classification score. To ensure homogeneity in the magnitudes of all multi-task losses and facilitate multi-task learning, we normalized the classification loss by dividing it by the number of classes $C$. This resulted in the combined loss function $\mathcal{L}_{object\ detection} = \mathcal{L}_{classification} \cdot \frac{1}{C} + \mathcal{L}_{regression} + \mathcal{L}_{centerness}$.

To reconstruct bounding boxes, the network produced a classification score per pixel, a centerness value and rectangle parameters. Centerness was shared among tasks, fostering efficient multi-task learning, and each detection task employed one convolutional layer for classification and one convolutional layer for regression. Similar to our segmentation task, the forward pass of the detection task generated one feature map for each downsampling step, typically resulting in five feature maps. These multi-level feature maps encapsulated spatial and semantic information at multiple resolutions, enhancing the method's efficacy in object detection and enabling the encoding of difficult ground-truth cases involving overlapping and variably sized boxes.

## Clinical validation

UMedPT was clinically validated using a diverse set of clinically relevant tasks. Our evaluation centered on two main aspects: the model's skill generalizability to new tasks and its proficiency in retrieving previously learned knowledge. These aspects were evaluated using two distinct benchmarks. We developed the downstream training schedule and tuned the hyperparameters using a simple synthetic dataset based on simple, 2D geometric shapes for all label types. We then performed the clinical evaluation exactly once without any further hyperparameter tuning. We evaluated the model after training for a fixed number of epochs. For this reason, we did not use a validation set in our experiments unless otherwise stated.

The in-domain benchmark tested the model's ability to recall and adapt learned skills to new tasks. The out-of-domain benchmark measured the model's ability to adapt its learned skills to tasks distinct from those in the pretraining database.

Two distinct usage settings were considered in our evaluation: frozen and fine-tuning. In the frozen scenario, we directly extracted image representations from the shared blocks, thereby showing the usefulness of the learned representations. Here, we used a single linear layer for all target tasks (including segmentation and object detection), also known as linear probing. Subsequently, the fine-tuning stage enabled the training of the shared blocks such that the parameters learned during the frozen training setting were used to initialize the task-specific head. The learning rates for the shared blocks in the transformer were set at $10^{-5}$, and the task-specific sections of the target tasks were trained at learning rates of $10^{-4}$. Both frozen and fine-tuning settings were trained for 100 epochs each.

To simulate data scarcity and evaluate sample efficiency, we took multiple samples from the original training set of target tasks at sizes of 1%, 5%, 10%, 25%, 50% and 100%. For pretraining, we always used the full pretraining datasets. We utilized five splits of the training data to account for random selection effects, and ensured that all data from smaller splits also appear in the corresponding larger splits. Each method was evaluated for exactly the same five random train–test splits. As a consequence, a paired $t$-test was applied (significance level $P < 0.05$), treating each baseline-UMedPT result on a single train–test split as a pair.

## In-domain benchmark

We formulated a benchmark that aimed at examining the recoverability of knowledge encapsulated in UMedPT. We selected knowledge already present in the pretraining database and examined their re-discovery potential by measuring the number of samples needed for re-identification, and compared the outcome with the ImageNet-1K baseline on novel images. The training images of one of the target tasks, CRC-WSI, were included in both pretraining and benchmarking.

**CRC tissue classification (CRC-WSI).** The CRC-WSI dataset[4] consisted of hematoxylin and eosin (HE) stained images with nine tissue classifications that are largely balanced. The training set comprised 100,000 images extracted from 86 WSIs, and the test data came from 25 different patients with CRC. The label type of the CRC-WSI task was multi-class classification. The authors of the dataset[4], including pathologists, manually delineated regions corresponding to pure tissue textures to generate labels and extract image patches. Images with artefacts such as tissue folds or without tumor components were excluded.

**Pneumonia in pediatric cohort (Pneumo-CXR).** The Pneumo-CXR dataset[18] consisted of pediatric chest X-rays, each labeled as either normal or pneumonia. Consistent with our approach for all datasets, we preserved the original label imbalance when downsizing the training datasets. The training set contained 1,349 normal cases and 3,883 pneumonia cases, and the test set contained 234 normal cases and 390 pneumonia cases. We treated Pneumo-CXR as a multi-class classification task with two classes.

The images were acquired as part of the routine clinical care of the patients. To generate a high-quality dataset for model training, the authors performed an initial screening on the dataset to exclude poor-quality or unreadable scans. Then, two expert physicians annotated the remaining images and classified them for the presence of pneumonia. As an additional quality measure, a third expert reviewed the test set to verify the accuracy of the diagnoses.

**Detection of nuclei in WSIs (NucleiDet-WSI).** In oncology, the distribution and appearance of nuclei are important for the diagnosis and study of cancer. To assess the ability of UMedPT to detect these nuclei, the NucleiDet WSI dataset[19] was used. This dataset consists of WSIs and covers ten cancer types. In the pretraining database, only prostate and colon cancer were included. We randomly divided the dataset into 950 images for training and 406 images for testing. The authors of the dataset created the annotations with the help of two pathologists and three graduate students, using an AI tool.

## Out-of-domain benchmark

We evaluated the transfer effectiveness of our method across a variety of clinically relevant tasks by establishing an out-of-domain generalization benchmark. In this benchmark, each task's domain had to be different from all domains of the pretraining tasks. We were able to increase the certainty of this claim by including only supervised pretraining tasks. Because the problem setting for each sample in a supervised task is known, this approach reduced the likelihood of pretraining knowledge overlapping with the benchmark.

**Tuberculosis diagnosis in CXR (Tuber-CXR).** The Tuber-CXR dataset[20] consisted of postero–anterior (PA) chest radiographs that we used for multi-class classification. We randomly divided the images into a training set (70% of the data) and a test set (30% of the data) before any evaluation. The training set contained 239 tuberculosis samples and 225 normal samples, and the test set contained 51 tuberculosis samples and 101 normal samples. This dataset was used as a multi-class classification task and considered out-of-domain because none of the pretraining datasets had tuberculosis labels. The images were collected from routine hospital practice over a period of one month. For a subset of 68 images, two radiologists provided consensus annotations to confirm the established ground truth of the dataset.

**CNS neoplasia diagnosis in MRI (CNS-MRI).** The dataset of the CNS-MRI multi-class classification task[21] consisted of 7,023 2D slices derived from MRI scans categorized into four classes: glioma, meningioma, no tumor and pituitary tumor. The slices originate from T1, T2 and FLAIR sequences and were selected by the authors of the dataset following manual annotation by three experienced radiologists. Before any evaluation, we randomly partitioned the images into a training set containing 70% of the data and a test set containing the remaining 30%.

**Breast cancer classification in WSI (BC-Bach-WSI).** The dataset of the BC-Bach-WSI multi-class classification task[66] was used for breast cancer classification in HE-stained whole histological slide images (WSIs). It consisted of four classes: normal, benign tumors, as well as in situ and invasive carcinomas. The dataset was derived from 30 WSIs and was divided into image patches by the authors of the dataset. Each resulting image was annotated by two expert pathologists. From these images we used 76 normal tissues, 79 benign tumors, 80 in situ carcinomas and 85 invasive cancers for training. For testing, we used 24 normal, 21 benign, 20 in situ and 15 invasive images.

**Breast cancer classification in microscopic images (BC-BHis-MIC).** The dataset of the BC-BHis-MIC multi-class classification task focused on the binary classification of microscopic images from HE-stained breast tumors into benign lesions as opposed to malignant tumors[67]. Benign lesions included adenosis, fibroadenoma, phylloides tumor and tubular adenoma. The malignant tumor class contained four subtypes of invasive carcinoma: ductal carcinoma (currently referred to as 'unspecific type'), lobular carcinoma, mucinous carcinoma and papillary carcinoma. The authors of the dataset had achieved a maximum AUC at ×200 magnification, which we also adopted for our analysis. The dataset contained 7,909 image patches from 82 patients, with 2,480 benign and 5,429 malignant images. To prepare for evaluation, we randomly divided the images into a training set (70% of the data) and a test set (30% of the data) before any analysis.

To determine the labels, initial identification of tumor regions within each slide was performed by an anatomopathologist. Final diagnoses were then made by experienced pathologists, with additional validation provided by other methods of analysis such as immunohistochemistry.

**Polyp segmentation in coloscopy (PolypSeg-RGB).** The PolypSeg-RGB task[68] focused on segmenting polyps from the background in coloscopy images. Because polyps can be precursors to CRC, coloscopy is an important diagnostic tool. The early detection and removal of polyps is essential to prevent the development of CRC. However, the effectiveness of coloscopy is often hampered by high miss rates. Studies have found that polyp miss rates during coloscopy can range from 14 to 30%, depending on the type of polyp[68]. We randomly divided the dataset into 700 training images and 300 test images.

## Comparison of benchmark results

We compared our results with the best previously reported study results for the target tasks and the mean performance for the MedMNIST database[16]. From the MedMNIST database, we only considered tasks that were available in the largest spatial size (224 × 224) and were not part of the UMedPT pretraining or clinical benchmark. In this context, we determined the percentage of data that UMedPT required to achieve performance comparable to the external reference result. For each target task, the evaluation criteria from the respective reference papers were used. In five cases, the dataset had to be split manually because the creators had not defined the test data. In these cases, manual splitting was performed only once to avoid bias.

## External multi-center validation of UMedPT

We submitted a UMedPT-based classifier for the external validation of the frozen image representations in the tumor classification task of the SemiCOL challenge[22]. This branch provided gigapixel histological HE-stained images of CRC, each labeled with a binary indicator of tumor presence.

Although our in-domain and out-of-domain benchmarks showed that reliable results can be obtained directly when basing a model on UMedPT without hyperparameter tuning and a fixed training schedule, in real-world settings, developers can be interested in applying UMedPT directly to gigapixel images. Because gigapixel images do not directly fit in GPU memory, we utilized UMedPT to extract features, subsequently constructing neural gigapixel image representations following the methodology introduced in ref. 69.

We next applied a straightforward image classification CNN with two convolutional layers ($1 \times 1 \to 3 \times 3$), global max pooling and a classification layer, amounting to 47,264 parameters. The training set consisted of 499 images (WSIs) from five different scanners and four different centers. The test data consisted of 2,300 images (WSIs) from eight different scanners and six centers, four of which did not contribute to the training data. Although we had no access to this dataset for pretraining UMedPT, it was still considered in-domain because of its similarity to the pretraining datasets CRC-WSI and Crag-WSI.

## Experiments with MedMNIST

In addition to the primary studies where UMedPT was applied to the full-scale data (224 × 224) within MedMNIST, we also performed separate trainings using the MedMNIST database[16].

MedMNIST is a collection of 18 medical datasets that were downscaled to enable quick experimentation with medical datasets. We used the same training schedule and hyperparameters as for the main study. Nevertheless, MedMNIST differs from the pretraining database of UMedPT in the following aspects:

- Spatial size: MedMNIST images are scaled down to a uniform 28 × 28 (2D) or 28 × 28 × 28 (3D) size, while UMedPT was trained using images at their original dimensions.
- Task type: MedMNIST exclusively includes classification tasks. UMedPT was trained with classification, segmentation and object detection tasks.
- Augmentations: we applied weak standard augmentations to the MedMNIST datasets, avoiding flips, whereas UMedPT used domain-specific augmentations tailored to each task type within its training set.
- Data loading: MedMNIST datasets have a fixed dataset length. For the UMedPT database, we developed domain-specific data loading strategies to be able to augment loading of the raw data.
- Meta-learning: MedMNIST does not include any meta-learning datasets, whereas UMedPT includes four large datasets for the purpose of general applicability, including non-medical data.

For our separate multi-task trainings with MedMNIST, of the 18 MedMNIST datasets, 12 were selected. We excluded three datasets (Organ{A,C,S}MNIST) because they were composed of 2D images from one of the included 3D datasets. A further three datasets (RetinaMNIST, TissueMNIST amd FractureMNIST3D) were excluded as the authors had reported low performance. In total, the subset included 370,980 imaging studies, or 1,087,104 image instances for training, validation and testing.

**Assessing 3D context preservation in pretraining.** Transforming inherently 3D medical imaging data into 2D slices for pretraining purposes can result in the loss of 3D contextual information. This dilemma presents a challenge when building a unified database for pretraining. While large pretraining databases are populated with 2D images, many tomographic medical imaging techniques capture complex anatomical structures in three dimensions.

To evaluate the ability to maintain 3D context in our pretraining approach, we trained MedMNIST multi-task networks that handled 2D and 3D tasks simultaneously. For the 3D tasks, we used a simple strategy based on a learned weighted average over the neural representations of the slices. This resulted in a single feature vector per 3D case, allowing use of the same linear classification head as in two dimensions, as described in Supplementary Section 5. Intuitively, this allowed the network to learn focusing on the most relevant slices of a 3D case before a prediction.

Our objective was to determine the effectiveness of this strategy compared to a network using 3D convolutional layers. We assessed this by directly comparing our learned weighted average-based classification results with the performance reported by the MedMNIST authors using a standard 3D CNN. For a useful comparison, we analyzed the results not only for a ResNet-50 CNN[70], as used by the MedMNIST authors, but also for the Swin Tiny Transformer[33], which is a smaller variant of the encoder architecture used in UMedPT.

**Reporting summary**

Further information on research design is available in the Nature Portfolio Reporting Summary linked to this Article.

## Data availability

Training and evaluation data were obtained from their original repositories and selected based on availability, clinical relevance and satisfactory performance metrics. A detailed list of data sources and access information is available in Supplementary Section 8. Source data are provided with this paper.

## Code availability

The maintained code for reproducing our results is available at https://github.com/FraunhoferMEVIS/MedicalMultitaskModeling. An archive

with code for all experiments, including the archived version of the training framework, is also available[51]. We provide a full list of software dependencies in the corresponding repositories. In particular, our work used PyTorch[36], Python (3.10.6), MONAI (1.1.0) and Albumentations (1.3.1). For data analysis and visualization, we used Pandas (1.5.3) and Scipy (1.10.1), matplotlib (3.8.2) and Seaborn (0.13.1). All dependencies are available from the public Python Package Index.

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

## Acknowledgements

This research was funded by the German ministry of education and research (BMBF) through the project SynDICAD (01IS21067C; R.S., T.N., A.L., D.M., H.H., F.F., J.L.) and the German Research Foundation (DFG), CRC 1382 (403224013; F.K.). Our work uses datasets that are licensed under CC BY NC-SA 4.0 (ref. 44,71,72), CC BY 4.0 (ref. 4,16,73) and CC BY SA 4.0 (ref. 23). We thank the authors of the datasets for their contributions.

## Author contributions

R.S. and T.N. built the pretraining and benchmark databases and implemented software. R.S. conducted the experiments, analyzed the results, and wrote the paper, with feedback from all authors. T.N. generated the plots and diagrams, and analyzed the results. A.L., D.M., H.H., F.F., V.S. revised the paper. F.F. was the pathology advisor. J.L. coordinated the study. F.K. coordinated the study and was radiology advisor.

## Funding

V.

## Competing interests

The applicant 'Fraunhofer-Gesellschaft zur Förderung der angewandten Forschung eingetragener Verein' has a patent pending related to the training algorithm and neural architecture components presented in this Article (patent application no. EP23209015.9; names of inventors, R.S., T.N., H.H., J.L., F.K.).

## Additional information

**Extended data** is available for this paper at https://doi.org/10.1038/s43588-024-00662-z.

**Correspondence and requests for materials** should be addressed to Johannes Lotz or Fabian Kiessling.

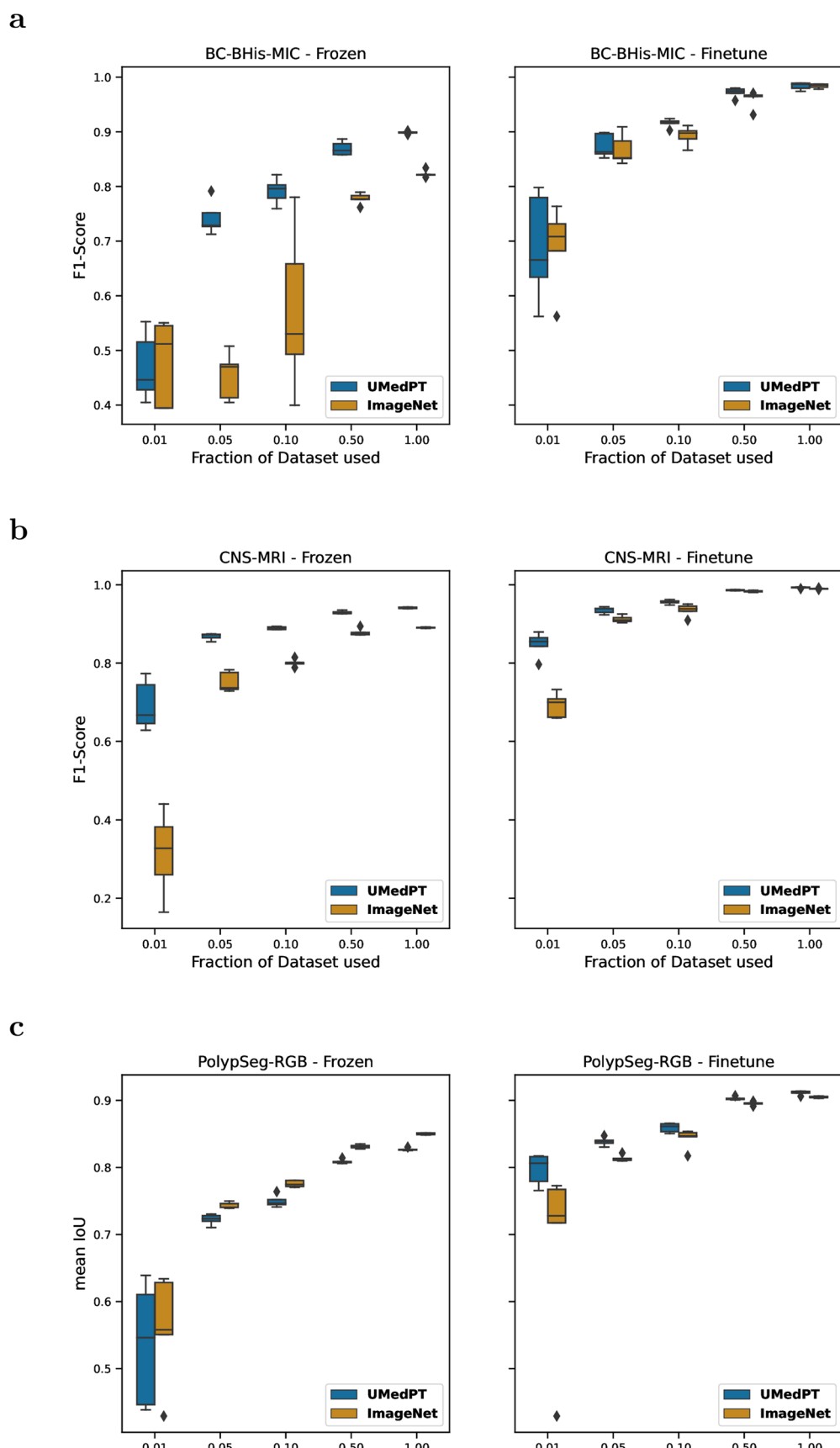

**Extended Data Fig. 1 | Results of remaining out-of-distribution tasks.**
**a** BC-BHis-MIC was used to classify breast tumors as benign or malignant in microscopic images. **b** CNS-MRI evaluated UMedPT for classification of CNS neoplasms from MRI scans. **c** PolypSeg-RGB was used to segment polyps in coloscopy images. In each setting, 5 independent trainings were derived for each training subset and method. The middle line of the boxes represents the median, the boundaries are the Q1 and Q3 quartiles, the whiskers extend to 1.5 IQR, and outliers beyond are shown as single points.

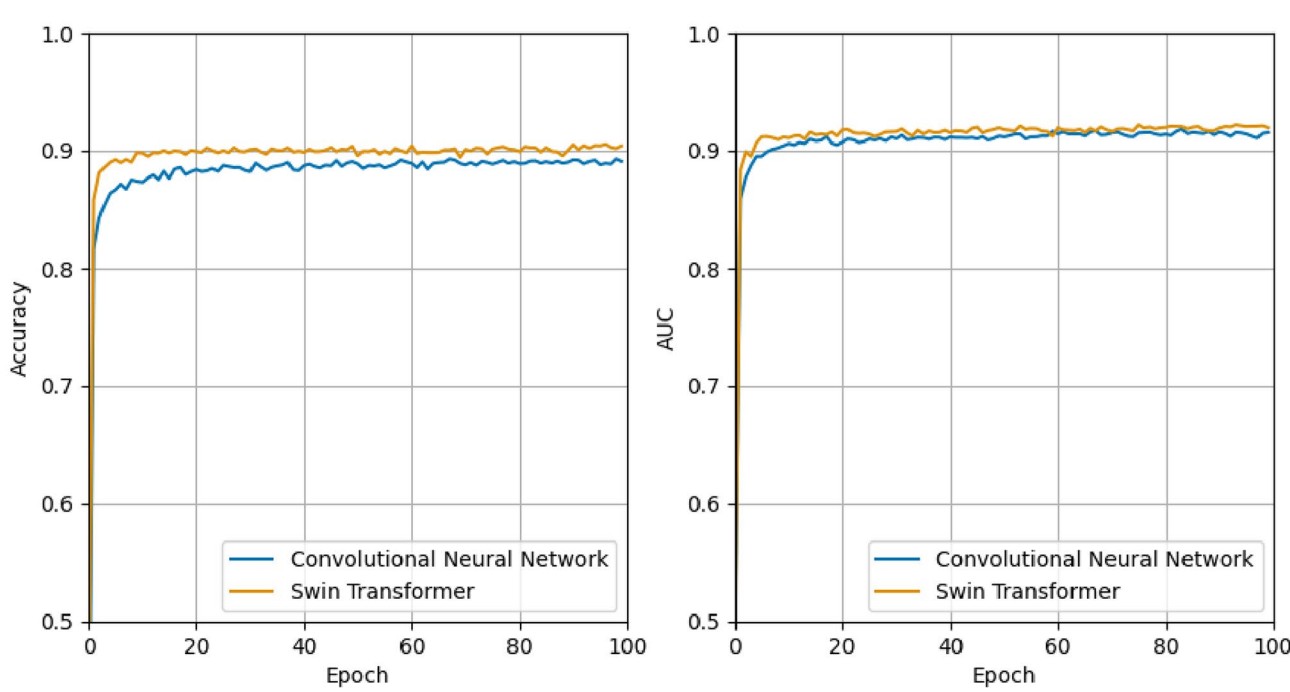

**a** Architecture

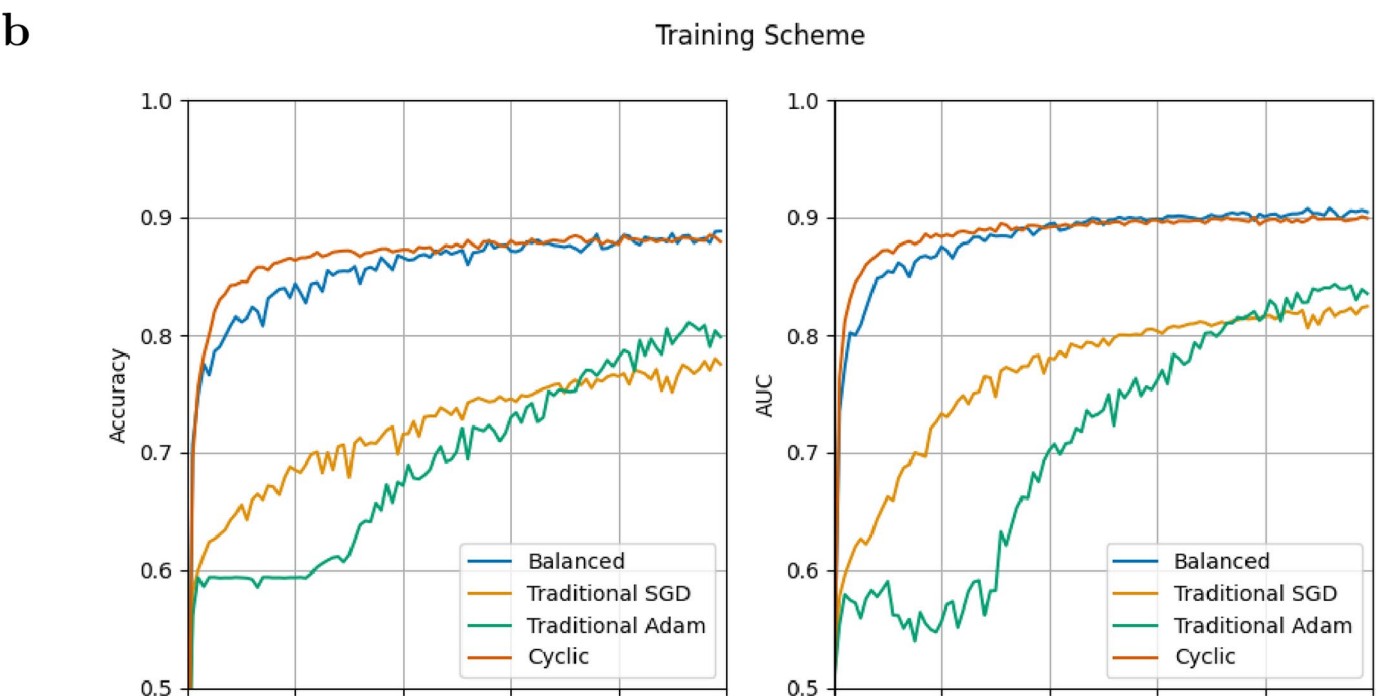

**b** Training Scheme

**Extended Data Fig. 2 | MedMNIST training convergence. a** Architecture comparison between ResNet-50[70] and Swin Transformer in the "tiny" variant[33], evaluated on combined 2D and 3D multi-task trainings. **b** Comparison of training schemes for the Swin Transformer tiny architecture. Traditional SGD used SGD optimizer without momentum and without gradient accumulation. Traditional Adam used the same setting but with the Adam optimizer. Balanced added 12 gradient accumulation steps to the traditional Adam setting. Cyclic systematically sampled each task exactly once per update step, identical to the method used to train UMedPT. The average standard deviation across five independent experiments of the last 10 epochs of validation accuracy was 1.81 ± 1.79% for balanced sampling and 1.17 ± 1.09% for cyclic sampling (Mean ± SD).

# Reporting Summary

## Statistics

For all statistical analyses, confirm that the following items are present in the figure legend, table legend, main text, or Methods section.

| n/a | Confirmed | |
|---|---|---|
| ☐ | ☒ | The exact sample size (*n*) for each experimental group/condition, given as a discrete number and unit of measurement |
| ☐ | ☒ | A statement on whether measurements were taken from distinct samples or whether the same sample was measured repeatedly |
| ☐ | ☒ | The statistical test(s) used AND whether they are one- or two-sided *Only common tests should be described solely by name; describe more complex techniques in the Methods section.* |
| ☐ | ☒ | A description of all covariates tested |
| ☐ | ☒ | A description of any assumptions or corrections, such as tests of normality and adjustment for multiple comparisons |
| ☐ | ☒ | A full description of the statistical parameters including central tendency (e.g. means) or other basic estimates (e.g. regression coefficient) AND variation (e.g. standard deviation) or associated estimates of uncertainty (e.g. confidence intervals) |
| ☐ | ☒ | For null hypothesis testing, the test statistic (e.g. *F*, *t*, *r*) with confidence intervals, effect sizes, degrees of freedom and *P* value noted *Give P values as exact values whenever suitable.* |
| ☒ | ☐ | For Bayesian analysis, information on the choice of priors and Markov chain Monte Carlo settings |
| ☐ | ☒ | For hierarchical and complex designs, identification of the appropriate level for tests and full reporting of outcomes |
| ☒ | ☐ | Estimates of effect sizes (e.g. Cohen's *d*, Pearson's *r*), indicating how they were calculated |

*Our web collection on statistics for biologists contains articles on many of the points above.*

## Software and code

Policy information about availability of computer code

| Data collection | We provide a full list of software used and their exact versions within the code archive. In particular, we used Python (3.10.6) & PyTorch for training deep neural networks. MONAI (1.1.0) and Albumentations (1.3.0) for data preprocessing. We have included an exact list of a installed dependencies in the code archives for each version available from Zenodo (https://zenodo.org/records/11383543): code/medmnist_version/requirements.txt and code/original_version/requirements.txt. The latest version of our training framework is available at https://github.com/FraunhoferMEVIS/MedicalMultitaskModeling, and the latest version of UMedPT is available at https://github.com/FraunhoferMEVIS/UMedPT. |
|---|---|
| Data analysis | - Pandas (1.5.3)<br>- Scipy (1.10.1)<br>- Matplotlib (3.8.2)<br>- Seaborn (0.13.1)<br>All dependencies are available from the public Python Package Index. |

For manuscripts utilizing custom algorithms or software that are central to the research but not yet described in published literature, software must be made available to editors and reviewers. We strongly encourage code deposition in a community repository (e.g. GitHub). See the Nature Portfolio guidelines for submitting code & software for further information.

# Data

Policy information about availability of data

All manuscripts must include a data availability statement. This statement should provide the following information, where applicable:

- Accession codes, unique identifiers, or web links for publicly available datasets
- A description of any restrictions on data availability
- For clinical datasets or third party data, please ensure that the statement adheres to our policy

Source Data for Figures 1, 3, 4 and 5 are available with this manuscript. The training and evaluation data were obtained from their original repositories and selected based on availability, clinical relevance, and satisfactory performance metrics.All data sources are listed below:

- Amos22 (organ segmentation in CT): https://amos22.grand-challenge.org/
- Conic-WSI (cell detection): https://conic-challenge.grand-challenge.org/
- PICAL-MRI (prostate cancer classification) https://pi-cai.grand-challenge.org/:
- Panda-WSI (prostate tissue semantic segmentation \& classification): https://www.kaggle.com/c/prostate-cancer-grade-assessment
- VinBigData-CXR hest-xray-abnormalities-detection (Thorax pathology pathology detection): https://www.kaggle.com/competitions/vinbigdata-chest-xray-abnormalities-detection
- Crag-WSI (Colorectal tissue semantic segmentation): https://github.com/XiaoyuZHK/CRAG-Dataset_Aug_ToCOCO
- Brats2020-MRI (brain semantic segmentation): https://www.kaggle.com/datasets/awsaf49/brats20-dataset-training-validation
- Avaniti-WSI (prostate multi-label classification): https://doi.org/10.7910/DVN/OCYCMP
- Cyto-WSI (bone marrow single cell multi-class classification): https://wiki.cancerimagingarchive.net/pages/viewpage.action?pageId=101941770
- Chexpert-CXR (Thorax pathology multi-label classification): https://stanfordaimi.azurewebsites.net/datasets/8cbd9ed4-2eb9-4565-affc-111cf4f7ebe2 \& https://github.com/rajpurkarlab/cheXpert-test-set-labels
- SIIM-CXR (pneumothorax semantic segmentation): https://www.kaggle.com/competitions/siim-acr-pneumothorax-segmentation/data
- ImageNet (real world image classification): https://www.image-net.org/download.php
- RadImageNet (radiology multi-class classification): request access at https://www.radimagenet.com/copy-of-home-1
- COCO (real world semantic segmentation \& object detection): https://cocodataset.org/#download
- CRC-WSI (colorectal cancer tissue classification): https://zenodo.org/record/1214456
- Pneumo-CXR (pneumonia in pediatric cohort): https://data.mendeley.com/datasets/rscbjbr9sj/3
- Tuber-CXR (tuberculosis diagnosis in CXR): https://www.kaggle.com/datasets/raddar/tuberculosis-chest-xrays-shenzhen
- CNS-MRI (CNS neoplasia diagnosis in MRI): https://www.kaggle.com/datasets/masoudnickparvar/brain-tumor-mri-dataset
- BC-Bach-WSI (breast cancer classification in WSI): https://iciar2018-challenge.grand-challenge.org/
- BC-BHis-MIC (breast cancer classification in microscopic images): https://web.inf.ufpr.br/vri/databases/breast-cancer-histopathological-database-breakhis/
- PolypSeg-RGB (polyp segmentation in coloscopy): https://datasets.simula.no/kvasir-seg/
- NucleiDet-WSI (detection of nuclei in whole slide images): https://www.nature.com/articles/s41597-020-0528-1
- Medical Segmentation Decathlon (3D segmentation experiment): https://decathlon-10.grand-challenge.org/
- MedMNIST database (Application of UMedPT to MedMNIST and separate experiments with MedMNIST): https://zenodo.org/records/5208230

All data are either directly publicly accessible or can be obtained by requesting access at the specified URL from the authors of the dataset.

# Research involving human participants, their data, or biological material

Policy information about studies with human participants or human data. See also policy information about sex, gender (identity/presentation), and sexual orientation and race, ethnicity and racism.

| | |
|---|---|
| Reporting on sex and gender | Not applicable. For reproducibility, we use all cohorts as provided by their publishers. |
| Reporting on race, ethnicity, or other socially relevant groupings | Not applicable. For reproducibility, we use all cohorts as provided by their publishers. |
| Population characteristics | Not applicable. |
| Recruitment | Not applicable. |
| Ethics oversight | Not applicable. |

Note that full information on the approval of the study protocol must also be provided in the manuscript.

# Field-specific reporting

Please select the one below that is the best fit for your research. If you are not sure, read the appropriate sections before making your selection.

☒ Life sciences          ☐ Behavioural & social sciences          ☐ Ecological, evolutionary & environmental sciences

For a reference copy of the document with all sections, see nature.com/documents/nr-reporting-summary-flat.pdf

# Life sciences study design

All studies must disclose on these points even when the disclosure is negative.

| | |
|---|---|
| Sample size | We used 23 independent, external data sets that are publicly available. In consequence, the sample size for all data sources was predetermined. However, for simulating data scarcity we chose random subsets of the full datasets at 1%, 5%, 10%, 50% and 100% by intuition. For experiments with MedMNIST, we chose 1%, 10%, and 100% to simulate data scarcity due to the large number of experiments. Given the available computational resources, we chose the number of subsamples and the number of replicates per experimental setting (n=5 each) to be as large as possible. |
| Data exclusions | No data exclusions. |
| Replication | We make all code publicly available and stated the origin for all data sources. Additionally, we tuned hyperparameters using toy data and ran the real evaluation exactly once without intervention. |
| Randomization | For each task, we had to decide on which data to use for training and validation for training the networks, and we had to decide how to allocate the training and test data for evaluation. Wherever the original data publishers already made a decision we used their strategy. Otherwise, we randomly split the data using 2/3 of the data for training and 1/3 for testing. We made sure the same patient can only appear either in training or in the testing data for all evaluation tasks. |
| Blinding | The evaluation in our study did not include any subjective components. All data were based on objective findings, leaving no room for personal interpretation or bias. Consequently, there was no risk of bias that could have been mitigated by blinding. |

# Reporting for specific materials, systems and methods

We require information from authors about some types of materials, experimental systems and methods used in many studies. Here, indicate whether each material, system or method listed is relevant to your study. If you are not sure if a list item applies to your research, read the appropriate section before selecting a response.

## Materials & experimental systems

| n/a | Involved in the study |
|---|---|
| ☒ | ☐ Antibodies |
| ☒ | ☐ Eukaryotic cell lines |
| ☒ | ☐ Palaeontology and archaeology |
| ☒ | ☐ Animals and other organisms |
| ☒ | ☐ Clinical data |
| ☒ | ☐ Dual use research of concern |
| ☒ | ☐ Plants |

## Methods

| n/a | Involved in the study |
|---|---|
| ☒ | ☐ ChIP-seq |
| ☒ | ☐ Flow cytometry |
| ☒ | ☐ MRI-based neuroimaging |

## Plants

| | |
|---|---|
| Seed stocks | Not applicable |
| Novel plant genotypes | Not applicable |
| Authentication | Not applicable |

