## [Peer Review File · Nature Computational Science]

Peer Review Information

Journal: Nature Computational Science

Manuscript Title: Overcoming Data Scarcity in Biomedical Imaging with a Foundational Multi-Task Model

Corresponding author name(s): Professor Fabian Kiessling

Editorial Notes:

Reviewer Comments & Decisions:

Decision Letter, initial version:

Date: 19th December 23 13:23:06

Last Sent: 19th December 23 13:23:06

Triggered By: Ananya Rastogi

From: ananya.rastogi@nature.com

To: fkiessling@ukaachen.de

BCC: ananya.rastogi@nature.com

Subject: Decision on Nature Computational Science manuscript NATCOMPUTSCI-23-1256-T

Message: ** Please ensure you delete the link to your author homepage in this e-mail if you wish to forward it to your co-authors. **

Dear Professor Kiessling,

Your manuscript "Overcoming Data Scarcity in Biomedical Imaging with a Foundational Multi-Task Model" has now been seen by 2 referees, whose comments are appended below. You will see that while they find your work of interest, they have raised points that need to be addressed before we can make a decision on publication.

The referees' reports seem to be quite clear. Naturally, we will need you to address *all* of the points raised.

While we ask you to address all of the points raised, the following points need to be substantially worked on:

- Please discuss how the datasets for pretraining are selected.

- As both reviewers have mentioned, an increase in data volume can enhance the model's performance. Are there any reasons analyzed here for the weakening of performance?
- In the study, when standardizing 3D images into 2D images, a significant amount of the original three-dimensional information is lost. It should be discussed if the performance of this foundational model been directly compared to the training performance of 3D networks for 3D images.
- Please include an ablation study involving the proposed algorithm gradient accumulation and traditional training schemes.
- Please include statistical information about the pretraining datasets, such as the total amount of data in pretraining, the amount of data per dataset.
- As indicated by Reviewer #1, the README file should be provided.

In addition to these points, it would also be beneficial to address the following concerns:

- Please avoid the use of the phrase "foundational model". Instead, please use "LLM".

Please use the following link to submit your revised manuscript and a point-by-point response to the referees' comments (which should be in a separate document to any cover letter):

[REDACTED]

** This url links to your confidential homepage and associated information about manuscripts you may have submitted or be reviewing for us. If you wish to forward this e-mail to co-authors, please delete this link to your homepage first. **

To aid in the review process, we would appreciate it if you could also provide a copy of your manuscript files that indicates your revisions by making use of Track Changes or similar mark-up tools. Please also ensure that all correspondence is marked with your Nature Computational Science reference number in the subject line.

In addition, please make sure to upload a Word Document or LaTeX version of your text, to assist us in the editorial stage.

To improve transparency in authorship, we request that all authors identified as 'corresponding author' on published papers create and link their Open Researcher and Contributor Identifier (ORCID) with their account on the Manuscript Tracking System (MTS), prior to acceptance. ORCID helps the scientific community achieve unambiguous attribution of all scholarly contributions. You can create and link your ORCID from the home page of the MTS by clicking on 'Modify my Springer Nature account'. For more information please visit please visit www.springernature.com/orcid.

We hope to receive your revised paper within three weeks. If you cannot send it within this time, please let us know.

Best regards,

Ananya Rastogi, PhD
Senior Editor
Nature Computational Science

Reviewers comments:

Reviewer #1 (Remarks to the Author):

This study developed a foundational pretrained model using a multi-task learning strategy across various biomedical modalities and label types. This model demonstrated effective knowledge transfer capabilities to reduce the amount of data and time for unseen tasks. The foundation model is valuable for biomedical imaging, especially in data-scarce scenarios. While the research holds promise, there are several concerns and areas of improvement.

1. Technical concerns:

- a. How the datasets for pretraining are selected? Why them?
- b. Is ImageNet being used in reference to ImageNet 1K? Any performance comparisons with ImageNet 21K pre-training? As I understand it, pre-training with 21K classes exhibits notably stronger performance.
- c. In lines 160-162, "Surprisingly, for UMedPT, increasing the training data beyond 1% did not enhance the model's performance and sometimes tended to degrade it." In theory, an increase in data volume can enhance the model's performance. Are there any reasons analyzed here for the weakening of performance?
- d. In line 460: "To accommodate these different data types, the encoder of UMedPT used a standardized 2D image input format." To my knowledge, 3D networks perform better with 3D images compared to 2D networks. In the study, when standardizing 3D images into 2D images, a significant amount of the original three-dimensional information is lost. Has the performance of this foundational model been directly compared to the training performance of 3D networks for 3D images?
- e. In lines 388-389, "No information on the dataset's length was needed beforehand, which allowed each epoch to have a different length depending on data augmentation." Confused about the varying lengths per epoch. Are different batch sizes used for each task?
- f. In lines 586-590, "For this reason, we did not use a validation set in our experiments." How to determine the endpoint of training without validation set for downstream tasks?
- g. The study claim a contribution that one model covers various data modalities and tasks. But the comparison over other methods is limited, including different pretraining methods, backbones, learning strategies, etc. If the pretrained model is universal, I suggest to report experiments on MedMNIST (Scientific Data, 2023), where 12 2D data and 6 3D data are standardized to be compared with other methods.

2. Ablation studies:

- a. The paper lacks ablation study involving the proposed algorithm gradient

accumulation and traditional training schemes. There can be many variants, including loss weights and dataset balanced sampling.

b. Only one backbone is used in this study. Convnet-based models are also encouraged.

3. On writing:

a. I don't understand the difference between UMedPT, UMedPT-fixed and UMedPT-affine, the description in the paper is hard to follow.

b. The article lacks statistical information about the pretraining datasets, such as the total amount of data in pretraining, the amount of data per dataset.

4. Minor suggestions :

a. In Extended Data Table 1, "PPneumo-CXR" should be "Pneumo-CXR"; If UMedPT-A and UMedPT-affine refer to the same thing, their naming needs to be unified; Is the result "58.21±9.50%" representing the mean and standard deviation? It's not specified in the table header.

b. In line 239 and 304, SemiCOL challenge dataset should be classification task, not tumor detection subtask.

Reviewer #1 (Remarks on code availability):

Lack of clarity due to lack of README.

The absence of instructions on project structure, environment setup, required packages, and reproduction steps is a major obstacle. Yet, the presence of many test files is a positive sign, showing extensive testing of the code blocks.

Reviewer #2 (Remarks to the Author):

Overcoming Data Scarcity in Biomedical Imaging with a Foundational Multi-Task Model

Training models with scarce data is a major problem in the field of image analysis. In addition, combining multimodal datasets (eg imaging, pathology) is another major challenge for precision medicine and it is not usually the case that image analysis solutions at the tissue level can successfully be applied to in vivo imaging. The authors have used a multi-task foundational model to overcome these issues by using simultaneous training of a single model that generalizes across multiple tasks. This could therefore be applied to the many small datasets that are currently available given the absence of larger datasets. The approach used here included three supervised label types: object detection, segmentation, and classification. The authors developed a fully supervised foundational model for biomedical imaging which they termed UMedPT, using 17 tasks based on 15 datasets and their original annotations.

The authors divided the assessment in two ways: in-domain benchmark to assess the applicability of UMedPT to problems closely related to its training database and the out-of-domain benchmark to evaluate its performance in unfamiliar domains. The

UMedPT outperformed the pretrained ImageNet network in both in and out-of-domain tasks. Their results were impressive with UMedPT matching the best performance of the ImageNet baseline over all configurations using only 1% of the original training data. Increasing the training data beyond 1% did not enhance performance and sometimes tended to degrade it – this is counterintuitive, and the authors should provide some explanation for this result.

The examples given include classification of colorectal and breast cancer on pathological slides, diagnosing pneumonia/TB on chest X-ray, and brain tumours/organ segmentation on MRI. It is important to understand how all of these were validated: there is mention of two expert pathologists annotating the breast cancer slides but it is not clear how the diagnoses were confirmed on some of the other datasets (eg TB).

The approach for the colorectal cancer slide was applied to data acquired from multiple separate sites showing its applicability on data not from the primary training set. Was this multi-site approach also performed on the X-ray and MRI data? The authors state that these foundational models should be robust to multi-center variances, thereby improving generalizability, but appear to provide the evidence for this from histological analysis only.

In summary, the authors have presented some very interesting work on how a novel multitask training strategy can be used for unseen target tasks and scarce data. It would be interesting to understand the authors views on how far this approach could be extended to other out-of-domain tasks: what are the limits of how this approach could be applied to new data and what are the factors affecting these limits?

Author Rebuttal to Initial comments

Our direct replies are presented in this document, and the actual changes in the manuscript have been highlighted in green.

Editor

Editor: Please discuss how the datasets for pretraining are selected.

Reply: We are happy to add this information and added details on the process of constructing the pretraining database (page 18, lines 563-581):

“Pretraining Tasks

We selected 15 publicly available datasets for pretraining and extracted 17 tasks from them. Several criteria guided the selection of datasets:

- Availability: All datasets should be publicly available.
- Clinical Relevance: Datasets should include imaging modalities that are widely used in radiology and pathology. For that reason, we included tasks from histopathology, X-ray, and tomography.
- Diversity of Label Types: Where possible, we included tasks with a classification, segmentation, and detection label type for each category.
- Performance: We prioritized datasets that demonstrated satisfactory performance when trained individually. We defined satisfactory performance as either aligning with the metrics reported by the dataset creators where available, or passing a plausibility check conducted by a medical expert.

We included four auxiliary datasets for the purpose of meta-learning. These datasets were not intended to directly improve a specific clinical application, but rather to enhance the model’s general image understanding capabilities, drawing inspiration from the strong foundational capabilities of ImageNet pre-trained models. Detailed statistics on the pretraining database are reported in Table 4.”

Editor: As both reviewers have mentioned, an increase in data volume can enhance the model’s performance. Are there any reasons analyzed here for the weakening of performance?

Reply: We appreciate the opportunity to discuss the counterintuitive phenomenon observed in our study. We agree that, intuitively, collecting more data for an unseen clinical target task should improve performance. The phenomenon of a smaller dataset beating a larger dataset was observed primarily for fine-tuning, suggesting that the current method of fine-tuning may in some cases lead to “catastrophic forgetting” of the well generalizing multi-task parameters. Catastrophic forgetting is a phenomenon in which AI systems lose information from previous tasks as they learn new ones.

In fact, our training schedule used a fixed number of epochs. Consequently, as the size of the fine-tuning dataset increased, there were naturally more update steps. This gives the training process more time to overwrite or ‘forget’ the useful features learned during the pretraining phase, especially as all parameters are allowed to change freely during fine-tuning.

To test the hypothesis, we performed an additional experiment (page 27, lines 879-899):

“In our clinical benchmark, we observed that increasing the amount of data for a target task could paradoxically lead to a decrease in performance during the fine-tuning phase. To explore the potential role of catastrophic forgetting, where neural networks lose previously learned information as they acquire new knowledge, we designed an experiment focused on the fine-tuning phase of our model.

We pretrained a multi-task deep learning network on the MedMNIST database. From this, we selected four tasks (SynapseMNIST3D, VesselMNIST3D, BreastMNIST, PneumoniaMNIST) that had shown improved performance with multi-task learning compared to single-task learning. Our aim was to analyse how further training of these tasks, individually with their full datasets, would affect their test accuracy.

For the experiment, we first measured the test accuracy for the four selected tasks after multi-task learning with 12 MedMNIST tasks for 100 epochs. We then continued to train the selected tasks individually using 100% of their respective training data for 100 epochs, after which we recorded the test accuracy again.

If, after this further individual fine-tuning, test accuracy fell to levels comparable to those achieved by single-task training, this would indicate that catastrophic forgetting had occurred.”

Our experiments revealed that it depended on the dataset whether catastrophic forgetting occurred during fine-tuning or not. We present the findings in page 11, lines 282-296:

“[...] The results varied between datasets, suggesting that whether catastrophic forgetting affects fine-tuning is not consistent across datasets and may be task-dependent:

- SynapseMNIST3D: $83.81 \pm 0.31\%$ -> $82.90 \pm 0.66\%$ (decrease)
- VesselMNIST3D: $93.66 \pm 0.31\%$ -> $93.77 \pm 0.87\%$ (no decrease)
- BreastMNIST: $86.92 \pm 1.04\%$ -> $85.90 \pm 0.91\%$ (decrease)
- PneumoniaMNIST: $91.54 \pm 0.48\%$ -> $91.70 \pm 0.49\%$ (no decrease)”

Unfortunately, at the moment we do not have a solution to predict the appearance of catastrophic forgetting.

The phenomenon of catastrophic forgetting could be addressed by tools developed for language models. In this context, parameter-efficient fine-tuning techniques could be BitFit (where only the bias-terms are fine-tuned) and LoRA (where a small number of additional parameters are fine-tuned). We discuss these potential future improvements on our approach in page 12, lines 353-363:

“Investigating catastrophic forgetting in target tasks

In some cases we observed that the performance of UMedPT decreased as the size of the training dataset increased. This could be due to catastrophic forgetting of the well generalizing features learned during pretraining. However, this effect was not consistent, suggesting that it may be dataset dependent.

Recently, more sophisticated fine-tuning strategies have been proposed for foundational models in natural language processing, such as BitFit [33], where only the bias-terms are fine-tuned, or LoRA [34], where a small number of additional parameters are fine-tuned. A training configuration targeted specifically to foundational vision models might combine the strengths of frozen and fine-tuned training configurations, and could improve on the problem of catastrophic forgetting [25].”

Editor: *In the study, when standardizing 3D images into 2D image data, a significant amount of the original three-dimensional information is lost. It should be discussed if the performance of this foundational model been directly compared to the training performance of 3D networks for 3D images.*

Reply: We agree that the conversion of 3D images into 2D slices during pretraining results in a loss of spatial context, and quantifying this loss is important for the interpretation of our results. To quantify the performance differences between 2D pre-training for 3D tasks and 3D convolutional neural networks, we performed additional analyses, which are described in the Methods section of the additional MedMNIST experiments on page 26, lines 842-862:

“Assessing 3D context preservation in pretraining

Transforming inherently 3D medical imaging data into 2D slices for pretraining purposes can result in the loss of three-dimensional contextual information. This dilemma presents a challenge when building a unified database for pretraining: while large pretraining databases are populated with 2D images, many tomographic medical imaging techniques capture complex anatomical structures in three dimensions.

To evaluate the ability to maintain three-dimensional context in our pre-training approach, we trained MedMNIST multi-task networks that handled 2D and 3D tasks simultaneously. For the 3D tasks, we used a simple strategy based on a learned weighted average across slices with the classification task described in section 5. Intuitively, this allows the network to learn focusing on the most relevant slices of a 3D case.

Our objective was to determine the effectiveness of this strategy compared to a network using three-dimensional convolutional layers. We assessed this by directly comparing our learned weighted average-based classification results with the performance reported by the MedMNIST authors using a standard 3D CNN. For a useful comparison, we analysed the results not only for a Resnet-50 CNN [53], as used by the MedMNIST authors, but also for the Swin Tiny Transformer [38], which is a smaller variant of the encoder architecture used in UMedPT.”

We present the results in page 10, lines 253-264:

“Assessing 3D context preservation in pretraining

To assess the impact of transforming inherently 3D medical imaging data into 2D slices for pretraining, we evaluated single-task learning with 2D data, single-task learning with original 3D data [24] and multi-task learning with 2D data. When only taking the 3D tasks, the average accuracies were:

- 83.22 ± 1.61% for single-task learning pretrained using Imagenet;
- 83.76% for the single-task MedMNIST 3D CNN;
- 86.46 ± 1.13% for multi-task learning with weighted averaging.

The results showed that single-task 3D CNNs performed better than single-task 2D CNNs pretrained using Imagenet. However, multi-task learning 2D networks outperformed the single-task 3D CNNs. We present the details in Extended Data Table 5.”

Here we show an excerpt from Extended Data Table 5 with only the 3D results for the CNNs:

Extended Data Table 5 MedMNIST test performance. Multi-task learning (MTL) networks were trained using all tasks, including both 2D and 3D tasks. Single-task (ST) were trained independently. Metrics are reported as mean \pm standard deviation in percentage. The reference accuracy (ACC Ref) and area under the curve (AUC Ref) are taken from the publication associated with the MedMNIST database [52].

CNN	ACC MTL	AUC MTL	ACC ST	AUC ST	ACC Ref	AUC Ref
AdrenalMNIST3D	83.89 \pm 0.88%	86.85 \pm 1.95%	80.87 \pm 3.40%	76.96 \pm 17.78%	74.50%	82.80%
NoduleMNIST3D	85.94 \pm 0.97%	85.95 \pm 1.45%	86.13 \pm 0.54%	90.87 \pm 0.71%	84.70%	87.50%
OrganMNIST3D	87.11 \pm 1.25%	99.00 \pm 0.25%	86.82 \pm 2.74%	98.69 \pm 0.25%	88.30%	99.40%
SynapseMNIST3D	81.08 \pm 1.50%	81.81 \pm 1.32%	73.52 \pm 0.55%	65.62 \pm 6.14%	79.50%	85.10%
VesselMNIST3D	94.29 \pm 0.45%	95.64 \pm 0.75%	88.74 \pm 0.00%	69.60 \pm 2.80%	91.80%	90.70%

Editor: Please include an ablation study involving the proposed algorithm gradient accumulation and traditional training schemes

Reply: Thank you for your valuable suggestion to include an ablation study comparing our proposed gradient accumulation-based training strategy with traditional training schemes. We understand the importance of demonstrating whether our method is superior to or differs from traditional training schemes, and we agree that this information would be highly interesting for the readers of the article. Therefore, we extended the UMedPT ablation study that was part of the initial submission. In this context, we have performed several additional ablation studies related to the training strategy and included the following paragraph in the Methods section (page. 27, lines. 868-878):

“Investigating training schemes

All datasets in MedMNIST have a fixed number of cases. This distinction enabled us to conduct ablation studies comparing infinite task sampling with balanced sampling based on dataset size. Besides this, we used the same training schedule and hyperparameters as in the main study, and accumulated the gradients of as many steps as there were tasks. In addition, for comparison with traditional training schemes, we used the same setting without gradient accumulation and also with the SGD optimizer instead of Adam.

To quantify the effect of task scheduling, we reported the stabilities of the training processes by the standard deviations of the validation performances over the last 10 training epochs.”

We report the new results in page 10, lines 272-281:

“The exploration of training schemes showed that balanced (by dataset size) and cyclic sampling (as in UMedPT) exhibited similar behaviour in terms of convergence. However, balanced sampling occasionally showed reduced stability; it yielded a standard deviation of 1.81 \pm 1.79% in validation accuracy over the previous ten epochs, across five different experiments. In comparison, cyclic sampling showed a more stable training process, achieving a comparatively lower standard deviation of 1.17 \pm 1.09%. When gradient accumulation was excluded, the resulting performance deteriorated, accompanied by longer convergence times. These results are shown in Extended Data Figure 2b.”

Extended Data Fig. 2 MedMNIST training convergence. *a* Architecture comparison [...] **b** Comparison of training schemes for the Swin Transformer tiny architecture. Traditional SGD used SGD optimizer without momentum and without gradient accumulation. Traditional Adam used the same setting but with the Adam optimizer. Balanced added 12 gradient accumulation steps to the traditional Adam setting. Cyclic systematically sampled each task exactly once per update step, identical to the method used to train UMedPT. The average standard deviation across five independent experiments of the last 10 epochs of validation accuracy was $1.81 \pm 1.79\%$ for balanced sampling and $1.17 \pm 1.09\%$ for cyclic sampling.

Editor: Please include statistical information about the pretraining datasets, such as the total amount of data in pretraining, the amount of data per dataset.

Reply: Additional details on the datasets used in the pretraining of our model are now presented in Extended Data Table 4:

Extended Data Table 4 Pretraining database statistics. Image instances refer to individual 2D images that can be used directly for pretraining. Composite data types, including 3D volumes and gigapixel images, can be divided into multiple image instances per imaging study. In total, the pretraining database included more than 3 million 2D images, more than 1000 large image tiles such as tissue microarrays or whole slide sections, more than 10000 whole slide images and more than 1000 3D volumes, totalling more than 10 million annotated image instances for pretraining UMedPT.

Identifier	Description	Dataset size
Amos22-CT	Segmentation of 15 organs in abdominal CT.	200 3D-CT volumes
Conic-WSI	Nuclei detection in colon tissue from 6 different data sources with 6 classes.	4981 image instances
PICAL-MRI	Multilabel classification of clinically relevant prostate cancer (csPCa) and whether or not a lesion is visible.	1476 cases, each with 3 3D-MR sequences
Panda-WSI-Clf Panda-WSI-Seg	This data source yields two pretraining tasks. A classification task for predicting presence of tumor, and a segmentation task with the classes stroma, healthy epithelium and gleason grades 3, 4 and 5.	10616 WSI
VinBigData-CXR	Object detection in chest X-ray with 14 classes.	15000 chest X-rays
Crag-WSI	Colorectal adenocarcinoma gland segmentation.	213 image tiles (size \approx (1500, 1500))
Brats2020-MRI	Brain tumor (glioma) segmentation into five classes.	369 cases. Each case comes with 4 3D-MR sequences.
CRC-WSI	Multi-class classification of H&E stained histological images of human colorectal cancer (CRC).	100,000/7,000 image instances extracted from 86 WSI
Avaniti-WSI	Multi-label classification into four classes (benign and 3 gleason grades). We extracted patches from tissue microarrays and predict all classes present.	886 TMA
Cyto-WSI	Expert-labeled single-cell images taken from peripheral blood smears. Used as a multi-class classification task with 21 classes.	137076 image instances
Chexpert-CXR	Multilabel classification in chest X-ray with 14 classes. We use the nine classes that have a good performance when measured with the provided validation set.	223414/234 image instances
SIIM-CXR	Segment the pneumothorax area in chest X-ray.	11583 image instances
ImageNet	Multi-class natural image classification dataset with 1000 classes.	1,281,167/50,000 images
RadImageNet	Multi-class classification database developed for the purpose of pretraining medical AI. Contains 2D image instances from CT, MR and ultrasound (US)	263118/29235 CT 605408/67267 MR 350897/38988 US
COCO-Seg COCO-Det	Natural image dataset with 80 segmentation and object detection classes.	118287 image instances

Editor: As indicated by Reviewer #1, the README file should be provided.

Reply: We appreciate the remark and think that simple code reuse and reproducibility is a cornerstone of valuable research. For this reason, we have extended the original README file (located in code/readme.md) and will soon release the training framework on Github: <https://github.com/FraunhoferMEVIS/UMedPT>.

For reproducibility, we also published a Docker image (`docker pull hub.cc-asp.fraunhofer.de/mtl-torch/mtl-torch-stack:main`) that already contains all requirements to train the latest open source version of UMedPT.

Editor: Please avoid the use of the phrase "foundational model". Instead, please use "LLM".

Reply: We feel that the term Large Language Model (LLM) would be inaccurate, as our work does not include language modelling. If we have misunderstood this request, we would welcome further clarification.

Reviewer 1

Reviewer: This study developed a foundational pretrained model using a multi-task learning strategy across various biomedical modalities and label types. This model demonstrated effective knowledge transfer capabilities to reduce the amount of data and time for unseen tasks. The foundation model is valuable for biomedical imaging, especially in data-scarce scenarios. While the research holds promise, there are several concerns and areas of improvement.

Reply: We thank the reviewer for the valuable time and effort spent in reviewing our manuscript and providing constructive feedback. In response to this reviewer's comments, we have

- added the suggested database MedMNIST as a second benchmark for our method;
- performed additional experiments to gain insights into combining 3D context with 2D pretraining;
- performed ablation studies;
- gained insights into the phenomenon that more data can be detrimental to the prediction performance for downstream tasks.

Reviewer: How the datasets for pretraining are selected? Why them?

Reply: We appreciate the reviewer's suggestion. We have added details about the process of building the pretraining database (page 18, lines 563-581):

"Pretraining Tasks

We selected 15 publicly available datasets for pretraining and extracted 17 tasks from them. Several criteria guided the selection of datasets:

- **Availability:** All datasets should be publicly available.
- **Clinical Relevance:** Datasets should include imaging modalities that are widely used in radiology and pathology. For that reason, we included tasks from histopathology, X-ray, and tomography.
- **Diversity of Label Types:** Where possible, we included tasks with a classification, segmentation, and detection label type for each category.
- **Performance:** We prioritized datasets that demonstrated satisfactory performance when trained individually. We defined satisfactory performance as either aligning with the metrics reported by the dataset creators where available, or passing a plausibility check conducted by a medical expert.

We included four auxiliary datasets for the purpose of meta-learning. These datasets were not intended to directly improve a specific clinical application, but rather to enhance the model's general image understanding capabilities, drawing inspiration from the strong foundational capabilities of ImageNet pre-trained models. Detailed statistics on the pretraining database are reported in Table 4."

Reviewer: Is ImageNet being used in reference to ImageNet 1K? Any performance comparisons with ImageNet 21K pre-training? As I understand it, pre-training with 21K classes exhibits notably stronger performance.

Reply: In the present study, we used ImageNet-1K. We specify the used version of ImageNet in several places in the manuscript, e.g.

Page 4, Figure 1: “The performance of UMedPT was compared with that of ImageNet-1K pretraining”.

Page 5, lines 146-147: “we compared UMedPT to [...] ImageNet-1K”.

As the reviewer rightly points out, recent research suggests that pretraining with ImageNet-21K may yield better performance.

However, the following practical considerations led us to use ImageNet-1K:

- We still perceive ImageNet-1k as the standard for medical pretraining.
- Model Compatibility: For our chosen architecture, specifically the Swin Transformer, pretrained weights on ImageNet-21K are not available. However, such external weights are important for a fair assessment.
- In addition, we tried to get access to ImageNet-21K on the 16th of April 2023. Unfortunately, up to now our request remained unanswered. In consequence, we were not able to integrate the dataset.

Reviewer: In lines 160-162, “Surprisingly, for UMedPT, increasing the training data beyond 1% did not enhance the model’s performance and sometimes tended to degrade it.” In theory, an increase in data volume can enhance the model’s performance. Are there any reasons analyzed here for the weakening of performance?

Reply: This is indeed a surprising finding, which was also commented by the editor and the other reviewer. We agree that, intuitively, collecting more data for an unseen clinical target task should improve performance. The phenomenon of a smaller dataset beating a larger dataset was observed primarily for fine-tuning, suggesting that the current method of fine-tuning may in some cases lead to “catastrophic forgetting” of the well generalizing multi-task parameters. Catastrophic forgetting is a phenomenon in which AI systems lose information from previous tasks as they learn new ones.

In fact, our training schedule used a fixed number of epochs. Consequently, as the size of the fine-tuning dataset increased, there were naturally more update steps. This gives the training process more time to overwrite or ‘forget’ the useful features learned during the pretraining phase, especially as all parameters are allowed to change freely during fine-tuning.

To test the hypothesis, we performed an additional experiment (page 27, lines 879-899):

“In our clinical benchmark, we observed that increasing the amount of data for a target task could paradoxically lead to a decrease in performance during the fine-tuning phase. To explore the potential role of catastrophic forgetting, where neural networks lose previously learned information as they acquire new knowledge, we designed an experiment focused on the fine-tuning phase of our model.

We pretrained a multi-task deep learning network on the MedMNIST database. From this, we selected four tasks (SynapseMNIST3D, VesselMNIST3D, BreastMNIST, PneumoniaMNIST) that had shown improved performance with multi-task learning compared to single-task learning. Our aim was to analyse how further training of these tasks, individually with their full datasets, would affect their test accuracy.

For the experiment, we first measured the test accuracy for the four selected tasks after multi-task learning with with 12 MedMNIST tasks for 100 epochs. We then continued to train the selected tasks individually using 100% of their respective training data for 100 epochs, after which we recorded the test accuracy again.

If, after this further individual fine-tuning, test accuracy fell to levels comparable to those achieved by single-task training, this would indicate that catastrophic forgetting had occurred.”

Our experiments revealed that it depended on the dataset whether catastrophic forgetting occurred during fine-tuning or not. We present the findings in page 11, lines 282-296:

“[...] The results varied between datasets, suggesting that whether catastrophic forgetting affects fine-tuning is not consistent across datasets and may be task-dependent:

- SynapseMNIST3D: $83.81 \pm 0.31\%$ -> $82.90 \pm 0.66\%$ (decrease)
- VesselMNIST3D: $93.66 \pm 0.31\%$ -> $93.77 \pm 0.87\%$ (no decrease)
- BreastMNIST: $86.92 \pm 1.04\%$ -> $85.90 \pm 0.91\%$ (decrease)
- PneumoniaMNIST: $91.54 \pm 0.48\%$ -> $91.70 \pm 0.49\%$ (no decrease)”

Unfortunately, at the moment we do not have a solution to predict the appearance of catastrophic forgetting.

The phenomenon of catastrophic forgetting could be addressed by tools developed for language models. In this context, parameter-efficient fine-tuning techniques could be BitFit (where only the bias-terms are fine-tuned) and LoRA (where a small number of additional parameters are fine-tuned). We discuss these potential future improvements on our approach in page 12, lines 353-363:

“Investigating catastrophic forgetting in target tasks

In some cases we observed that the performance of UMedPT decreased as the size of the training dataset increased. This could be due to catastrophic forgetting of the well generalizing features learned during pretraining. However, this effect was not consistent, suggesting that it may be dataset dependent.

Recently, more sophisticated fine-tuning strategies have been proposed for foundational models in natural language processing, such as BitFit [33], where only the bias-terms are fine-tuned, or LoRA [34], where a small number of additional parameters are fine-tuned. A training configuration targeted specifically to foundational vision models might combine the strengths of frozen and fine-tuned training configurations, and could improve on the problem of catastrophic forgetting [25].”

Reviewer: In line 460: “To accommodate these different data types, the encoder of UMedPT used a standardized 2D image input format.” To my knowledge, 3D networks perform better with 3D images compared to 2D networks. In the study, when standardizing 3D images into 2D images, a significant amount of the original three-dimensional information is lost. Has the performance of this foundational model been directly compared to the training performance of 3D networks for 3D images?

Reply: We appreciate the reviewer's point and addressed it by performing additional experiments using the MedMNIST benchmark. The editor has adopted this point as well. We agree that the conversion of 3D images into 2D slices during pretraining results in a loss of spatial context, and quantifying this loss is important for the interpretation of our results. To quantify the performance differences between 2D pre-training for 3D tasks and 3D convolutional neural networks, we

performed additional analyses, which are described in the Methods section of the additional MedMNIST experiments on page 26, lines 842-862:

“Assessing 3D context preservation in pretraining

Transforming inherently 3D medical imaging data into 2D slices for pretraining purposes can result in the loss of three-dimensional contextual information. This dilemma presents a challenge when building a unified database for pretraining: while large pretraining databases are populated with 2D images, many tomographic medical imaging techniques capture complex anatomical structures in three dimensions.

To evaluate the ability to maintain three-dimensional context in our pre-training approach, we trained MedMNIST multi-task networks that handled 2D and 3D tasks simultaneously. For the 3D tasks, we used a simple strategy based on a learned weighted average across slices with the classification task described in section 5. Intuitively, this allows the network to learn focusing on the most relevant slices of a 3D case.

Our objective was to determine the effectiveness of this strategy compared to a network using three-dimensional convolutional layers. We assessed this by directly comparing our learned weighted average-based classification results with the performance reported by the MedMNIST authors using a standard 3D CNN. For a useful comparison, we analysed the results not only for a Resnet-50 CNN [53], as used by the MedMNIST authors, but also for the Swin Tiny Transformer [38], which is a smaller variant of the encoder architecture used in UMedPT.”

We present the results in page 10, lines 253-264:

“Assessing 3D context preservation in pretraining

To assess the impact of transforming inherently 3D medical imaging data into 2D slices for pretraining, we evaluated single-task learning with 2D data, single-task learning with original 3D data [24] and multi-task learning with 2D data. When only taking the 3D tasks, the average accuracies were:

- 83.22 ± 1.61% for single-task learning pretrained using Imagenet;
- 83.76% for the single-task MedMNIST 3D CNN;
- 86.46 ± 1.13% for multi-task learning with weighted averaging.

The results showed that single-task 3D CNNs performed better than single-task 2D CNNs pretrained using Imagenet. However, multi-task learning 2D networks outperformed the single-task 3D CNNs. We present the details in Extended Data Table 5.”

Here we show an excerpt from Extended Data Table 5 with only the 3D results for the CNNs:

Extended Data Table 5 MedMNIST test performance. Multi-task learning (MTL) networks were trained using all tasks, including both 2D and 3D tasks. Single-task (ST) were trained independently. Metrics are reported as mean \pm standard deviation in percentage. The reference accuracy (ACC Ref) and area under the curve (AUC Ref) are taken from the publication associated with the MedMNIST database [52].

CNN	ACC MTL	AUC MTL	ACC ST	AUC ST	ACC Ref	AUC Ref
AdrenalMNIST3D	83.89 \pm 0.88%	86.85 \pm 1.95%	80.87 \pm 3.40%	76.96 \pm 17.78%	74.50%	82.80%
NoduleMNIST3D	85.94 \pm 0.97%	85.95 \pm 1.45%	86.13 \pm 0.54%	90.87 \pm 0.71%	84.70%	87.50%
OrganMNIST3D	87.11 \pm 1.25%	99.00 \pm 0.25%	86.82 \pm 2.74%	98.69 \pm 0.25%	88.30%	99.40%
SynapseMNIST3D	81.08 \pm 1.50%	81.81 \pm 1.32%	73.52 \pm 0.55%	65.62 \pm 6.14%	79.50%	85.10%
VesselMNIST3D	94.29 \pm 0.45%	95.64 \pm 0.75%	88.74 \pm 0.00%	69.60 \pm 2.80%	91.80%	90.70%

Reviewer: In lines 388-389, "No information on the dataset's length was needed beforehand, which allowed each epoch to have a different length depending on data augmentation." Confused about the varying lengths per epoch. Are different batch sizes used for each task?

Reply: We indeed used different batchsizes for each task depending on the memory requirements of its label type and the spatial dimensions of the image input. However, this does not change the number of image instances that a task contributes to a training epoch. We apologize for the confusing wording and have rephrased the respective part of the Methods section:

Page 15, lines 453-463: "To accommodate datasets of different sizes, we implemented an "infinite task sampler", which yielded one training batch of each task for every optimization step. Problematically, for 3D volumes the number of image instances used for training could depend on the randomly chosen axis of slicing, while for gigapixel images, the random zoom level could influence the number of image instances used for training. Our task sampler independently restarted the data loading for a task once all of its data points had been used. As a result, no information on the dataset's length was needed beforehand, allowing each epoch to have a different length depending on data augmentation."

Reviewer: In lines 586-590, "For this reason, we did not use a validation set in our experiments." How to determine the endpoint of training without validation set for downstream tasks?

Reply: We allocated a fixed computational budget for all evaluations and used the last model state. We added the missing information to the methods section:

Page 22, lines 663-668: "We developed the downstream training schedule and tuned the hyperparameters using a simple synthetic dataset and ran the clinical evaluation exactly once without further hyperparameter tuning. We evaluated the model after training for a fixed number of epochs. For this reason, we did not use a validation set in our experiments."

Page 22, lines 673-676: "Two distinct usage settings were considered in our evaluation: frozen and fine-tuning. [...] Both frozen and fine-tuning were trained for 100 epochs each."

In few-shot learning, curating a representative validation set can be challenging. This is because the available data are so limited that any data used for validation purposes could be better used in the training process to improve the performance of the model.

To avoid overfitting, which can be addressed by model selection with a validation set, we implemented strong regularization techniques during model training, such as domain-specific augmentation, dropout and weight decay.

This approach was validated by an additional experiment using the MedMNIST database; training a convolutional neural network (CNN) under these conditions resulted in an average test accuracy of $86.17 \pm 0.84\%$ with model selection using the validation set, compared to $86.35 \pm 1.01\%$ with the final model state after 100 epochs. The average was taken over all tasks with 5 repetitions.

Reviewer: The study claims a contribution that one model covers various data modalities and tasks. But the comparison over other methods is limited, including different pretraining methods, backbones, learning strategies, etc. If the pretrained model is universal, I suggest to report experiments on MedMNIST (Scientific Data, 2023), where 12 2D data and 6 3D data are standardized to be compared with other methods.

Reply: We appreciate the reviewer's suggestion to test the generality of our pretraining method with the MedMNIST database. We performed additional experiments on

- pretraining alternatives: we employed alternative scheduling strategies (2a) and used different backbones (2b);
- quantification of the 3D performance difference when applying 2D methods, 3D methods, and pretrained 2D methods (1d);
- reproducing and understanding the phenomenon of weakening performance with more data (1c).

We introduced the experiment as follows (page 25, lines 814-841):

“Experiments with MedMNIST

In addition to the primary studies conducted with UMedPT, we also performed supplementary analyses using the MedMNIST database [24, 52].

MedMNIST is a collection of 18 medical datasets that were downsampled to enable quick experimentation with medical datasets. We used the same training schedule and hyperparameters as for the main study. Nevertheless, MedMNIST differs from the pretraining database of UMedPT in the following aspects:

- **Spatial size:** MedMNIST images are scaled down to a uniform 28x28 (2D) or 28x28x28 (3D) size, while UMedPT was trained using images at their original dimensions;
- **Task type:** MedMNIST exclusively includes classification tasks. UMedPT was trained with classification, segmentation and object detection tasks;
- **Augmentations:** We applied weak standard augmentations to the MedMNIST datasets, avoiding flips, whereas UMedPT used domain-specific augmentations tailored to each task type within its training set;
- **Data loading:** MedMNIST datasets have a fixed dataset length. For the UMedPT database, we developed domain specific data loading strategies to be able to augment loading of the raw data;
- **Meta-learning:** MedMNIST does not include any meta-learning datasets, while UMedPT includes four large datasets for the purpose of general applicability, including non-medical data.

Out of the 18 MedMNIST datasets, 12 were selected. We excluded 3 datasets (Organ{A,C,S}MNIST) because they were composed of 2D images from one of the included 3D datasets. Further 3 datasets (RetinaMNIST, TissueMNIST & FractureMNIST3D) were excluded as the authors had reported a low

performance. In total, the subset included 370,980 imaging studies, or 1,087,104 image instances for training, validation, and testing.”

We present the new results in in Extended Data Table 5:

Extended Data Table 5 MedMNIST test performance. Multi-task learning (MTL) networks were trained using all tasks, including both 2D and 3D tasks. Single-task (ST) were trained independently. Metrics are reported as mean \pm standard deviation in percentage. The reference accuracy (ACC Ref) and area under the curve (AUC Ref) are taken from the publication associated with the MedMNIST database [52].

CNN	ACC MTL	AUC MTL	ACC ST	AUC ST	ACC Ref	AUC Ref
BloodMNIST	94.31 \pm 0.41%	99.64 \pm 0.02%	94.49 \pm 0.41%	99.70 \pm 0.06%	95.60%	99.70%
BreastMNIST	87.18 \pm 0.41%	86.81 \pm 1.58%	81.54 \pm 1.59%	83.86 \pm 1.49%	81.20%	85.70%
ChestMNIST	94.76 \pm 0.01%	73.34 \pm 0.30%	93.42 \pm 0.29%	66.95 \pm 0.30%	94.70%	76.90%
DermaMNIST	76.55 \pm 0.67%	91.97 \pm 0.20%	76.81 \pm 0.42%	92.00 \pm 0.33%	73.50%	91.30%
OCTMNIST	73.32 \pm 1.47%	95.53 \pm 0.31%	75.16 \pm 2.01%	95.03 \pm 0.42%	76.20%	95.20%
PathMNIST	85.97 \pm 1.94%	98.58 \pm 0.26%	87.92 \pm 1.03%	98.78 \pm 0.03%	91.10%	99.00%
PneumoniaMNIST	91.76 \pm 0.93%	97.91 \pm 0.25%	90.26 \pm 1.24%	97.97 \pm 0.18%	85.40%	94.80%
AdrenalMNIST3D	83.89 \pm 0.88%	86.85 \pm 1.95%	80.87 \pm 3.40%	76.96 \pm 17.78%	74.50%	82.80%
NoduleMNIST3D	85.94 \pm 0.97%	85.95 \pm 1.45%	86.13 \pm 0.54%	90.87 \pm 0.71%	84.70%	87.50%
OrganMNIST3D	87.11 \pm 1.25%	99.00 \pm 0.25%	86.82 \pm 2.74%	98.69 \pm 0.25%	88.30%	99.40%
SynapseMNIST3D	81.08 \pm 1.50%	81.81 \pm 1.32%	73.52 \pm 0.55%	65.62 \pm 6.14%	79.50%	85.10%
VesselMNIST3D	94.29 \pm 0.45%	95.64 \pm 0.75%	88.74 \pm 0.00%	69.60 \pm 2.80%	91.80%	90.70%
Transformer						
BloodMNIST	95.96 \pm 0.11%	99.77 \pm 0.01%	95.70 \pm 0.30%	99.76 \pm 0.04%	-	-
BreastMNIST	86.92 \pm 1.04%	86.26 \pm 1.38%	86.41 \pm 1.24%	86.81 \pm 0.78%	-	-
ChestMNIST	94.75 \pm 0.01%	75.19 \pm 0.35%	93.69 \pm 0.10%	68.04 \pm 0.42%	-	-
DermaMNIST	79.09 \pm 0.45%	93.04 \pm 0.30%	79.20 \pm 0.57%	93.34 \pm 0.31%	-	-
OCTMNIST	74.08 \pm 1.31%	95.43 \pm 0.41%	73.62 \pm 0.92%	93.55 \pm 0.60%	-	-
PathMNIST	90.60 \pm 0.48%	99.15 \pm 0.06%	92.11 \pm 0.98%	99.34 \pm 0.13%	-	-
PneumoniaMNIST	91.54 \pm 0.48%	98.22 \pm 0.37%	88.85 \pm 1.62%	97.69 \pm 1.01%	-	-
AdrenalMNIST3D	82.01 \pm 1.11%	85.28 \pm 0.77%	76.85 \pm 0.00%	66.54 \pm 8.03%	-	-
NoduleMNIST3D	83.81 \pm 1.74%	86.39 \pm 2.37%	83.29 \pm 2.56%	84.52 \pm 5.25%	-	-
OrganMNIST3D	84.89 \pm 1.17%	98.81 \pm 0.07%	90.52 \pm 2.06%	99.28 \pm 0.12%	-	-
SynapseMNIST3D	83.81 \pm 0.31%	85.75 \pm 0.89%	73.01 \pm 0.00%	57.11 \pm 4.79%	-	-
VesselMNIST3D	93.66 \pm 0.31%	93.94 \pm 1.40%	88.74 \pm 0.00%	66.98 \pm 5.30%	-	-

Reviewer: The paper lacks ablation study involving the proposed algorithm gradient accumulation and traditional training schemes. There can be many variants, including loss weights and dataset balanced sampling.

Reply: We appreciate the reviewer's feedback and addressed it by including data from additional experiments. The editor has adopted this point. For convenience, we will repeat our response here:

We understand the importance of demonstrating whether our method is superior to or differs from traditional training schemes, and we agree that this information would be highly interesting for the readers of the article. Therefore, we extended the UMedPT ablation study that was part of the initial submission. In this context, we have performed several additional ablation studies related to the training strategy and included the following paragraph in the Methods section (page. 27, lines. 868-878):

“Investigating training schemes

All datasets in MedMNIST have a fixed number of cases. This distinction enabled us to conduct ablation studies comparing infinite task sampling with balanced sampling based on dataset size. Besides this, we used the same training schedule and hyperparameters as in the main study, and

accumulated the gradients of as many steps as there were tasks. In addition, for comparison with traditional training schemes, we used the same setting without gradient accumulation and also with the SGD optimizer instead of Adam.

To quantify the effect of task scheduling, we reported the stabilities of the training processes by the standard deviations of the validation performances over the last 10 training epochs.”

We report the new results in page 10, lines 272-281:

“The exploration of training schemes showed that balanced (by dataset size) and cyclic sampling (as in UMedPT) exhibited similar behaviour in terms of convergence. However, balanced sampling occasionally showed reduced stability; it yielded a standard deviation of $1.81 \pm 1.79\%$ in validation accuracy over the previous ten epochs, across five different experiments. In comparison, cyclic sampling showed a more stable training process, achieving a comparatively lower standard deviation of $1.17 \pm 1.09\%$. When gradient accumulation was excluded, the resulting performance deteriorated, accompanied by longer convergence times. These results are shown in Extended Data Figure 2b.”

Extended Data Fig. 2 MedMNIST training convergence. a Architecture comparison [...] **b** Comparison of training schemes for the Swin Transformer tiny architecture. Traditional SGD used SGD optimizer without momentum and without gradient accumulation. Traditional Adam used the same setting but with the Adam optimizer. Balanced added 12 gradient accumulation steps to the traditional Adam setting. Cyclic systematically sampled each task exactly once per update step, identical to the method used to train UMedPT. The average standard deviation across five independent experiments of the last 10 epochs of validation accuracy was $1.81 \pm 1.79\%$ for balanced sampling and $1.17 \pm 1.09\%$ for cyclic sampling.

Reviewer: Only one backbone is used in this study. Convnet-based models are also encouraged.

Reply: We agree that including a convolutional network-based model provides additional insights into the performance of our pretraining strategy across different types of backbones. To address this point, we included a ResNet (ResNet-50 with 23.5m parameters) alongside a Swin Transformer of comparable capacity (Swin-Tiny with 27.5m parameters) in the additional MedMNIST experiments.

Page 27, lines 863-867: “For quantifying the effect of the encoder’s architecture, we chose the ResNet-50 as convolutional neural network (CNN) and the tiny variant of the Swin Transformer

because they are similar in size. The Swin Transformer has 27,582,570 trainable parameters compared to 23,508,032 for the ResNet-50 CNN."

We present the results in page 10, lines 265-271:

"Comparing convolutional networks and transformer

The comparison of the Swin Transformer and ResNet-50 CNN architectures showed a minimal impact on model performance for the MedMNIST database. The Swin Transformer achieved an average test accuracy of $86.76 \pm 0.79\%$ over 5 repetitions, while the ResNet-50 CNN achieved an accuracy of $86.34 \pm 1.01\%$. In addition, a discrepancy in training convergence rates was observed between the two architectures, as shown in Extended Data Figure 2a."

Extended Data Fig. 2 MedMNIST training convergence. **a** Architecture comparison between ResNet-50 [53] and Swin Transformer in the "tiny" variant [38], evaluated on combined 2D and 3D multi-task trainings. **b** [...]

Reviewer: I don't understand the difference between UMedPT, UMedPT-fixed and UMedPT-affine, the description in the paper is hard to follow.

We apologize for the confusion and have clarified the distinctions between UMedPT, UMedPT-fixed, and UMedPT-affine in the short introduction in the results section:

Page 5, lines 135-143: "Ablation studies are included in Extended Data Tables 1, 2 and 3. UMedPT-fixed consistently used an image size of (224, 224), while UMedPT used the full image dimensions for each task. In addition, we tested UMedPT-affine, which also used image dimensions of (224, 224) but added a learnable bias and scaling parameter to UMedPT's static layernorms, adding an affine transformation. In our evaluations across various tasks, UMedPT outperformed ImageNet by an average of 8.5% and surpassed UMedPT-fixed by 2.97%. Compared to UMedPT-fixed, UMedPT-affine showed an average performance gain of 0.37%."

In addition, we clarified the setup of UMedPT-fixed:

Page 18, lines 552-558: "In an ablation study, we evaluated how a uniform image size affected the performance of our model. We trained a version called UMedPT-fixed and downsized all image instances to 224 x 224 pixels. This contrasts with our standard UMedPT, where the gradient

accumulation technique allows for dynamic image sizes to suit the requirements of each task. Besides this, the preparation of the 2D image inputs for UMedPT-fixed followed the same process as for the original UMedPT."

Additionally, we added details regarding UMedPT-affine:

Page 17, lines 523-530: "We empirically analysed the effect of layernorms with affine parameters on our approach using an adaptation of UMedPT (UMedPT-affine). UMedPT used layernorms without parameters in the form $y = \frac{x-\mu}{\sigma}$, where $\sigma = \sqrt{\text{var } x + \epsilon}$ and x was the input. The mean μ and standard deviation σ were computed over all channels of an input, but not over the batch dimension. UMedPT-affine added trainable parameters including a bias β and a scaling factor γ in the form $y = \gamma \frac{x-\mu}{\sigma} + \beta$ for each channel. Similar to UMedPT-fixed, UMedPT-affine was only used with images downscaled to 224 x 224 pixels."

Reviewer: The article lacks statistical information about the pretraining datasets, such as the total amount of data in pretraining, the amount of data per dataset.

Reply: We appreciate the reviewer's feedback. The editor has adopted this point. For convenience, we will repeat the response here:

Additional details on the datasets used in the pretraining of our model are now presented in Extended Data Table 4:

Extended Data Table 4 Pretraining database statistics. Image instances refer to individual 2D images that can be used directly for pretraining. Composite data types, including 3D volumes and gigapixel images, can be divided into multiple image instances per imaging study. In total, the pretraining database included more than 3 million 2D images, more than 1000 large image tiles such as tissue microarrays or whole slide sections, more than 10000 whole slide images and more than 1000 3D volumes, totalling more than 10 million annotated image instances for pretraining UMedPT.

Identifier	Description	Dataset size
Amos22-CT	Segmentation of 15 organs in abdominal CT.	200 3D-CT volumes
Conic-WSI	Nuclei detection in colon tissue from 6 different data sources with 6 classes.	4981 image instances
PICAL-MRI	Multilabel classification of clinically relevant prostate cancer (csPCa) and whether or not a lesion is visible.	1476 cases, each with 3 3D-MR sequences
Panda-WSI-Clf Panda-WSI-Seg	This data source yields two pretraining tasks. A classification task for predicting presence of tumor, and a segmentation task with the classes stroma, healthy epithelium and gleason grades 3, 4 and 5.	10616 WSI
VinBigData-CXR	Object detection in chest X-ray with 14 classes.	15000 chest X-rays
Crag-WSI	Colorectal adenocarcinoma gland segmentation.	213 image tiles (size \approx (1500, 1500))
Brats2020-MRI	Brain tumor (glioma) segmentation into five classes.	369 cases. Each case comes with 4 3D-MR sequences.
CRC-WSI	Multi-class classification of H&E stained histological images of human colorectal cancer (CRC).	100,000/7,000 image instances extracted from 86 WSI
Avaniti-WSI	Multi-label classification into four classes (benign and 3 gleason grades). We extracted patches from tissue microarrays and predict all classes present.	886 TMA
Cyto-WSI	Expert-labeled single-cell images taken from peripheral blood smears. Used as a multi-class classification task with 21 classes.	137076 image instances
Chexpert-CXR	Multilabel classification in chest X-ray with 14 classes. We use the nine classes that have a good performance when measured with the provided validation set.	223414/234 image instances
SIIM-CXR	Segment the pneumothorax area in chest X-ray.	11583 image instances
ImageNet	Multi-class natural image classification dataset with 1000 classes.	1,281,167/50,000 images
RadImageNet	Multi-class classification database developed for the purpose of pretraining medical AI. Contains 2D image instances from CT, MR and ultrasound (US)	263118/29235 CT 605408/67267 MR 350897/38988 US
COCO-Seg COCO-Det	Natural image dataset with 80 segmentation and object detection classes.	118287 image instances

Reviewer: In Extended Data Table 1, “PPneumo-CXR” should be “Pneumo-CXR”; If UMedPT-A and UMedPT-affine refer to the same thing, their naming needs to be unified; Is the result “58.21±9.50%” representing the mean and standard deviation? It’s not specified in the table header.

Reply: Thank you for the valuable feedback. We corrected the labels, unified the names, and specified in all extended data tables that the table values refer to mean and standard deviations. For example:

Page 36: “The left pair of columns shows results with a frozen encoder, while the right pair shows results with fine-tuning. F1-scores are reported as mean \pm standard deviation in percentage. P-values...”

Reviewer: In line 239 and 304, SemiCOL challenge dataset should be classification task, not tumor detection subtask.

Reply: Thank you for pointing out this point of confusion. We clarified the respective results section by describing the task as “classifying colorectal cancer histopathology images into tumor and healthy”:

Page 9, lines 237-239: “[...] it was applied in the classification task of the SemiCOL challenge. This task required the classification of colorectal cancer histopathology images into tumor and healthy.”

And in the discussion:

Page 13, lines 368-371: “We tested this using the SemiCOL challenge [...] in the task of classifying colorectal cancer histopathology images into tumor and healthy”

Reviewer: Lack of clarity due to lack of README.

The absence of instructions on project structure, environment setup, required packages, and reproduction steps is a major obstacle. Yet, the presence of many test files is a positive sign, showing extensive testing of the code blocks.

Reply: We appreciate the positive remark regarding our extensive testing strategy. We also think that simple code reuse and reproducibility is a cornerstone of valuable research. For this reason, we have extended the original README file (located in code/readme.md) and will release the training framework on Github: <https://github.com/FraunhoferMEVIS/UMedPT>.

For reproducibility, we also published a Docker image (docker pull hub.cc-asp.fraunhofer.de/mtl-torch/mtl-torch-stack:main) that already contains all the requirements to train the latest open source version of UMedPT.

Reviewer 2

Reviewer: Training models with scarce data is a major problem in the field of image analysis. In addition, combining multimodal datasets (eg imaging, pathology) is another major challenge for precision medicine and it is not usually the case that image analysis solutions at the tissue level can successfully be applied to in vivo imaging. The authors have used a multi-task foundational model to overcome these issues by using simultaneous training of a single model that generalizes across multiple tasks. This could therefore be applied to the many small datasets that are currently available given the absence of larger datasets. The approach used here included three supervised label types: object detection, segmentation, and classification. The authors developed a fully supervised foundational model for biomedical imaging which they termed UMedPT, using 17 tasks based on 15 datasets and their original annotations.

Reply: We thank the reviewer for the thoughtful assessment of our work and the insightful comments regarding the challenges that our study aims to address.

Reviewer: The authors divided the assessment in two ways: in-domain benchmark to assess the applicability of UMedPT to problems closely related to its training database and the out-of-domain benchmark to evaluate its performance in unfamiliar domains. The UMedPT outperformed the pretrained ImageNet network in both in and out-of-domain tasks. Their results were impressive with UMedPT matching the best performance of the ImageNet baseline over all configurations using only 1% of the original training data. Increasing the training data beyond 1% did not enhance performance and sometimes tended to degrade it – this is counterintuitive, and the authors should provide some explanation for this result.

Reply: We appreciate the reviewer's insightful feedback and addressed it including additional experiments. The editor and the other reviewer have adopted this point as well. For convenience, we will repeat our response from above:

We agree that, intuitively, collecting more data for an unseen clinical target task should improve performance. The phenomenon of a smaller dataset beating a larger dataset was observed primarily for fine-tuning, suggesting that the current method of fine-tuning may in some cases lead to “catastrophic forgetting” of the well generalizing multi-task parameters. Catastrophic forgetting is a phenomenon in which AI systems lose information from previous tasks as they learn new ones.

In fact, our training schedule used a fixed number of epochs. Consequently, as the size of the fine-tuning dataset increased, there were naturally more update steps. This gives the training process more time to overwrite or ‘forget’ the useful features learned during the pretraining phase, especially as all parameters are allowed to change freely during fine-tuning.

To test the hypothesis, we performed an additional experiment (page 27, lines 879-899):

“In our clinical benchmark, we observed that increasing the amount of data for a target task could paradoxically lead to a decrease in performance during the fine-tuning phase. To explore the potential role of catastrophic forgetting, where neural networks lose previously learned information as they acquire new knowledge, we designed an experiment focused on the fine-tuning phase of our model.

We pretrained a multi-task deep learning network on the MedMNIST database. From this, we selected four tasks (SynapseMNIST3D, VesselMNIST3D, BreastMNIST, PneumoniaMNIST) that had shown improved performance with multi-task learning compared to single-task learning. Our aim

was to analyse how further training of these tasks, individually with their full datasets, would affect their test accuracy.

For the experiment, we first measured the test accuracy for the four selected tasks after multi-task learning with with 12 MedMNIST tasks for 100 epochs. We then continued to train the selected tasks individually using 100% of their respective training data for 100 epochs, after which we recorded the test accuracy again.

If, after this further individual fine-tuning, test accuracy fell to levels comparable to those achieved by single-task training, this would indicate that catastrophic forgetting had occurred.”

Our experiments revealed that it depended on the dataset whether catastrophic forgetting occurred during fine-tuning or not. We present the findings in page 11, lines 282-296:

“[...] The results varied between datasets, suggesting that whether catastrophic forgetting affects fine-tuning is not consistent across datasets and may be task-dependent:

- SynapseMNIST3D: $83.81 \pm 0.31\%$ -> $82.90 \pm 0.66\%$ (decrease)
- VesselMNIST3D: $93.66 \pm 0.31\%$ -> $93.77 \pm 0.87\%$ (no decrease)
- BreastMNIST: $86.92 \pm 1.04\%$ -> $85.90 \pm 0.91\%$ (decrease)
- PneumoniaMNIST: $91.54 \pm 0.48\%$ -> $91.70 \pm 0.49\%$ (no decrease)”

Unfortunately, at the moment we do not have a solution to predict the appearance of catastrophic forgetting.

The phenomenon of catastrophic forgetting could be addressed by tools developed for language models. In this context, parameter-efficient fine-tuning techniques could be BitFit (where only the bias-terms are fine-tuned) and LoRA (where a small number of additional parameters are fine-tuned). We discuss these potential future improvements on our approach in page 12, lines 353-363:

“Investigating catastrophic forgetting in target tasks

In some cases we observed that the performance of UMedPT decreased as the size of the training dataset increased. This could be due to catastrophic forgetting of the well generalizing features learned during pretraining. However, this effect was not consistent, suggesting that it may be dataset dependent.

Recently, more sophisticated fine-tuning strategies have been proposed for foundational models in natural language processing, such as BitFit [33], where only the bias-terms are fine-tuned, or LoRA [34], where a small number of additional parameters are fine-tuned. A training configuration targeted specifically to foundational vision models might combine the strengths of frozen and fine-tuned training configurations, and could improve on the problem of catastrophic forgetting [25].”

Reviewer: The examples given include classification of colorectal and breast cancer on pathological slides, diagnosing pneumonia/TB on chest X-ray, and brain tumours/organ segmentation on MRI. It is important to understand how all of these were validated: there is mention of two expert pathologists annotating the breast cancer slides but it is not clear how the diagnoses were confirmed on some of the other datasets (eg TB).

Reply: Thank you for your feedback and inquiry about the validation process for the datasets. We revised the respective paragraphs about each dataset and now describe the authors’ annotation and verification procedure:

Regarding the tuberculosis task “Tuber-CXR”, the labels of a subset of 68 images were verified by consensus annotations of two radiologists:

Page 23, lines 734-737: “The images were collected from routine hospital practice over a period of one month. For a subset of 68 images, two radiologists provided consensus annotations to confirm the established ground truth of the dataset.”

For CNS neoplasia diagnosis, three experienced radiologists annotated the imaging studies:

Page 24, lines 741-743: “The slices originate from T1, T2 and FLAIR sequences and were selected by the authors of the dataset following manual annotation by three experienced radiologists.”

For CRC-WSI, the annotation team included pathologists and only clear regions were included:

Page 23, lines 700-703: “The authors of the dataset [3], including pathologists, manually delineated regions corresponding to pure tissue textures to generate labels and extract image patches. Images with artefacts such as tissue folds or without tumor components were excluded.”

In the task of predicting the presence of pneumonia in a pediatric cohort, the dataset was cleaned up, graded by two physicians, and the test data was reviewed by another expert:

Page 23, lines 712-717: “The images were acquired as part of the routine clinical care of the patients. To generate a high quality dataset for model training, the authors performed an initial screening on the dataset to exclude poor quality or unreadable scans. Then, two expert physicians annotated the remaining images and classified them for the presence of pneumonia. As an additional quality measure, a third expert reviewed the test set to verify the accuracy of the diagnoses.”

For BreakHis (BC-BHis-MIC), the dataset creators mentioned the additional verification using immunohistochemistry:

Page 24, lines 770-773: “To determine the labels, initial identification of tumor regions within each slide was performed by an anatomopathologist. Then, final diagnoses were made by experienced pathologists, with additional validation provided by other methods of analysis such as immunohistochemistry.”

The labels of OrganSeg-MRI were refined and validated through an iterative process involving eight radiologists:

Page 24, lines 780-783: “The segmentation labels were created through a collaborative effort involving 5 junior radiologists and 3 senior radiologists, who iteratively reviewed and refined the labels using semi-automated tools.”

Reviewer: The approach for the colorectal cancer slide was applied to data acquired from multiple separate sites showing its applicability on data not from the primary training set. Was this multi-site approach also performed on the X-ray and MRI data? The authors state that these foundational models should be robust to multi-center variances, thereby improving generalizability, but appear to provide the evidence for this from histological analysis only.

Reply: With this exploratory experiment, we wanted to answer the question why deep learning systems often do not generalize well outside their training data and often fail due to subtle multi-site variances. As the experiment was part of a challenge, it provided a unique opportunity of an unbiased evaluation of the proposed method but is difficult to conduct timely for the other modalities.

We were not able to perform such an evaluation for x-ray and MRI data and now discuss this in the manuscript:

Page 13, lines 379-383: "However, the challenge evaluation was limited to a single histological imaging task, while multi-center robustness is a major obstacle for deep learning systems in tomographic and X-ray imaging as well [36]. A systematic assessment of UMedPT for multi-center robustness including the use of MRI and X-ray data as well as training speed poses a task for future studies."

Reviewer: In summary, the authors have presented some very interesting work on how a novel multitask training strategy can be used for unseen target tasks and scarce data. It would be interesting to understand the authors views on how far this approach could be extended to other out-of-domain tasks: what are the limits of how this approach could be applied to new data and what are the factors affecting these limits?

Reply: We appreciate the reviewer's insightful inquiry on the scalability and limitations of our multi-task training strategy when applied to new, out-of-domain tasks.

Our transfer performance to target tasks is limited by the similarity between training data and new target tasks. The more similar the pretraining and target tasks are, the smaller the amount of training data required for high performance. Consequently, we believe that the extensibility of UMedPT is limited by its training database.

Encouragingly, we did not observe any saturation with respect to the data, the variety of tasks, or the number of different label types that we can include in the pretraining of a single universal model. Consequently, we do not believe that there are any practical limits to the scalability of the method to new data. Nevertheless, incorporating additional data into the pre-training requires a collaborative effort involving clinicians, software engineering expertise, and compliance with regulatory requirements.

Ultimately, we envision a system that is pretrained on virtually all medical tasks in clinical practice, including non-imaging label types such as language and time-series signals, allowing clinicians to rapidly experiment with custom AI solutions. A preliminary version of this system, which integrates UMedPT with a widely used open-source annotation tool capable of handling diverse data and label types, will be made available in the code repository associated with this publication.

These aspects are now discussed on page 11, lines 306-312:

"However, the performance advantage of UMedPT in in-domain tasks compared to out-of-domain tasks indicates that it is not entirely universal for all biomedical imaging applications yet, necessitating a broader scale of training.

The extent to which such pretraining should be scaled remains an open question as we did not observe any saturation with respect to the data, the variety of tasks, or the number of different label types that can be included into a single multi-task training."

Decision Letter, first revision:

Date: 9th March 24 12:29:08
Last Sent: 9th March 24 12:29:08
Triggered By: Ananya Rastogi
From: ananya.rastogi@nature.com
To: fkiessling@ukaachen.de
BCC: ananya.rastogi@nature.com
Subject: Decision on Nature Computational Science manuscript NATCOMPUTSCI-23-1256A
Message: ** Please ensure you delete the link to your author homepage in this e-mail if you wish to forward it to your co-authors. **

Dear Professor Kiessling,

Your manuscript "Overcoming Data Scarcity in Biomedical Imaging with a Foundational Multi-Task Model" has now been seen by 3 referees, whose comments are appended below. You will see that while they find your work of interest, they have raised points that need to be addressed before we can make a decision on publication.

While we ask you to address all of the points raised, the following points need to be substantially worked on:

- It has been mentioned by Reviewer #1 that several paragraphs that have been added to the manuscript in response to referees' comments seem abrupt and lack explicit motivation. Please revise these parts to ensure better integration into the manuscript.
- The method allocates a fixed computational budget for all evaluations and used the last model state which implicitly treats the test set as a validation set. This could inadvertently lead to an overestimation of the model's performance. Therefore, the strategy to mitigate potential biases should be re-evaluated.
- Please include CNN backbone results on UMedPT.
- It should be assessed whether or not UMedPT's pre-trained weights are crucial for a thorough analysis.
- Some methodological contributions have been overstated. Please address this.
- The default implementation of layer norm in PyTorch and in the original paper includes learnable bias and scaling factors. Therefore, please update the language used to ensure that readers don't confuse this as a new contribution.
- All downstream tasks are classification tasks, except for one segmentation task where the baseline is not too strong. Therefore, please include an ablation study that assesses how much the segmentation and object detection pretraining tasks actually benefit downstream performance on various task types.

Please use the following link to submit your revised manuscript and a point-by-point response to the referees' comments (which should be in a separate document to any cover letter):

[REDACTED]

** This url links to your confidential homepage and associated information about manuscripts you may have submitted or be reviewing for us. If you wish to forward this e-mail to co-authors, please delete this link to your homepage first. **

To aid in the review process, we would appreciate it if you could also provide a copy of your manuscript files that indicates your revisions by making use of Track Changes or similar mark-up tools. Please also ensure that all correspondence is marked with your Nature Computational Science reference number in the subject line.

In addition, please make sure to upload a Word Document or LaTeX version of your text, to assist us in the editorial stage.

To improve transparency in authorship, we request that all authors identified as 'corresponding author' on published papers create and link their Open Researcher and Contributor Identifier (ORCID) with their account on the Manuscript Tracking System (MTS), prior to acceptance. ORCID helps the scientific community achieve unambiguous attribution of all scholarly contributions. You can create and link your ORCID from the home page of the MTS by clicking on 'Modify my Springer Nature account'. For more information please visit please visit www.springernature.com/orcid.

We hope to receive your revised paper within three weeks. If you cannot send it within this time, please let us know.

Best regards,

Ananya Rastogi, PhD
Senior Editor
Nature Computational Science

Reviewers comments:

Reviewer #1 (Remarks to the Author):

I appreciate the authors' efforts in addressing my previous concerns and providing detailed clarifications. The modifications made in response to these concerns have notably enhanced the manuscript's quality. Nevertheless, I still have some concerns:

1. Integration of New Paragraphs: The authors' detailed responses and comprehensive experiments are commendable for their clarity, particularly given their organization around specific queries. However, I observed that several paragraphs newly added to the manuscript seem abrupt and lack explicit motivation. The necessity of these sections is not immediately clear. I recommend revising these parts to ensure better integration into the manuscript.

2. Evaluation Strategy Regarding the Validation Set: The authors state, "We allocated a fixed computational budget for all evaluations and used the last model state". This approach, however, implicitly treats the test set as a validation set. Such a methodology could inadvertently lead to an overestimation of the model's performance in a machine learning context. I suggest that the authors re-evaluate their strategy to mitigate potential biases.

3. CNN Backbone Results on UMedPT: The addition of results using a CNN backbone is noted; however, these were only conducted on MedMNIST and not on UMedPT. Given that recent works in medical imaging have found transformer-based methods not to outperform CNNs, it would be beneficial to include CNN backbone results on UMedPT as well.

4. Generalization Experiments on MedMNIST: Concerning the added generalization experiments on MedMNIST, an important setting seems to be missing. The analysis only distinguishes between MTL and ST differences without examining the impact of pre-trained weights from UMedPT. Assessing whether or not UMedPT's pre-trained weights are loaded is crucial for a thorough analysis.

Minor Points:

a. UMedPT-affine: The explanation regarding UMedPT-affine, which mentions "UMedPT used layernorms without parameters," is still somewhat confusing to me. Layer normalization typically includes parameters by default, so I am curious about the rationale behind this specific analysis by the authors.

b. Formatting of "revised_manuscript.pdf": Many tables appear to be incompletely formatted.

Reviewer #2 (Remarks to the Author):

The authors have adequately addressed my queries.

Reviewer #3 (Remarks to the Author):

Summary:

The authors propose a supervised pretraining strategy that leverages a multitude of medical datasets and tasks to reach ImageNet scale medical supervised pretraining. After they pretrain their model using their dataset, they evaluate their model on 2 in domain tasks and 5 out of domain tasks. They show that their method significantly outperforms an ImageNet pretrained model on the in domain tasks, and also outperforms the ImageNet baseline on the out of distribution tasks. Furthermore, the authors compare their model performance to external baselines, demonstrating significantly improved data efficiency.

Strengths:

- The authors aggregate a large number of tasks (15 datasets/17 tasks)
- The authors demonstrate how these heterogeneous tasks can be leveraged to enhance downstream classification performance
- The authors show how their method significantly outperforms ImageNet pretraining, as well as external baselines
- This paper is a nice demonstration that supervised pretraining is beneficial for medical imaging. As more medical imaging datasets come online with associated labels, this method could continue to improve.

Major Weaknesses:

- Overall, the major weaknesses are that the authors may overstate some methodological contributions and use language that risks leading the reader to think that they generated new contributions/insights, which were in fact developed/observed previously. The authors can successfully address these by modifying how they describe their contributions. Below are several examples:
 - Page 14, line 435: The authors claim "To address this challenge, we developed a novel training strategy for UMedPT that mostly decouples the number of training tasks from the memory requirements." The authors further state "Our strategy achieved this by establishing an independent architecture or 'computational graph' for each task. The graph is dynamically constructed and stored only during the active computation stage of each task. To combine the individual graphs, we implement gradient accumulation before the optimization step." Could the authors clarify what they mean by "Our strategy achieved this by establishing an independent architecture or 'computational graph' for each task"? It appears that the authors are using PyTorch, which would generate a single computational graph for the full model, including multiple task-specific heads. Furthermore, the language should not confuse the reader into thinking that implementing the computational graphs is a part of the author's contribution, when this is how Pytorch operates under the hood.
 - The authors further state, "We ensure that the model's weights and gradients are stored only once, rather than duplicating them for each task. Additionally, only the activations for one task are kept in memory at a time." The language may be a bit strong and overclaim contributions here. Gradient accumulation as implemented in the code below is used routinely, with all handling of the computational graph by PyTorch. GA implementation requires simply not calling `loss.backward()` at every step in PyTorch. The contribution here is sampling all tasks within a global GA step. I would recommend that the authors soften their language in this section, and make more clear what they contributed versus previous methods implemented by others that they are explaining for the education of the reader.

```
for i, (batch, task) in enumerate(task_iterator):
    batch_extraction_ms = time.perf_counter() - iteration_start_time

    # is_last_step = i >= self.__len__() - 1
```

```

is_update_step = (
self.has_len() and (i >= self.__len__() - 1)
) or (
i % self.accumulate_losses_for_steps == 0
)
is_update_step = self.training_mode and is_update_step and (i > 0)

assert task.training == self.training_mode
# Set active task for all shared blocks
for block in self.shared_blocks.module.shared_modules.values(): # type: ignore
block: SharedBlock
block.set_active_task(task.get_name())

if self.training_mode and is_update_step and (self.sync_on is not None):
# the forward step needs to know if a sync will happen, so turn on sync here already
self.sync_on()
# with Join([self.shared_blocks], enable=False):
try:
task_step_result = self.task_step(batch, task)

```

- Layer norm is typically used in vision transformers, including in the SwinTransformer architecture. Therefore, the authors should make sure that their language does not risk confusing the author into thinking that this is a new finding/contribution. As the default SwinTransformer uses layernorm, the following language should be softened - "To address this problem, we recursively replaced the original normalization layers in all shared blocks with layer normalization, which by design do not require inter-task computation".

- The authors state - "We empirically analysed the effect of using layer norms with affine parameters on our approach using an adaptation of UMedPT(UMedPT-affine). "..."UMedPT-affine added trainable parameters including a bias and a scaling factor γ in the form $y = \gamma x - \mu \sigma + \beta$ for each channel." The default implementation of layernorm in PyTorch and in the original paper includes learnable bias and scaling factors. Therefore, I would update the language used to ensure that readers don't confuse this as a new contribution.

Minor Weaknesses:

- Page 9, line 226 "A comparison for the OrganSeg-MRI task could not be performed, because no results specific to the MRI-only subtask of the challenge were reported". I would request that the authors train a baseline nnUNet or other state of the art baseline for comparison. Otherwise, it is difficult to understand the segmentation performance on this dataset.

- If I am not mistaken, all downstream tasks are classification tasks, except for one segmentation task where the baseline is not too strong. Therefore, something that would really strengthen this work, perhaps as future work, is an ablation study that assesses how much the segmentation and object detection pretraining tasks actually benefit downstream performance on various task types. The latents in the encoder decoder architecture trained for segmentation or object detection need to retain

geometrical information but may not require pixel intensity information. On the other hand, performing well on downstream classification tasks does not require encoding precise geometrical information. It could in fact be the case that these are at odds and requiring capacity for precise geometric information reduces embedding quality for classification tasks. Investigating which types of tasks should actually be included during pretraining would be a nice contribution that would put this paper into better context.

- I understand that this study has been for the most part limited to supervised pretraining. However, this method will need to compete with self-supervised pretraining which could potentially scale more easily. A comparison to such methods, like MAE and DINO (i.e. RAD-DINO), would be a nice to have.

- Page 14, line 439: Can the authors clarify what this means: "This graph is dynamically constructed and stored only during the active computation stage of each task"?

- Page 17, line 527 - What does "recursively replaced" mean here? The norm layers should not be nested so what does recursing mean here?

- Page 18, line 570: could the authors make it more clear what "the need for pre-extraction of images" means?

- Page 21, line 622 - if "flips and mirroring" are applied as augmentations, the network could lose the ability to differentiate left sided vs right sided diseases, which is an area of study for medical foundation models. Can the authors justify the inclusion of these augmentations?

- Page 21, line 614 - what are the "standard 3D augmentations"? Can the authors include those in the paper?

- Page 23, line 712 - The authors should make it clearer that the 1%, 5%, 10%, 25%, 50%, and 100% corresponds to the downstream datasets, not the pretraining dataset, if that is in fact the case. The authors should clarify whether there is any overlap in the downstream datasets and the pretraining datasets. It seems that CRC-WSI may be present in pretraining (Extended Table 4) and was also used for downstream validation in the comparison with ImageNet. If this is the case, it may not be fair to claim that finetuning with 1% of downstream data compared to ImageNet pretraining, as the training dataset was seen during pretraining. Could the authors clarify whether this is a typo in Extended Data Table 4?

- Page 23, line 705 - The authors state "In the frozen scenario, we directly extracted image representations from the shared blocks, thereby showing the usefulness of the learned representations. Both frozen and fine-tuning were trained for epochs each." Does training in the frozen case mean training a linear probe for classification tasks? If so, I would use this common terminology.

- Page 23, line 723 - What does "re-discovery" and "re-identification" mean here? Would like to clarify that this does not mean that downstream datasets were used during pretraining. If a downstream dataset was used during pretraining, it does not

seem fair to claim that fine tuning on 1% of downstream data yielded similar performance to ImageNet baseline.

- Extended Data Table 3: Which version of the model are you using for your results in the main paper? Can you add this to the caption?

- The authors should add more details about how inference is done with 3D data. Are predictions averaged across slices? Is the following method applied to the origin UMedPT model as well? "For the 3D tasks, we used a simple strategy based on a learned weighted average across slices with the classification task described in section."

- Could the authors add a bit more explanation for their choices in normalization factors in equations (1) and (2)?

Point of Discussion:

Without validation sets, it may be difficult to understand whether catastrophic forgetting is the main culprit for decreasing performance with increasing dataset size or if the authors are overfitting to the downstream task, with the number of optimization steps increasing along with the fraction of the downstream dataset used for fine-tuning. Inevitably, some forgetting is happening when adapting to a specific downstream dataset. However, a somewhat related issue could be overfitting with 100 epochs of fine-tuning. I understand the challenge here where the authors want to be able to make the claim that truly 1% of the dataset was used for training, vs a larger fraction of training dataset size + validation dataset size. It may be necessary to have a larger validation set to get a clear signal about model performance. I commend the authors for truly using 1% of data for training, versus using a small training dataset but then a much larger validation dataset. A nice to have ablation to include in this work or future work would be investigating performance if you use full validation sets and modulate only the training dataset size. If using a validation set actually causes model performance to increase with training dataset size, then you can be confident that the performance decrease is only coming from suboptimal checkpoint selection. This would add additional support for the efficacy of the method.

Grammatical/Syntactical Errors:

- Extended Data Table 4: the authors should specify in the caption what "/" means. In, for example, the third column of the CRC-WSI row, where 100,000/7,000 is written. Is this train/test data?

- Fig. 2 - In the caption: is Tuber-CXR the same dataset as Pneumo-CXR in the plot titles? BC-Bach-WSI referenced in caption, as opposed to CRC-WSI.

- Page 1, line 29 - maybe consider updating "required not more than 50%" to "required only 50% of the original training data".

- Page 1, line 19 - I'm not sure that I would consider medical dataset to be more heterogenous than natural domain datasets that can comprise any scene/object.

Medical images generally look similar globally with differences coming from finer grain features.

- Would the authors mind justifying this description or revising it?
- Page 2, line 48 - for clarity, I would consider updating "increasingly large pretrainings" to "increasingly large pretraining datasets"
- Page 2, line 53 - May want to add LAION, in addition to ImageNet.
- Page 9, Table 1. Should there be a citation for CNS-MRI?
- Page 23, line 696 - what does "synthetic dataset" mean here? Usually it refers to data which isn't real, but generated.
- Page 23, line 708 - What does subsequently refer to here? "After frozen and fine-tuning for 100 epochs, subsequently fine-tuning stage enabled training of shared blocks". Is the order implied by the word subsequently significant here?
- Extended Data Table 5 extends beyond the page width
- Would encourage the authors to remove commented code from their codebase and also add comments within most functions/classes that describe their purpose, along with descriptions of arguments, their types, and any outputs.

Reviewer #3 (Remarks on code availability):

Overall the code seems modular and it seems as though lots of effort was put into making the code robust.

A few suggestions are:

- There are currently two subfolders in the top level directory and no readme. I would put a readme in the top level directory so that users know what the two subfolders are for. This readme should describe the code within each subfolder on a high level. It should tell the user why code is split into two subfolders and what is different about each code base.
- I would change the name of the "code" subdirectory to be more descriptive.
- In the readmes within each subdirectory, I would include a description of the organization of the code. What are each of the "neural", "optimization", "trainer", "logging", "interactive", and "data_loading" folders for?
- Also within these readmes, I would use code blocks to demonstrate to the user how to run the code, as opposed to "use universal_pretraining.py".
- If possible, I would add links to all datasets that the user needs to download to the readme. This would make it significantly easier for others to collect the datasets and reproduce the results in this paper.

- The specific wheels in the requirements.txt file are not supported by certain systems. Instead of including the wheel links, I would instead specify versions. I had to remove these to install the code.
- Would encourage the authors to remove commented code from their codebase.
- Would encourage the authors to add comments within most functions/classes that describe their purpose, along with descriptions of arguments, their types, and any outputs.

Author Rebuttal, first revision:

Dear Editor and Reviewers,

Thank you for your time, valuable comments and contributions to improve our work. We have addressed all your concerns and revised the manuscript accordingly. Our direct replies are shown in this document, and the actual changes in the manuscript are highlighted in green.

We have also uploaded additional material at this link: Additional Review Materials. The uploaded material includes a confidentially attached follow-up paper that we included for its comparison between CNN and Swin Transformer, the network weights for the revised version of the current manuscript, and a description of the files and how to load the weights.

Editor: It has been mentioned by Reviewer #1 that several paragraphs that have been added to the manuscript in response to referees' comments seem abrupt and lack explicit motivation. Please revise these parts to ensure better integration into the manuscript.

Reply: We revised the manuscript to improve the text flow. For better continuity, we integrated the MedMNIST experiments into the introductory paragraph of the results section (page 5, lines 125-), and moved the complementary experiments to the supplement:

"We evaluated our models in three benchmarks. The first, the 'in-domain benchmark', aimed to determine UMedPT's performance on tasks closely related to its pretraining database. The second, the 'out-of-domain benchmark', aimed to assess how well UMedPT adapted to new tasks outside its immediate training domain. The third, the MedMNIST benchmark [16], was used to evaluate the proposed multi-task training strategy on a separate training database and, independently, to test UMedPT."

The UMedPT ablation studies have also grown with the new label type ablation study. For this reason, we have provided better integration by moving the detailed description to the supplement and just embedded a short description and the key results into the main text on page 5, lines 141-152:

"For our in-domain and out-of-domain clinical benchmarks, we conducted ablation studies for UMedPT to investigate the effects of the variable input image size of UMedPT compared to the fixed input image size of 224 × 224 with UMedPT-fixed, and whether to include trainable parameters in the LayerNormalizations within its architecture with UMedPT-affine, which are detailed in Extended Data Section S1. We found that a variable input size was beneficial for the performance of UMedPT, while UMedPT-affine had a minor impact on the results. In addition, we compared the performance of UMedPT with a variant that was trained only with the classification tasks UMedPT-clf, as described in Extended Data Section S2. This showed a great benefit of including segmentation and object detection tasks, especially for other similar tasks."

As discussed below in this letter, we applied UMedPT to MedMNIST to investigate the model's generalizability. The obtained results have been integrated into Table 1 (cropped in this letter).

Table 1 Amount of data required by UMedPT to match state-of-the-art performance on classification tasks from different imaging domains. Datasets marked with an asterisk (*) were compared across different test splits.

Task	Reference results	UMedPT	
		Frozen	Fine-Tuning
MedMNIST mean AUC	See Ext. Data Figure 3 [16]	100%	10%
MedMNIST mean ACC	See Ext. Data Figure 3 [16]	-	10%

The outcome of the comparison with external reference results is presented in page 9, lines 246-248:

“[...] UMedPT surpassed the external reference results in [...] and the average AUC in the MedMNIST database [16].”

While we have moved the results of the architectural comparison, the training schemes, and from the catastrophic forgetting investigation to the supplement, we have chosen to keep the MedMNIST 3D contextual preservation experiment in the main body of the manuscript and added context to keep the integration seamless (page 10, lines 266-292):

“2D multi-task learning outperforms single-task 3D CNNs

Independently of our training database, we evaluated the training strategy on MedMNIST. MedMNIST contains a variety of standardized and downscaled biomedical image datasets, including both 2D and 3D images. To assess the impact of transforming inherently 3D medical image data into 2D slices for pretraining, we evaluated single-task learning with 2D data, single-task learning with original 3D data [15] and multi-task learning with 2D data. For our multi-task and single-task trainings, we converted the 3D data into 2D slices by slicing through the last dimension and applied a multiple-instance learning classification task that was based on a weighted averaging operation across slices. When only the 3D tasks were considered, the average accuracies were:

- 83.22 ± 1.61% for single-task learning pretrained using Imagenet;
- 83.76% for the single-task MedMNIST 3D CNN [15];
- 86.46 ± 1.13% for multi-task learning.

The results showed that single-task 3D CNNs performed better than single-task 2D CNNs pretrained using Imagenet. However, multi-task 2D networks outperformed the single-task 3D CNNs. We present the details in Extended Data Table 5.

In this context, we also investigated the performance difference between CNN and Transformer architectures for our multi-task learning strategy, as presented in Extended Data Section S3 and aspects of the training algorithm in Extended Data Section S4. We found that the Swin Transformer architecture has a minimal positive impact, as shown in Extended Data Figure 4a. Regarding training schemes, we found that without gradient accumulation, both convergence and performance were worse with our training strategy, as shown in Extended Data Figure 4b.”

As the sections regarding architecture and training schemes were moved to the supplement, we improved their coherent organization around specific queries:

“S3 Comparing convolutional networks and transformer

For quantifying the effect of the encoder’s architecture, we used the MedMNIST database including two-dimensional and three-dimensional classification tasks. We chose the ResNet-50 as convolutional neural network (CNN) and the tiny variant of the Swin Transformer because they are similar in size. The Swin Transformer has 27,582,570 trainable parameters compared to 23,508,032 for the ResNet-50 CNN.

The comparison of the Swin Transformer and ResNet-50 CNN architectures showed a minimal impact on model performance for the MedMNIST database. The Swin Transformer achieved an average test accuracy of 86.76 ± 0.79% over 5 repetitions, while the ResNet-50 CNN achieved an accuracy of 86.34 ± 1.01%. In addition, a discrepancy in training

convergence rates was observed between the two architectures, as shown in Extended Data Figure 4a.

S4 Investigating training schemes

We performed an analysis of training schemes on the MedMNIST database, including the two- and three-dimensional tasks. All datasets in MedMNIST have a fixed number of cases. This distinction enabled us to conduct ablation studies comparing infinite task sampling with balanced sampling based on dataset size. Besides this, we used the same training schedule and hyperparameters as in the main study, and accumulated the gradients of as many steps as there were tasks. In addition, for comparison with traditional training schemes, we used the same setting without gradient

accumulation and also with the SGD optimizer instead of Adam. The exploration of training schemes showed that balanced (by dataset size) and cyclic sampling (as in UMedPT) exhibited similar behaviour in terms of convergence. However, balanced sampling occasionally showed reduced stability; it yielded a standard deviation of $1.81 \pm 1.79\%$ in validation accuracy over the previous ten epochs, across five different experiments. In comparison, cyclic sampling showed a more stable training process, achieving a comparatively lower standard deviation of $1.17 \pm 1.09\%$. When gradient accumulation was excluded, the resulting performance deteriorated, accompanied by longer convergence times. These results are shown in Extended Data Figure 4b.”

Editor: The method allocates a fixed computational budget for all evaluations and used the last model state which implicitly treats the test set as a validation set. This could inadvertently lead to an overestimation of the model's performance. Therefore, the strategy to mitigate potential biases should be re-evaluated.

Reply: We appreciate the concern about the potential overestimation of our model's performance. In response, we would like to clarify that no test data were used for validation in our study. Consequently, the risk associated with our method, whenever we have not used a validation set, is an underestimation of model performance rather than an overestimation.

However, it is still of great importance to examine the best way to apply pretrained foundational models to tiny and large target datasets. For this reason, we appreciate the feedback and conducted the experiment suggested by reviewer 3, and integrated it with the previous experiments regarding this topic on page 50, lines 1327-1351:

“Inverse relationship between performance and dataset size

Our evaluation within the clinical benchmark revealed an unexpected trend in some datasets: increasing the dataset size for fine-tuning sometimes led to a decrease in model performance.

To investigate the potential influence of catastrophic forgetting [37] or overfitting during fine-tuning, we first evaluated this phenomenon using four MedMNIST tasks that had shown improved performance with multi-task learning compared to single-task learning. We first measured the test accuracy of these tasks after multi-task learning, followed by further individualised training with the full dataset of each task, and assessed the test accuracy again. The results varied between datasets, suggesting that whether datasets are affected by forgetting the well generalizing state from multi-task learning is inconsistent and may be task-dependent:

- SynapseMNIST3D: $83.81 \pm 0.31\% \rightarrow 82.90 \pm 0.66\%$ (decrease)
- VesselMNIST3D: $93.66 \pm 0.31\% \rightarrow 93.77 \pm 0.87\%$ (no decrease)
- BreastMNIST: $86.92 \pm 1.04\% \rightarrow 85.90 \pm 0.91\%$ (decrease)
- PneumoniaMNIST: $91.54 \pm 0.48\% \rightarrow 91.70 \pm 0.49\%$ (no decrease)

For our in-domain and out-of-domain target tasks, we always used 100 epochs. Consequently, larger datasets used more optimization steps and could overfit more easily. We investigated by keeping large validation sets (30% of the full training data) in one in-domain and one out-of-domain task where the phenomenon occurred and performed model selection using the validation set. Extended Data Figure 5 shows that for one task the model selection with the validation set was better, for the other task it was worse.”

a

b

Extended Data Fig. 5 Model selection with and without validation sets. For the target tasks in our clinical benchmark, we did not use validation sets to really only use the given percentage of training data (UMedPT). This could lead to overfitting on the training data, which is usually solved by using a validation set, as done with UMedPT-Val. We investigated this using a representative out-of-domain data set, Tuber-CXR (a), and an in-domain target task, Pneumo-CXR (b).

We reference the key findings in the main text on page 6, lines 175-179:

“We further investigated whether this could be due to catastrophic forgetting of the well generalizing pretrained features or overfitting to the training data and found that the phenomenon is dataset specific, as detailed in Extended Data Section S5.”

Additionally, when applying UMedPT to MedMNIST we followed the standard protocol that was set out by the authors of the MedMNIST database. As opposed to the original experiments with UMedPT, these new results included the use of a validation set. The results of the model selection with validation set, together with the selection of the final model, are presented in the new Extended Data Figure 3 of our paper. This figure shows that the model selected by the top F1 score on the validation set performs slightly better than the final model, but at the cost of acquiring additional validation data:

Extended Data Fig. 3 UMedPT’s application to MedMNIST. First, UMedPT was applied to MedMNIST [15] with the shared encoder frozen and a randomly initialized linear head (linear probing) and evaluated on the test set using area under the curve (AUC, left) and accuracy (ACC, right). The whole model was then fine-tuned independently for each task. Blue and green lines represent the test performances when the model was selected using the validation set provided by the authors. Red and orange lines represent the test performance when the last model state was selected (validation data not used). Horizontal lines represent the theoretically best performance when the best reference method is selected for each task and metric independently (red) or when the best method is selected for all tasks (grey). We evaluated UMedPT with 1%, 10% and 100% of the training data. Details are given in Extended Data Table 6.

From these four findings, we see that there is no clear winning training strategy across all target tasks. The need for further in-depth evaluation is now motivated in the discussion section on page 12, lines 356-372:

“In some cases we observed that the performance of UMedPT decreased as the size of the training dataset increased. We investigated both, catastrophic forgetting [35] of the well generalizing features learned during pretraining and overfitting to the training set due to using all data for training instead of a validation set for model selection in Extended Data Section S6. The inconsistency of the results raises questions about the best practices for

using foundational models in tasks with varying data sizes and varying degrees of similarity to the pretraining database. There were tasks that performed best with model selection using a validation set, and tasks that performed best with all the data used for training. Similarly, some tasks performed best with the frozen training setting, and others with fine-tuning of all pretrained parameters of UMedPT. Recently, more sophisticated fine-tuning strategies have been proposed for foundational models in natural language processing, such as BitFit [36], where only the bias-terms are fine-tuned, or LoRA [37], where a small number of additional parameters are fine-tuned. A training configuration targeted specifically to foundational vision models could combine the strengths of the different training configurations."

Editor: Please include CNN backbone results on UMedPT.

Reply: We did not include CNN backbone results on UMedPT because we already included CNN backbone results with MedMNIST and the inclusion for UMedPT would be costly and not provide substantial new insights due to the following reasons:

- In the context of follow-up work on a specialized pretraining for histology, we used the presented training method with full-scale data within a modern CNN known as "ConvNeXt". Preliminary results show a negligible difference between the two architectures, with a small preference for the Swin Transformer. This will be reported as part of the confidentially attached follow-up paper.
- Furthermore, in the revised version of the current manuscript, we included the MedMNIST database for pretraining and evaluation separately from UMedPT. These MedMNIST pretrainings, which are already included in this study, used a traditional CNN. The performance of the two architectures was similar, although the Swinformer backbone had a slight advantage. This finding is consistent with the results of our follow-up work on specialized pretraining for histology.

In summary, our study and preliminary results from the follow-up work show a negligible difference between the results of the two architectures. We have elaborated on this point in the discussion (p. 13, lines 399-407):

"In addition to the ability to handle arbitrary image sizes, for the UMedPT encoder we needed a general base architecture capable of generating multi-scale feature maps, a feature found in both convolutional neural networks (CNNs) and swin transformers [37]. Our experiments with MedMNIST showed a minimal difference between CNN and Swin Transformer, slightly in favour of the latter. This suggests that the proposed pretraining strategy can be implemented with both convolutional and transformer-based encoders, with literature showing that CNNs can also work well with large datasets of full-size images [38]."

And we improved the integration of the CNN backbone results on MedMNIST within the methods section on page 17, lines 524-529:

"UMedPT's decoders are compatible with any encoder that can generate multi-scale feature maps [...] We also investigated the compatibility of the CNNs with the proposed multitask training loop and included an additional comparison in Extended Data Section 54."

Editor: It should be assessed whether or not UMedPT's pre-trained weights are crucial for a thorough analysis.

Reply: We assessed whether UMedPT's pre-trained weights are crucial by comparing the pretrained weights of the ImageNet pretrained weights with the pretrained weights of UMedPT on an in-domain benchmark, an out-of-domain benchmark, the separate MedMNIST database and an external evaluation.

The main contribution of our work is the multi-task pretraining strategy. While two independent multi-task databases and evaluations ensured that the pretraining strategy was not database specific, reviewer 1 correctly pointed out that an interesting setting was missing for the MedMNIST database: UMedPT pretrained weights.

We added a description of this setting to the Methods section “Comparison of benchmark results” (page 26, lines 855-859):

“We compared our results with the best previously reported study results for the target tasks and the mean performance for the MedMNIST database [15]. From the MedMNIST database, we only considered tasks that were available in the largest spatial size (224 × 224) and were not part of the UMedPT pretraining or clinical benchmark.”

Corresponding results are also reported in Table 1 (cropped):

Table 1 Amount of data required by UMedPT to match state-of-the-art performance on classification tasks from different imaging domains. Datasets marked with an asterisk (*) were compared across different test splits.

Task	Reference results	UMedPT	
		Frozen	Fine-Tuning
MedMNIST mean AUC	See Ext. Data Figure 3 [16]	100%	10%
MedMNIST mean ACC	See Ext. Data Figure 3 [16]	-	10%

The results are presented in Extended Data Figure 2:

Extended Data Fig. 2 UMedPT’s application to MedMNIST. First, UMedPT was applied to MedMNIST [15] with the shared encoder frozen and a randomly initialized linear head (linear probing) and evaluated on the test set using area under the curve (AUC, left) and accuracy (ACC, right). The whole model was then fine-tuned independently for each task. Blue and green lines represent the test performances when the model was selected using the validation set provided by the authors. Red and orange lines represent the test performance when the last model state was selected (validation data not used). Horizontal lines represent the theoretically best performance when the best reference method is selected for each task and metric independently (red) or when the best method is selected for all tasks (grey). We evaluated UMedPT with 1%, 10% and 100% of the training data. Details are given in Extended Data Table 6.

Further results are reported in Extended Data Table 6 (rotated for readability in the paper):

Extended Data Table 6 Detailed Performance of UMedPT on MedMNIST. Comparison of accuracy (ACC) and area under the curve (AUC) at different stages of training UMedPT on the MedMNIST database: using the frozen encoder and fine-tuning the whole model. Performance metrics are provided for UMedPT with 1%, 10% and 100% of the training data. Reference results are included from ResNet-50 (Ref. CNN) or the theoretical best results obtained by selecting the method with the strongest test performance for each dataset and metric independently (Ref. Cherrypick).

dataset	stage metric fraction	UMedPT - Frozen		UMedPT - Finetune		Ref. Cherrypick		Ref. CNN	
		ACC	AUC	ACC	AUC	ACC	AUC	ACC	AUC
bloodmnist	1%	82.81±1.10%	98.88±0.12%	96.85±0.48%	99.88±0.01%	-	-	-	-
	10%	96.11±0.16%	99.82±0.00%	98.02±0.08%	99.95±0.00%	-	-	-	-
	100%	98.20±0.03%	99.91±0.00%	99.14±0.08%	99.95±0.00%	96.60%	90.80%	95.00%	99.70%
breastmnist	1%	60.08±14.84%	63.07±3.45%	59.62±15.97%	64.89±2.53%	-	-	-	-
	10%	73.72±1.38%	76.95±2.15%	83.76±1.60%	88.28±1.70%	-	-	-	-
	100%	82.91±0.30%	90.99±0.31%	94.02±0.60%	94.41±0.33%	86.30%	91.90%	84.20%	86.60%
chestmnist	1%	94.77±0.03%	72.55±0.60%	94.76±0.06%	74.86±0.88%	-	-	-	-
	10%	94.82±0.01%	78.37±0.15%	94.81±0.01%	80.53±0.09%	-	-	-	-
	100%	94.84±0.00%	80.24±0.01%	94.90±0.01%	84.11±0.04%	94.80%	77.80%	94.80%	77.30%
dermamnist	1%	66.92±0.06%	71.68±2.38%	68.96±0.55%	79.58±2.18%	-	-	-	-
	10%	71.49±0.53%	85.01±1.63%	78.42±0.27%	92.20±0.17%	-	-	-	-
	100%	78.22±0.08%	93.30±0.01%	91.21±0.23%	98.64±0.09%	76.80%	92.00%	73.10%	91.20%
octmnist	1%	69.70±3.31%	97.15±0.08%	84.97±2.19%	98.85±0.14%	-	-	-	-
	10%	76.33±0.61%	97.78±0.09%	88.37±0.12%	98.72±0.24%	-	-	-	-
	100%	77.53±0.39%	97.84±0.02%	92.00±0.83%	99.50±0.16%	77.60%	96.30%	77.60%	95.80%
organamnist	1%	70.95±1.53%	96.52±0.19%	86.69±1.21%	98.93±0.07%	-	-	-	-
	10%	83.28±0.21%	98.49±0.01%	95.16±0.18%	99.80±0.02%	-	-	-	-
	100%	87.01±0.10%	99.04±0.01%	96.86±0.18%	99.89±0.01%	95.10%	99.80%	94.70%	99.80%
organemnist	1%	38.53±1.46%	88.87±0.73%	70.16±2.19%	95.72±0.45%	-	-	-	-
	10%	72.25±0.27%	96.58±0.04%	89.82±0.36%	99.36±0.05%	-	-	-	-
	100%	81.67±0.12%	98.31±0.00%	95.24±0.13%	99.86±0.01%	92.00%	99.40%	91.10%	99.30%
organismnist	1%	47.20±3.01%	88.40±0.89%	64.33±0.83%	92.99±0.32%	-	-	-	-
	10%	70.32±0.16%	95.59±0.16%	80.83±1.27%	97.88±0.13%	-	-	-	-
	100%	77.08±0.13%	97.16±0.02%	84.89±0.08%	98.54±0.06%	81.30%	97.50%	78.50%	97.50%
retinamnist	1%	42.50±0.35%	58.07±5.78%	42.42±0.06%	60.64±4.02%	-	-	-	-
	10%	47.08±1.85%	71.89±4.02%	57.75±3.89%	81.63±2.89%	-	-	-	-
	100%	57.00±0.41%	83.96±0.08%	65.83±0.92%	88.57±0.24%	53.10%	75.00%	51.10%	71.60%
tissuemnist	1%	54.86±0.22%	86.12±0.10%	59.52±0.30%	88.80±0.04%	-	-	-	-
	10%	59.32±0.11%	88.74±0.02%	68.06±0.16%	92.78±0.02%	-	-	-	-
	100%	60.21±0.09%	89.28±0.00%	76.63±0.14%	96.06±0.01%	70.30%	94.10%	68.00%	93.20%

Editor: Some methodological contributions have been overstated. Please address this.

Reply: We clarified that we used PyTorch and implementing computational graphs is not part of our contribution (page 15, lines 450-451):

“We used PyTorch [39] to create an independent architecture, or ‘computational graph’, for each task. [...]”

In this way we clarified that by “an independent architecture or ‘computational graph’ for each task” we meant the PyTorch computational graph, and made the implication for our training strategy clear (page 15, lines 457-461):

“GA allowed [...] a single update step could consist of heterogeneous tasks in any order. This allowed the training strategy to use an adaptive architecture, where each type of label can be solved by a specialized combination of model components, such as a UNet for segmentation labels [45].”

Regarding our application of gradient accumulation, in our view, the simplicity of this part of the multi-task training strategy is an advantage. We softened the language, clarified that we did not propose gradient accumulation, and made it clearer that, unlike previous methods, we have implemented a way to appropriately apply gradient accumulation to multitask learning. (page 15, lines 462-465):

“GA is a common method for incorporating more data into a single optimization step. In the case of our multi-task learning strategy, unlike traditional deep multi-task learning, GA allowed the weights and gradients of the shared part of the model to be stored only once, rather than duplicated for each task. [...]”

Editor: The default implementation of layer norm in PyTorch and in the original paper includes learnable bias and scaling factors. Therefore, please update the language used to ensure that readers don’t confuse this as a new contribution.

Reply: We updated the language to make clear that using Layernorm including learnable bias and scaling factors within Swin Transformer is not a new scientific methodology developed in this work (page 17, lines 548-549):

“Notably, the Swin Transformer encoder used in UMedPT already used layer normalization, which comes with trainable parameters by default.”

And later on page 18, lines 557-560:

“To empirically assess the impact of excluding such trainable parameters in UMedPT, we compared it to a variant of our model UMedPT-affine that included trainable bias and scaling layer normalization parameters, which is the default for the Swin Transformer, the UMedPT encoder. [...]”

Editor: All downstream tasks are classification tasks, except for one segmentation task where the baseline is not too strong. Therefore, please include an ablation study that assesses how much the segmentation and object detection pretraining tasks actually benefit downstream performance on various task types.

Reply: We added another segmentation task with coloscopy data, consisting of data that did not occur within the UMedPT’s pretraining database and is more closely related to natural images on which the ImageNet baseline was trained with.

We added the out-of-domain task Polyp-RGB, which was used to train a model for polyp segmentation in coloscopy data. It was included in the methods section (page. 26, lines 845-853):

“Polyp segmentation in coloscopy (PolypSeg-RGB):

The PolypSeg-RGB task [25] focused on segmenting polyps from the background in coloscopy images. Since polyps can be precursors to colorectal cancer, coloscopy is an important diagnostic tool. Early detection and removal of polyps is essential to prevent the development of colorectal cancer. However, the effectiveness of coloscopy is often hampered by high miss rates; studies have found that polyp miss rates during coloscopy can range from 14 to 30%, depending on the type of polyp [54]. We randomly divided the dataset into 700 training images and 300 test images.”

Consistent with the other results, we find further evidence that pretraining with UMedPT does not impair learning, even in cases where UMedPT was not pretrained for: (page 9, lines 233-243):

“Polyp segmentation in coloscopy (PolypSeg-RGB):

The PolypSeg-RGB target task focused on the segmentation of polyps in coloscopy images. When using the entire dataset for fine-tuning, ImageNet achieved its best average result, demonstrating a mean Intersection over Union (mIoU) of 0.905. Here, UMedPT achieved an mIoU of 0.911. The ImageNet pretrained model showed better results when the encoder was frozen, as presented in Extended Data Figure 1c. The best performance across all fractions was achieved by UMedPT with fine-tuning. In addition, while UMedPT with fine-tuning outperformed ImageNet for all fractions, the biggest difference occurred with 1% of the data (0.797 ± 0.09 compared to 0.683 ± 0.144 of ImageNet).”

The details are presented within the out-of-distribution plots in the supplement (cropped here):

Extended Data Fig. 1 Results of remaining out-of-distribution tasks. a BC-BHIS-MIC b CNS-MRI. c PolypSeg-RGB.

The strong results of both ImageNet and UMedPT are further evidenced by the fact that they outperform i) the original baseline provided by the dataset authors, and ii) in the case of UMedPT, even recent approaches developed specifically for polyp segmentation. This is now shown in our comparison with external reference results (shown here cropped):

Table 1 Amount of data required by UMedPT to match state-of-the-art performance on classification tasks from different imaging domains. Datasets marked with an asterisk (*) were compared across different test splits.

Task	Reference results	UMedPT	
		Frozen	Fine-Tuning
PolypSeg-RGB	0.778 mIoU [24]	50%	1%
PolypSeg-RGB	0.9051 mIoU [25]	-	100%

Additionally, we added a detection task for nuclei counting. While it used tissue types that were not part of the pretraining database, we still classified it as in-domain due to its similarity to one of the histological pretraining detection tasks.

In the methods section (page 25, lines 780-788):

“Detection of nuclei in whole slide images (NucleiDet-WSI):

In oncology, the distribution and appearance of nuclei are important for the diagnosis and study of cancer. To assess the ability of UMedPT to detect these nuclei, the NucleiDet WSI dataset [52] was used. This dataset consists of whole slide images (WSI) and covers ten cancer types. In the pretraining database, only prostate and colon cancer were included. We randomly divided the dataset into 950 images for training and 406 images for testing. The authors of the dataset created the annotations with the help of two pathologists and three graduate students, using an AI tool.”

The results show that only 50% of the training data is required to outperform ImageNet (page 6, lines 187-196):

“Detection of nuclei in whole slide images (NucleiDet-WSI):

We used the NucleiDet-WSI dataset [18] to detect nuclei in 10 different cancer types from whole slide images (WSI). The best ImageNet performance was achieved using 100% of the data together with fine-tuning, resulting in a mean average precision (mAP) of 0.71. UMedPT was able to replicate this performance with 50% of the training data and no fine-tuning. However, fine-tuning tended to improve the results for both models. Interestingly, compared to ImageNet, UMedPT showed superior performance across all data fractions with both fine-tuning and a frozen pre-trained model. This resulted in a maximum performance of 0.792 mAP when using the full training data set and fine-tuning."

We present the results as part of Figure 2:

Fig. 2 Results for in-domain tasks. **a** In diagnosing pneumonia (Pneumo-CXR), UMedPT matched the full fine-tuned performance of ImageNet, even with a frozen encoder and a reduced dataset size (1%). **b** For CRC-WSI, the only target task for which its training dataset was also part of the pretraining, performance was stable with a frozen encoder. When the encoder was fine-tuned, performance decreased to the result obtained with ImageNet pretraining. **c** For NucleiDet-WSI, an object detection task for counting nuclei in whole slide images, UMedPT outperformed ImageNet across all training settings. Best performance was achieved with 100% of the training data and fine-tuning.

Now that we have added a segmentation task where the baseline is strong and another detection task, we can evaluate a new ablation that excluded segmentation and detection tasks from its pretraining "UMedPT-clf". We introduced this in the Methods section, integrated into the section about the pretraining tasks (page 21, lines 622-627):

"To further understand the importance of pretraining diversity, we conducted an ablation study focusing only on classification tasks. We trained an ablation UMedPT-clf using only the classification pretraining tasks. We evaluated UMedPT-clf on one representative task from classification (Pneumo-CXR), segmentation (PolypSeg-RGB) and object detection (NucleiDet-WSI) and compared it to the full model UMedPT."

We integrated the key finding alongside the results of the other pretraining ablation studies (page 5, lines 148-152):

"In addition, we compared the performance of UMedPT with a variant that was trained only with the classification tasks UMedPT-clf, as described in Extended Data Section S2. This

showed a great benefit of including segmentation and object detection tasks, especially for other similar tasks.”

The detailed results of the label type ablation were added to the supplement:

“S2 Benefit of segmentation and object detection in pretraining

To quantify the effect of including multiple label types in the pretraining, we compared UMedPT with a model trained on our classification pretraining tasks only, which we call UMedPT-clf. The results are shown in Extended Data Figure 2. There is a large average difference and consistently better performance of UMedPT for tasks requiring high spatial resolution features. For the object detection task NucleiDet-WSI, UMedPT achieved a 0.282 higher mean Average Precision (mAP), and for the segmentation task Coloscopy-RGB, it outperformed UMedPT-clf by 0.057 mIoU. Interestingly, although the difference was smaller for Pneumo-CXR (classification), a clear positive knowledge transfer between the label types was found, with an advantage of 2.42% F1-score in favour of UMedPT.”

Extended Data Fig. 2 Results of label type ablation study with UMedPT-clf. UMedPT-clf was trained with the same classification tasks as UMedPT, but excluded segmentation and object detection tasks. **a** Pneumo-CXR (classification). **b** NucleiDet-WSI (object detection). **c** PolypSeg-RGB (segmentation).

Reviewer #1

Reviewer: I appreciate the authors' efforts in addressing my previous concerns and providing detailed clarifications. The modifications made in response to these concerns have notably enhanced the manuscript's quality. Nevertheless, I still have some concerns:

1. Integration of New Paragraphs: The authors' detailed responses and comprehensive experiments are commendable for their clarity, particularly given their organization around specific queries. However, I observed that several paragraphs newly added to the manuscript seem abrupt and lack explicit motivation. The necessity of these sections is not immediately clear. I recommend revising these parts to ensure better integration into the manuscript.

Reply: Thank you for the kind words. We revised the manuscript to improve the text flow. For better continuity, we integrated the MedMNIST experiments into the introductory paragraph of the results section (page 5, lines 125-), and moved the complementary experiments to the supplement:

"We evaluated our models in three benchmarks. The first, the 'in-domain benchmark', aimed to determine UMedPT's performance on tasks closely related to its pretraining database. The second, the 'out-of-domain benchmark', aimed to assess how well UMedPT adapted to new tasks outside its immediate training domain. The third, the MedMNIST benchmark [16], was used to evaluate the proposed multi-task training strategy on a separate training database and, independently, to test UMedPT."

The UMedPT ablation studies have also grown with the new label type ablation study. For this reason, we have provided better integration by moving the detailed description to the supplement and just embedded a short description and the key results into the main text on page 5, lines 141-152:

"For our in-domain and out-of-domain clinical benchmarks, we conducted ablation studies for UMedPT to investigate the effects of the variable input image size of UMedPT compared to the fixed input image size of 224×224 with UMedPT-fixed, and whether to include trainable parameters in the LayerNormalizations within its architecture with UMedPT-affine, which are detailed in Extended Data Section S1. We found that a variable input size was beneficial for the performance of UMedPT, while UMedPT-affine had a minor impact on the results. In addition, we compared the performance of UMedPT with a variant that was trained only with the classification tasks UMedPT-clf, as described in Extended Data Section S2. This showed a great benefit of including segmentation and object detection tasks, especially for other similar tasks."

As discussed below in this letter, we applied UMedPT to MedMNIST to investigate the model's generalizability. The obtained results have been integrated into Table 1 (cropped in this letter).

Table 1 Amount of data required by UMedPT to match state-of-the-art performance on classification tasks from different imaging domains. Datasets marked with an asterisk (*) were compared across different test splits.

Task	Reference results	UMedPT	
		Frozen	Fine-Tuning
MedMNIST mean AUC	See Ext. Data Figure 3 [16]	100%	10%
MedMNIST mean ACC	See Ext. Data Figure 3 [16]	-	10%

The outcome of the comparison with external reference results is presented in page 9, lines 246-248:

"[...] UMedPT surpassed the external reference results in [...] and the average AUC in the MedMNIST database [16]."

While we have moved the results of the architectural comparison, the training schemes, and from the catastrophic forgetting investigation to the supplement, we have chosen to keep the MedMNIST 3D contextual preservation experiment in the main body of the manuscript and added context to keep the integration seamless (page 10, lines 266-292):

"2D multi-task learning outperforms single-task 3D CNNs

Independently of our training database, we evaluated the training strategy on MedMNIST. MedMNIST contains a variety of standardized and downscaled biomedical image datasets, including both 2D and 3D images. To assess the impact of transforming inherently 3D medical image data into 2D slices for pretraining, we evaluated single-task learning with 2D data, single-task learning with original 3D data [15] and multi-task learning with 2D data. For our multi-task and single-task trainings, we converted the 3D data into 2D slices by slicing through the last dimension and applied a multiple-instance learning classification task that was based on a weighted averaging operation across slices. When only the 3D tasks were considered, the average accuracies were:

- $83.22 \pm 1.61\%$ for single-task learning pretrained using Imagenet;
- 83.76% for the single-task MedMNIST 3D CNN [15];
- $86.46 \pm 1.13\%$ for multi-task learning.

The results showed that single-task 3D CNNs performed better than single-task 2D CNNs pretrained using Imagenet. However, multi-task 2D networks outperformed the single-task 3D CNNs. We present the details in Extended Data Table 5.

In this context, we also investigated the performance difference between CNN and Transformer architectures for our multi-task learning strategy, as presented in Extended Data Section S3 and aspects of the training algorithm in Extended Data Section S4. We found that the Swin Transformer architecture has a minimal positive impact, as shown in Extended Data Figure 4a. Regarding training schemes, we found that without gradient accumulation, both convergence and performance were worse with our training strategy, as shown in Extended Data Figure 4b. "

As the sections regarding architecture and training schemes were moved to the supplement, we improved their coherent organization around specific queries:

"S3 Comparing convolutional networks and transformer

For quantifying the effect of the encoder's architecture, we used the MedMNIST database including two-dimensional and three-dimensional classification tasks. We chose the ResNet-50 as convolutional neural network (CNN) and the tiny variant of the Swin Transformer because they are similar in size. The Swin Transformer has 27,582,570 trainable parameters compared to 23,508,032 for the ResNet-50 CNN.

The comparison of the Swin Transformer and ResNet-50 CNN architectures showed a minimal impact on model performance for the MedMNIST database. The Swin Transformer achieved an average test accuracy of $86.76 \pm 0.79\%$ over 5 repetitions, while the ResNet-50 CNN achieved an accuracy of $86.34 \pm 1.01\%$. In addition, a discrepancy in training convergence rates was observed between the two architectures, as shown in Extended Data Figure 4a.

S4 Investigating training schemes

We performed an analysis of training schemes on the MedMNIST database, including the two- and three-dimensional tasks. All datasets in MedMNIST have a fixed number of cases. This distinction enabled us to conduct ablation studies comparing infinite task sampling with balanced sampling based on dataset size. Besides this, we used the same training schedule and hyperparameters as in the main study, and accumulated the gradients of as many steps as there were tasks. In addition, for comparison with traditional training schemes, we used the same setting without gradient

accumulation and also with the SGD optimizer instead of Adam. The exploration of training schemes showed that balanced (by dataset size) and cyclic sampling (as in UMedPT) exhibited similar behaviour in terms of convergence. However, balanced sampling occasionally showed reduced stability; it yielded a standard deviation of $1.81 \pm 1.79\%$ in validation accuracy over the previous ten epochs, across five different experiments. In comparison, cyclic sampling showed a more stable training process, achieving a comparatively lower standard deviation of $1.17 \pm 1.09\%$. When gradient accumulation was excluded, the resulting performance deteriorated, accompanied by longer convergence times. These results are shown in Extended Data Figure 4b."

Reviewer: 2. Evaluation Strategy Regarding the Validation Set: The authors state, "We allocated a fixed computational budget for all evaluations and used the last model state". This approach, however, implicitly treats the test set as a validation set. Such a methodology could inadvertently lead to an overestimation of the model's performance in a machine learning context. I suggest that the authors re-evaluate their strategy to mitigate potential biases.

Reply: We appreciate the concern about the potential overestimation of our model's performance. In response, we would like to clarify that no test data were used for validation in our study. Consequently, the risk associated with our method, whenever we have not used a validation set, is an underestimation of model performance rather than an overestimation.

However, it is still of great importance to examine the best way to apply pretrained foundational models to tiny and large target datasets. For this reason, we appreciate the feedback and conducted the experiment suggested by reviewer 3, and integrated it with the previous experiments regarding this topic on page 50, lines 1327-1351:

"Inverse relationship between performance and dataset size

Our evaluation within the clinical benchmark revealed an unexpected trend in some datasets: increasing the dataset size for fine-tuning sometimes led to a decrease in model performance.

To investigate the potential influence of catastrophic forgetting [37] or overfitting during fine-tuning, we first evaluated this phenomenon using four MedMNIST tasks that had shown improved performance with multi-task learning compared to single-task learning. We first measured the test accuracy of these tasks after multi-task learning, followed by further individualised training with the full dataset of each task, and assessed the test accuracy again. The results varied between datasets, suggesting that whether datasets are affected by forgetting the well generalizing state from multi-task learning is inconsistent and may be task-dependent:

- SynapseMNIST3D: $83.81 \pm 0.31\% \rightarrow 82.90 \pm 0.66\%$ (decrease)

- VesselMNIST3D: $93.66 \pm 0.31\% \rightarrow 93.77 \pm 0.87\%$ (no decrease)
- BreastMNIST: $86.92 \pm 1.04\% \rightarrow 85.90 \pm 0.91\%$ (decrease)
- PneumoniaMNIST: $91.54 \pm 0.48\% \rightarrow 91.70 \pm 0.49\%$ (no decrease)

For our in-domain and out-of-domain target tasks, we always used 100 epochs. Consequently, larger datasets used more optimization steps and could overfit more easily. We investigated by keeping large validation sets (30% of the full training data) in one in-domain and one out-of-domain task where the phenomenon occurred and performed model selection using the validation set. Extended Data Figure 5 shows that for one task the model selection with the validation set was better, for the other task it was worse.”

Extended Data Fig. 5 Model selection with and without validation sets. For the target tasks in our clinical benchmark, we did not use validation sets to really only use the given percentage of training data (UMedPT). This could lead to overfitting on the training data, which is usually solved by using a validation set, as done with UMedPT-Val. We investigated this using a representative out-of-domain data set, Tuber-CXR (a), and an in-domain target task, Pneumo-CXR (b).

“We further investigated whether this could be due to catastrophic forgetting of the well generalizing pretrained features or overfitting to the training data and found that the phenomenon is dataset specific, as detailed in Extended Data Section S5.”

Additionally, when applying UMedPT to MedMNIST we followed the standard protocol that was set out by the authors of the MedMNIST database. As opposed to the original experiments with UMedPT, these new results included the use of a validation set. The results of the model selection with validation set, together with the selection of the final model, are presented in the new Extended Data Figure 3 of our paper. This figure shows that the model selected by the top F1 score on the validation set performs slightly better than the final model, but at the cost of acquiring additional validation data:

Extended Data Fig. 3 UMedPT’s application to MedMNIST. First, UMedPT was applied to MedMNIST [15] with the shared encoder frozen and a randomly initialized linear head (linear probing) and evaluated on the test set using area under the curve (AUC, left) and accuracy (ACC, right). The whole model was then fine-tuned independently for each task. Blue and green lines represent the test performances when the model was selected using the validation set provided by the authors. Red and orange lines represent the test performance when the last model state was selected (validation data not used). Horizontal lines represent the theoretically best performance when the best reference method is selected for each task and metric independently (red) or when the best method is selected for all tasks (grey). We evaluated UMedPT with 1%, 10% and 100% of the training data. Details are given in Extended Data Table 6.

From these four findings, we see that there is no clear winning training strategy across all target tasks. The need for further in-depth evaluation is now motivated in the discussion section on page 12, lines 356-372:

“In some cases we observed that the performance of UMedPT decreased as the size of the training dataset increased. We investigated both, catastrophic forgetting [35] of the well generalizing features learned during pretraining and overfitting to the training set due to using all data for training instead of a validation set for model selection in Extended Data Section S6. The inconsistency of the results raises questions about the best practices for using foundational models in tasks with varying data sizes and varying degrees of similarity to the pretraining database. There were tasks that performed best with model selection using a

validation set, and tasks that performed best with all the data used for training. Similarly, some tasks performed best with the frozen training setting, and others with fine-tuning of all pretrained parameters of UMedPT. Recently, more sophisticated fine-tuning strategies have been proposed for foundational models in natural language processing, such as BitFit [36], where only the bias-terms are fine-tuned, or LoRA [37], where a small number of additional parameters are fine-tuned. A training configuration targeted specifically to foundational vision models could combine the strengths of the different training configurations.”

Reviewer: 3. CNN Backbone Results on UMedPT: The addition of results using a CNN backbone is noted; however, these were only conducted on MedMNIST and not on UMedPT. Given that recent works in medical imaging have found transformer-based methods not to outperform CNNs, it would be beneficial to include CNN backbone results on UMedPT as well.

Reply: We did not include CNN backbone results on UMedPT because we already included CNN backbone results with MedMNIST and the inclusion for UMedPT would be costly and not provide substantial new insights due to the following reasons:

- In the context of follow-up work on a specialized pretraining for histology, we used the presented training method with full-scale data within a modern CNN known as "ConvNeXt". Preliminary results show a negligible difference between the two architectures, with a small preference for the Swin Transformer. This will be reported as part of the confidentially attached follow-up paper.
- Furthermore, in the revised version of the current manuscript, we included the MedMNIST database for pretraining and evaluation separately from UMedPT. These MedMNIST pretrainings, which are already included in this study, used a traditional CNN. The performance of the two architectures was similar, although the Swinformer backbone had a slight advantage. This finding is consistent with the results of our follow-up work on specialized pretraining for histology and recent literature.

In summary, our study and preliminary results from the follow-up work show a negligible difference between the results of the two architectures. We have elaborated on this point in the discussion (p. 13, lines 399-407):

“In addition to the ability to handle arbitrary image sizes, for the UMedPT encoder we needed a general base architecture capable of generating multi-scale feature maps, a feature found in both convolutional neural networks (CNNs) and swin transformers [37]. Our experiments with MedMNIST showed a minimal difference between CNN and Swin Transformer, slightly in favour of the latter. This suggests that the proposed pretraining strategy can be implemented with both convolutional and transformer-based encoders, with literature showing that CNNs can also work well with large datasets of full-size images [38].”

And we improved the integration of the CNN backbone results on MedMNIST within the methods section on page 17, lines 524-529:

“UMedPT’s decoders are compatible with any encoder that can generate multi-scale feature maps [...] We also investigated the compatibility of the CNNs with the proposed multitask training loop and included an additional comparison in Extended Data Section S4.”

Reviewer: 4. Generalization Experiments on MedMNIST: Concerning the added generalization experiments on MedMNIST, an important setting seems to be missing. The analysis only distinguishes

between MTL and ST differences without examining the impact of pre-trained weights from UMedPT. Assessing whether or not UMedPT's pre-trained weights are loaded is crucial for a thorough analysis.

Reply: We agree that this important setting was missing and applied UMedPT pretrained weights to the MedMNIST database.

We added a description of this setting to the Methods section "Comparison of benchmark results" (page 26, lines 855-859):

"We compared our results with the best previously reported study results for the target tasks and the mean performance for the MedMNIST database [15]. From the MedMNIST database, we only considered tasks that were available in the largest spatial size (224 × 224) and were not part of the UMedPT pretraining or clinical benchmark."

Corresponding results are also reported in Table 1 (cropped):

Table 1 Amount of data required by UMedPT to match state-of-the-art performance on classification tasks from different imaging domains. Datasets marked with an asterisk (*) were compared across different test splits.

Task	Reference results	UMedPT	
		Frozen	Fine-Tuning
MedMNIST mean AUC	See Ext. Data Figure 3 [16]	100%	10%
MedMNIST mean ACC	See Ext. Data Figure 3 [16]	-	10%

The results are presented in Extended Data Figure 2:

Extended Data Fig. 2 UMedPT's application to MedMNIST. First, UMedPT was applied to MedMNIST [15] with the shared encoder frozen and a randomly initialized linear head (linear probing) and evaluated on the test set using area under the curve (AUC, left) and accuracy (ACC, right). The whole model was then fine-tuned independently for each task. Blue and green lines represent the test performances when the model was selected using the validation set provided by the authors. Red and orange lines represent the test performance when the last model state was selected (validation data not used). Horizontal lines represent the theoretically best performance when the best reference method is selected for each task and metric independently (red) or when the best method is selected for all tasks (grey). We evaluated UMedPT with 1%, 10% and 100% of the training data. Details are given in Extended Data Table 6.

Further results are reported in Extended Data Table 6 (rotated for readability in the paper):

Extended Data Table 6 Detailed Performance of UMedPT on MedMNIST. Comparison of accuracy (ACC) and area under the curve (AUC) at different stages of training UMedPT on the MedMNIST database: using the frozen encoder and fine-tuning the whole model. Performance metrics are provided for UMedPT with 1%, 10% and 100% of the training data. Reference results are included from ResNet-50 (Ref. CNN) or the theoretical best results obtained by selecting the method with the strongest test performance for each dataset and metric independently (Ref. Cherrypick).

dataset	stage metric fraction	UMedPT - Frozen		UMedPT - Finetune		Ref. Cherrypick		Ref. CNN	
		ACC	AUC	ACC	AUC	ACC	AUC	ACC	AUC
bloodmnist	1%	82.81±1.10%	98.88±0.12%	96.85±0.48%	99.88±0.01%	-	-	-	-
	10%	96.11±0.16%	99.82±0.00%	98.02±0.08%	99.95±0.00%	-	-	-	-
	100%	98.20±0.03%	99.91±0.00%	99.14±0.08%	99.99±0.00%	96.60%	90.80%	95.00%	99.70%
breastmnist	1%	60.08±14.84%	63.07±3.45%	59.62±15.97%	64.89±2.53%	-	-	-	-
	10%	73.72±1.38%	76.95±2.15%	83.76±1.60%	88.28±1.70%	-	-	-	-
	100%	82.91±0.30%	90.99±0.31%	94.02±0.60%	94.41±0.33%	86.30%	91.90%	84.20%	86.60%
chestmnist	1%	94.77±0.03%	72.55±0.60%	94.76±0.06%	74.86±0.88%	-	-	-	-
	10%	94.82±0.01%	78.37±0.15%	94.81±0.01%	80.53±0.09%	-	-	-	-
	100%	94.84±0.00%	80.24±0.01%	94.90±0.01%	84.11±0.04%	94.80%	77.80%	94.80%	77.30%
dermamnist	1%	66.92±0.06%	71.68±2.38%	68.96±0.55%	79.58±2.18%	-	-	-	-
	10%	71.49±0.53%	85.01±1.63%	78.42±0.27%	92.20±0.17%	-	-	-	-
	100%	78.22±0.08%	93.30±0.01%	91.21±0.23%	98.64±0.09%	76.80%	92.00%	73.10%	91.20%
octmnist	1%	69.70±3.31%	97.15±0.08%	84.97±2.19%	98.85±0.14%	-	-	-	-
	10%	76.33±0.61%	97.78±0.09%	88.37±0.12%	98.72±0.24%	-	-	-	-
	100%	77.53±0.39%	97.84±0.02%	92.00±0.83%	99.50±0.16%	77.60%	96.30%	77.60%	95.80%
organamnist	1%	70.95±1.53%	96.52±0.19%	86.69±1.21%	98.93±0.07%	-	-	-	-
	10%	83.28±0.21%	98.49±0.01%	95.16±0.18%	99.80±0.02%	-	-	-	-
	100%	87.01±0.10%	99.04±0.01%	96.86±0.18%	99.89±0.01%	95.10%	99.80%	94.70%	99.80%
organmnist	1%	38.53±1.46%	88.87±0.73%	70.16±2.19%	95.72±0.45%	-	-	-	-
	10%	72.25±0.27%	96.58±0.04%	89.82±0.36%	99.36±0.05%	-	-	-	-
	100%	81.67±0.12%	98.31±0.00%	95.24±0.13%	99.86±0.01%	92.80%	99.40%	91.10%	99.30%
organismnist	1%	47.20±3.01%	88.40±0.89%	64.33±0.83%	92.99±0.32%	-	-	-	-
	10%	70.32±0.16%	95.59±0.16%	80.83±1.27%	97.88±0.13%	-	-	-	-
	100%	77.08±0.13%	97.16±0.02%	84.89±0.08%	98.54±0.06%	81.30%	97.50%	78.50%	97.50%
retinamnist	1%	42.50±0.35%	58.07±5.78%	42.42±0.06%	60.64±4.02%	-	-	-	-
	10%	47.08±1.85%	71.89±4.02%	57.75±3.89%	81.63±2.89%	-	-	-	-
	100%	57.00±0.41%	83.96±0.08%	65.83±0.92%	88.57±0.24%	53.10%	75.00%	51.10%	71.60%
tissuemnist	1%	54.86±0.22%	86.12±0.10%	59.52±0.30%	88.80±0.04%	-	-	-	-
	10%	59.32±0.11%	88.74±0.02%	68.06±0.16%	92.78±0.02%	-	-	-	-
	100%	60.21±0.09%	89.28±0.00%	76.63±0.14%	96.06±0.01%	70.30%	94.10%	68.00%	93.20%

Reviewer: (Minor Points) a. UMedPT-affine: The explanation regarding UMedPT-affine, which mentions "UMedPT used layernorms without parameters," is still somewhat confusing to me. Layer normalization typically includes parameters by default, so I am curious about the rationale behind this specific analysis by the authors.

Reply: Several networks in this study, or parts of them, use Batchnorm by default. Batchnorm with trainable parameters did not work as found in preliminary experiments (results not shown). We have clarified the rationale to test both the inclusion (UMedPT-affine) and the exclusion of learnable parameters in its normalization layers (UMedPT) on page 17, lines 535-556:

"However, in our experiments, batch normalization led to poor performance (results not shown). One assumption when using batch normalization is that all input batches follow the same distribution. When combining different tasks and datasets, this assumption no longer holds.

While we observed that normalization layers improved training speed, we believed that they would underperform similarly in layer normalization due to the ineffectiveness of trainable parameters in batch normalization. [...] As a result, the default configuration of UMedPT excludes trainable parameters in its normalization layers. First, previous studies have shown that trainable parameters such as bias and gain within layer normalization layers increase the risk of overfitting and generally do not contribute to improved performance [45]. Second, given the ineffectiveness of trainable parameters in batch normalization, we hypothesized that they might similarly underperform with layer normalization."

Reviewer: (Minor Points) b. Formatting of "revised_manuscript.pdf": Many tables appear to be incompletely formatted.

Reply: Thank you for the feedback, we fixed the table formatting.

Reviewer #2

Reviewer: The authors have adequately addressed my queries.

Reply: Thank you for your positive feedback. We appreciate the time you took to provide a valuable review of our work, which significantly helped us to improve our manuscript.

Reviewer #3

Reviewer: Summary:

The authors propose a supervised pretraining strategy that leverages a multitude of medical datasets and tasks to reach ImageNet scale medical supervised pretraining. After they pretrain their model using their dataset, they evaluate their model on 2 in domain tasks and 5 out of domain tasks. They show that their method significantly outperforms an ImageNet pretrained model on the in domain tasks, and also outperforms the ImageNet baseline on the out of distribution tasks. Furthermore, the authors compare their model performance to external baselines, demonstrating significantly improved data efficiency.

Strengths:

- The authors aggregate a large number of tasks (15 datasets/17 tasks)
- The authors demonstrate how these heterogeneous tasks can be leveraged to enhance downstream classification performance
- The authors show how their method significantly outperforms ImageNet pretraining, as well as external baselines
- This paper is a nice demonstration that supervised pretraining is beneficial for medical imaging. As more medical imaging datasets come online with associated labels, this method could continue to improve.

Reply: Thank you for your summary and thoughtful comments on our manuscript. We appreciate the time, effort, and expertise you have invested in reviewing our work.

Reviewer: Overall, the major weaknesses are that the authors may overstate some methodological contributions and use language that risks leading the reader to think that they generated new contributions/insights, which were in fact developed/observed previously. The authors can successfully address these by modifying how they describe their contributions. Below are several examples:

- Page 14, line 435: The authors claim "To address this challenge, we developed a novel training strategy for UMedPT that mostly decouples the number of training tasks from the memory requirements." The authors further state "Our strategy achieved this by establishing an independent architecture or 'computational graph' for each task. The graph is dynamically constructed and stored only during the active computation stage of each task. To combine the individual graphs, we implement gradient accumulation before the optimization step." Could the authors clarify what they mean by "Our strategy achieved this by establishing an independent architecture or 'computational graph' for each task"? It appears that the authors are using PyTorch, which would generate a single computational graph for the full model, including multiple task-specific heads. Furthermore, the language should not confuse the reader into thinking that implementing the computational graphs is a part of the author's contribution, when this is how Pytorch operates under the hood.

Reply: Thank you for pointing out this possible point of confusion. We clarified that we used PyTorch and implementing computational graphs is not part of our contribution (page 15, lines 450-451):

"We used PyTorch [39] to create an independent architecture, or 'computational graph', for each task. [...]"

In this way we clarified that by "an independent architecture or 'computational graph' for each task" we meant the PyTorch computational graph, and made the implication for our training strategy clear (page 15, lines 457-461):

"GA allowed [...] a single update step could consist of heterogeneous tasks in any order. This allowed the training strategy to use an adaptive architecture, where each type of label can be solved by a specialized combination of model components, such as a UNet for segmentation labels [45]."

Reviewer: The authors further state, "We ensure that the model's weights and gradients are stored only once, rather than duplicating them for each task. Additionally, only the activations for one task are kept in memory at a time." The language may be a bit strong and overclaim contributions here. Gradient accumulation as implemented in the code below is used routinely, with all handling of the computational graph by PyTorch. GA implementation requires simply not calling `loss.backward()` at every step in PyTorch. The contribution here is sampling all tasks within a global GA step. I would recommend that the authors soften their language in this section, and make more clear what they contributed versus previous methods implemented by others that they are explaining for the education of the reader.

[Code]

Reply: Regarding our application of gradient accumulation, in our view, the simplicity of this part of the multi-task training strategy is an advantage. We softened the language, clarified that we did not propose gradient accumulation, and made it clearer that, unlike previous methods, we have implemented a way to appropriately apply gradient accumulation to multitask learning. (page 15, lines 462-465):

"GA is a common method for incorporating more data into a single optimization step. In the case of our multi-task learning strategy, unlike traditional deep multi-task learning, GA allowed the weights and gradients of the shared part of the model to be stored only once, rather than duplicated for each task. [...]"

Reviewer: Layer norm is typically used in vision transformers, including in the SwinTransformer architecture. Therefore, the authors should make sure that their language does not risk confusing the author into thinking that this is a new finding/contribution. As the default SwinTransformer uses layernorm, the following language should be softened - "To address this problem, we recursively replaced the original normalization layers in all shared blocks with layer normalization, which by design do not require inter-task computation".

The authors state - "We empirically analysed the effect of using layer norms with affine parameters on our approach using an adaptation of UMedPT(UMedPT-affine). "..."UMedPT-affine added trainable parameters including a bias and a scaling factor γ in the form $y = \gamma x - \mu + \beta$ for each channel." The default implementation of layernorm in PyTorch and in the original paper includes learnable bias and scaling factors. Therefore, I would update the language used to ensure that readers don't confuse this as a new contribution.

Reply: Besides the encoder, we added decoders to the shared part of the architecture of UMedPT which by default come with Batchnorm. Batchnorm was found to be incompatible with our training strategy and thus we made sure that users of the training framework do not have to manually adapt their architectures to make them compatible.

"Batch normalization [48] is a widely used normalization technique and is the default normalization layer within the ResNet convolutional neural network that we used for the

MedMNIST benchmark [16], UMedPT's segmentation decoder [44] and UMedPT's objectdetection decoder [47]"

The respective part of the Methods section now ensures that readers do not consider the use of Layernorm within swin transformers as a new contribution (page 17, lines 548-549):

"Notably, the Swin Transformer encoder used in UMedPT already used layer normalization, which comes with trainable parameters by default."

And later on page 18, lines 557-560:

"To empirically assess the impact of excluding such trainable parameters in UMedPT, we compared it to a variant of our model UMedPT-affine that included trainable bias and scaling layer normalization parameters, which is the default for the Swin Transformer, the UMedPT encoder. [...]"

Reviewer: (minor weakness) Page 9, line 226 "A comparison for the OrganSeg-MRI task could not be performed, because no results specific to the MRI-only subtask of the challenge were reported". I would request that the authors train a baseline nnUNet or other state of the art baseline for comparison. Otherwise, it is difficult to understand the segmentation performance on this dataset.

Reply: We appreciate the suggestion and agree that nnU-Net provides a strong baseline to understand the performance of our method in 3D segmentation. For this analysis, we replaced OrganSeg-MRI with a lung nodule segmentation task with documented state-of-the-art performance. We describe the application of the proposed multi-task training method to this task on page 50, lines 1352-1380:

"Investigating the Applicability to 3D Segmentation Tasks

To evaluate the application of a stacked 2D segmentation approach to 3D images, we examined a lung nodule segmentation task from the medical segmentation decathlon [28]. For compatibility with the benchmark's results, we retrained UMedPT with the nine remaining tasks from the decathlon's dataset in addition to UMedPT's training database. We refer to this version of the model as UMedPT-large.

The pretraining methodology for UMedPT-large was the same as for UMedPT. To adapt to the target task, we then trained a linear task-specific head on the output of the frozen UMedPT-large. The model was trained using full slices. For inference, we used 2D inference on full slices and stacked the results to create a 3D prediction.

We compared with nnU-Net [53], as a baseline for medical 3D segmentation. While we used whole slices (512×512) for training, nnU-Net used a patch size of $128 \times 128 \times 128$. Our evaluation strategy followed the baseline's approach of 5-fold cross validation. For evaluation, we adopted the 3D Dice from [52], reporting only the foreground class. In terms of results, UMedPT-large achieved a Dice score of 71.96%, while for non-pretrained nnU-Net 52.68% and 66.85% are reported for 2D and 3D, respectively [53]. However, at the time of writing, the online leaderboard of the Medical Segmentation Decathlon reports higher metrics (using different test data).

For future work, we suggest following the workflow that was successful with the external evaluation of a colorectal cancer classifier in gigapixel image classification. In this process, we first used UMedPT-large to extract features, followed by the application of a smaller specialized CNN to the whole gigapixel image at once. For 3D segmentation, this specialized

network could be a 3D CNN. Alternatively, the pretraining segmentation task could be extended to incorporate 3D spatial context as we did for 3D classification with MedMNIST.”

We integrated the key findings into the results section for 3D classification on page 11, lines 293-297:

“In addition to 3D classification, we also investigated the direct applicability of our training method to 3D lung nodule segmentation in CT scans [28]. The experiments showed that a large 2D pretraining and the larger 2D spatial context it enables can compensate for the loss of 3D context, as detailed in the extended data section S6.”

Reviewer: (minor weakness) If I am not mistaken, all downstream tasks are classification tasks, except for one segmentation task where the baseline is not too strong. Therefore, something that would really strengthen this work, perhaps as future work, is an ablation study that assesses how much the segmentation and object detection pretraining tasks actually benefit downstream performance on various task types. The latents in the encoder decoder architecture trained for segmentation or object detection need to retain geometrical information but may not require pixel intensity information. On the other hand, performing well on downstream classification tasks does not require encoding precise geometrical information. It could in fact be the case that these are at odds and requiring capacity for precise geometric information reduces embedding quality for classification tasks. Investigating which types of tasks should actually be included during pretraining would be a nice contribution that would put this paper into better context.

Reply: We appreciate the feedback and have replaced the segmentation task where the baseline is not too strong with a more appropriate one with coloscopy data, consisting of data that did not occur within the UMedPT’s pretraining database and is more closely related to natural images on which the ImageNet baseline was trained with.

We added the out-of-domain task Polyp-RGB, which was used to train a model for polyp segmentation in coloscopy data. It was included in the methods section (page. 26, lines 845-853):

“Polyp segmentation in coloscopy (PolypSeg-RGB):

The PolypSeg-RGB task [25] focused on segmenting polyps from the background in coloscopy images. Since polyps can be precursors to colorectal cancer, coloscopy is an important diagnostic tool. Early detection and removal of polyps is essential to prevent the development of colorectal cancer. However, the effectiveness of coloscopy is often hampered by high miss rates; studies have found that polyp miss rates during coloscopy can range from 14 to 30%, depending on the type of polyp [54]. We randomly divided the dataset into 700 training images and 300 test images.”

Consistent with the other results, we find further evidence that pretraining with UMedPT does not impair learning, even in cases where UMedPT was not pretrained for: (page 9, lines 233-243):

“Polyp segmentation in coloscopy (PolypSeg-RGB):

The PolypSeg-RGB target task focused on the segmentation of polyps in coloscopy images. When using the entire dataset for fine-tuning, ImageNet achieved its best average result, demonstrating a mean Intersection over Union (mIoU) of 0.905. Here, UMedPT achieved an mIoU of 0.911. The ImageNet pretrained model showed better results when the encoder was frozen, as presented in Extended Data Figure 1c. The best performance across all fractions was achieved by UMedPT with fine-tuning. In addition, while UMedPT with fine-tuning outperformed ImageNet for all fractions, the biggest difference occurred with 1% of the data (0.797 ± 0.09 compared to 0.683 ± 0.144 of ImageNet).”

The details are presented within the out-of-distribution plots in the supplement (cropped here):

Extended Data Fig. 1 Results of remaining out-of-distribution tasks. a BC-BHis-MIC b CNS-MRI. c PolypSeg-RGB.

The strong results of both ImageNet and UMedPT are further evidenced by the fact that they outperform i) the original baseline provided by the dataset authors, and ii) in the case of UMedPT, even recent approaches developed specifically for polyp segmentation. This is now shown in our comparison with external reference results (shown here cropped):

Table 1 Amount of data required by UMedPT to match state-of-the-art performance on classification tasks from different imaging domains. Datasets marked with an asterisk (*) were compared across different test splits.

Task	Reference results	UMedPT	
		Frozen	Fine-Tuning
PolypSeg-RGB	0.778 mIoU [24]	50%	1%
PolypSeg-RGB	0.9051 mIoU [25]	-	100%

Additionally, we added a detection task for nuclei counting. While it used tissue types that were not part of the pretraining database, we still classified it as in-domain due to its similarity to one of the histological pretraining detection tasks.

In the methods section (page 25, lines 780-788):

“Detection of nuclei in whole slide images (NucleiDet-WSI):

In oncology, the distribution and appearance of nuclei are important for the diagnosis and study of cancer. To assess the ability of UMedPT to detect these nuclei, the NucleiDet WSI dataset [52] was used. This dataset consists of whole slide images (WSI) and covers ten cancer types. In the pretraining database, only prostate and colon cancer were included. We randomly divided the dataset into 950 images for training and 406 images for testing. The authors of the dataset created the annotations with the help of two pathologists and three graduate students, using an AI tool.”

The results show that only 50% of the training data is required to outperform ImageNet (page 6, lines 187-196):

“Detection of nuclei in whole slide images (NucleiDet-WSI):

We used the NucleiDet-WSI dataset [18] to detect nuclei in 10 different cancer types from whole slide images (WSI). The best ImageNet performance was achieved using 100% of the data together with fine-tuning, resulting in a mean average precision (mAP) of 0.71. UMedPT was able to replicate this performance with 50% of the training data and no fine-tuning. However, fine-tuning tended to improve the results for both models. Interestingly, compared to ImageNet, UMedPT showed superior performance across all data fractions with both fine-tuning and a frozen pre-trained model. This resulted in a maximum performance of 0.792 mAP when using the full training data set and fine-tuning.”

We present the results as part of Figure 2:

Fig. 2 Results for in-domain tasks. **a** In diagnosing pneumonia (Pneumo-CXR), UMedPT matched the full fine-tuned performance of ImageNet, even with a frozen encoder and a reduced dataset size (1%). **b** For CRC-WSI, the only target task for which its training dataset was also part of the pretraining, performance was stable with a frozen encoder. When the encoder was fine-tuned, performance decreased to the result obtained with ImageNet pretraining. **c** For NucleiDet-WSI, an object detection task for counting nuclei in whole slide images, UMedPT outperformed ImageNet across all training settings. Best performance was achieved with 100% of the training data and fine-tuning.

Now that we have added a segmentation task where the baseline is strong and another detection task, we can evaluate a new ablation that excluded segmentation and detection tasks from its pretraining “UMedPT-clf”. We introduced this in the Methods section, integrated into the section about the pretraining tasks (page 21, lines 622-627):

“To further understand the importance of pretraining diversity, we conducted an ablation study focusing only on classification tasks. We trained an ablation UMedPT-clf using only the classification pretraining tasks. We evaluated UMedPT-clf on one representative task from classification (Pneumo-CXR), segmentation (PolypSeg-RGB) and object detection (NucleiDet-WSI) and compared it to the full model UMedPT.”

We integrated the key finding alongside the results of the other pretraining ablation studies (page 5, lines 148-152):

"In addition, we compared the performance of UMedPT with a variant that was trained only with the classification tasks UMedPT-clf, as described in Extended Data Section S2. This showed a great benefit of including segmentation and object detection tasks, especially for other similar tasks."

The detailed results of the label type ablation were added to the supplement:

"S2 Benefit of segmentation and object detection in pretraining

To quantify the effect of including multiple label types in the pretraining, we compared UMedPT with a model trained on our classification pretraining tasks only, which we call UMedPT-clf. The results are shown in Extended Data Figure 2. There is a large average difference and consistently better performance of UMedPT for tasks requiring high spatial resolution features. For the object detection task NucleiDet-WSI, UMedPT achieved a 0.282 higher mean Average Precision (mAP), and for the segmentation task Coloscopy-RGB, it outperformed UMedPT-clf by 0.057 mIoU. Interestingly, although the difference was smaller for Pneumo-CXR (classification), a clear positive knowledge transfer between the label types was found, with an advantage of 2.42% F1-score in favour of UMedPT."

Extended Data Fig. 2 Results of label type ablation study with UMedPT-clf. UMedPT-clf was trained with the same classification tasks as UMedPT, but excluded segmentation and object detection tasks. **a** Pneumo-CXR (classification). **b** NucleiDet-WSI (object detection). **c** PolypSeg-RGB (segmentation).

Reviewer: (minor weakness) I understand that this study has been for the most part limited to supervised pretraining. However, this method will need to compete with self-supervised pretraining which could potentially scale more easily. A comparison to such methods, like MAE and DINO (i.e. RAD-DINO), would be a nice to have.

Reply: This is an interesting point, especially as our method also enables including both supervised and self-supervised tasks into the same pretraining. However, for a meaningful evaluation multiple methods should be included as part of a broader follow-up project. We included the suggestion into the discussion (page 12, lines 322-331):

“[...] Alternatively, self-supervised pretraining can be used to improve the data-efficiency for target tasks, as demonstrated with RAD-DINO [30]. However, recent literature suggests that label-supervised pretraining for imaging typically outperforms self-supervised pretraining empirically [10, 31] and theoretically [32]. Nonetheless, it offers great value in regularizing models and might help in further reducing the required volume of labelled data. Our approach can be extended to include an arbitrary number of self-supervised tasks into the pretraining, which may further enhance the generalizability of UMedPT, especially in domains where abundant data are available but labelling is difficult or costly.”

Reviewer: (minor weakness) Page 14, line 439: Can the authors clarify what this means: “This graph is dynamically constructed and stored only during the active computation stage of each task”?

Reply: We improved the structure of the paragraph and improved the text on page 15, lines 451-:

“This graph was dynamically constructed so that each label type could be solved by a different architecture, but still shares almost all model parameters. For example, in the case of UMedPT, a UNet [44] for segmentation labels was assembled by combining the shared Swin Transformer encoder with the shared pixel-dense decoder and a small task-specific part. To combine the individual graphs, we used gradient accumulation (GA) [...]. It also made the tasks independent, so that the dynamic sub-graph of a task only needed to be loaded into memory during the active computation stage of that task.”

Reviewer: (minor weakness) Page 17, line 527 - What does “recursively replaced” mean here? The norm layers should not be nested so what does recursing mean here?

Reply: The proposed training strategy can be used with a wide range of architectures. We used the PyTorch’s capability to enumerate children of a module for replacing the original normalization with normalization layers that are compatible with the gradient-accumulation-based training strategy (see `mmm.torch_ext.replace_childen_recursive`). We have clarified this on page 17, lines 542-544:

“Consequently, we took advantage of the tree-like property of PyTorch neural networks and recursively replaced the original normalization layers in all shared blocks with layer normalization [...]”

Reviewer: (minor weakness) Page 18, line 570: could the authors make it more clear what “the need for pre-extraction of images” means?

Reply: For faster convergence, batches should contain diverse image instances (e.g., patches etc.) from multiple images. Loading entire images for the extraction of image instances is time- and memory-intensive. In this context, “pre-extraction” means the storage of extracted image instances to disk before training. Here, we propose a strategy to enable training with a large number of medical images where each image potentially consists of many image instances. We have explained this on page 19, lines 596-599:

"[...] to load whole slide images and 3D volumes. It is common practice to pre-extract image instances to disk or memory to minimize loading times, but this requires a lot of memory and loses the ability of perform augmentation on the original data. The proposed caching component [...]"

Reviewer: (minor weakness) Page 21, line 622 - if "flips and mirroring" are applied as augmentations, the network could lose the ability to differentiate left sided vs right sided diseases, which is an area of study for medical foundation models. Can the authors justify the inclusion of these augmentations?

Reply: We apologize for the incomplete explanation in the text. Flips and mirroring were used for histological images. We clarified this on page 21, lines 636-640:

"For X-ray images, we added image inversion with a probability of 30%. For histological images, flipping and mirroring were used to improve orientation invariance, and channel shuffling was used to improve the model's robustness to stain and color variations."

Reviewer: (minor weakness) Page 21, line 614 - what are the "standard 3D augmentations"? Can the authors include those in the paper?

Reply: We included those on page 21, lines 629-630:

"For 3D tomographic images, we applied standard 3D augmentations using the MONAI library [50] (3D rotations, scale and crop), followed by slicing [...]"

Reviewer: (minor weakness) Page 23, line 712 - The authors should make it clearer that the 1%, 5%, 10%, 25%, 50%, and 100% corresponds to the downstream datasets, not the pretraining dataset, if that is in fact the case.

Reply: Thanks, we clarified this on page 24, lines 740-743:

"To simulate data-scarcity and evaluate sample efficiency, we took multiple samples from the original training set of target tasks at sizes of 1%, 5%, 10%, 25%, 50%, and 100%. For pretraining, we always used the full pretraining data sets."

Reviewer: (minor weakness) The authors should clarify whether there is any overlap in the downstream datasets and the pretraining datasets. It seems that CRC-WSI may be present in pretraining (Extended Table 4) and was also used for downstream validation in the comparison with ImageNet. If this is the case, it may not be fair to claim that finetuning with 1% of downstream data compared to ImageNet pretraining, as the training dataset was seen during pretraining. Could the authors clarify whether this is a typo in Extended Data Table 4?

Reply: WSI-CRC was included as both a pretraining and downstream data set to test for the recoverability of pretraining knowledge, while making sure that the test images are strictly kept out of all trainings. However, the task-specific pretrained part was discarded and had to be relearned from a new random initialization during the downstream training. We clarified that in the caption of Figure 2 (results):

"CRC-WSI was the only target task where the training dataset was also part of the pretraining. Here, performance was stable across dataset fractions with a frozen encoder. When the encoder was fine-tuned, [...]"

And in the methods on page 24, lines 754-756:

“The training images of one of the target tasks, CRC-WSI, were included in both pretraining and benchmarking.”

Reviewer: (minor weakness) Page 23, line 705 - The authors state “In the frozen scenario, we directly extracted image representations from the shared blocks, thereby showing the usefulness of the learned representations. Both frozen and fine-tuning were trained for epochs each.” Does training in the frozen case mean training a linear probe for classification tasks? If so, I would use this common terminology.

Reply: Training in the frozen case means training a linear probe for all tasks, not just classification. We have clarified this on page 23, lines 730-733:

“In the frozen scenario, [...] we used a single linear layer for all target tasks (including segmentation and object detection), also known as linear probing. Subsequently [...]”

Reviewer: (minor weakness) Page 23, line 723 - What does “re-discovery” and “re-identification” mean here? Would like to clarify that this does not mean that downstream datasets were used during pretraining. If a downstream dataset was used during pretraining, it does not seem fair to claim that fine tuning on 1% of downstream data yielded similar performance to ImageNet baseline.

Reply: We used the training images from CRC-WSI in both pretraining and downstream, and refer to the answer two points above.

Reviewer: (minor weakness) Extended Data Table 3: Which version of the model are you using for your results in the main paper? Can you add this to the caption?

Reply: When not stated otherwise, we obtained our results in the main paper with UMedPT. We added this to the caption of Table 1 and Extended Data Table 1:

“Unless otherwise stated, all results in the main paper were obtained with UMedPT.”

Reviewer: (minor weakness) The authors should add more details about how inference is done with 3D data. Are predictions averaged across slices? Is the following method applied to the origin UMedPT model as well? “For the 3D tasks, we used a simple strategy based on a learned weighted average across slices with the classification task described in section.”

Reply: For 3D classification the neural representations were averaged (not the predictions). Subsequently, linear probing was applied as in the 2D classification.

We clarified this procedure on page 11, lines 274-276 (results) and on page 28, lines 926-931 (methods):

“We then applied a multiple-instance learning classification task that was based on a weighted averaging operation over the neural representations of the slices.”

“For the 3D tasks, we used a simple strategy based on a learned weighted average over the neural representations of the slices. This results in a single feature vector per 3D case, allowing the use of the same linear classification head as in 2D, as described in section 5. Intuitively, this allows the network to learn focusing on the most relevant slices of a 3D case before a prediction.”

Reviewer: (minor weakness) Could the authors add a bit more explanation for their choices in normalization factors in equations (1) and (2)?

Reply: The normalization factors were determined based on random weights and random inputs such that the normalization factors

- prevent one task from dominating the others during training.
- allow meaningful model selection based on average loss (not all tasks can use the same metric for meaningful averaging).

We have extended and moved the explanation into the respective section (page 21, lines 642-947):

“We have included classification, segmentation, and detection tasks. These have different loss functions with different magnitudes. We normalized the respective loss functions for each task type so that the observed value for random inputs for reinitialized weights was close to one. This strategy prevented the loss of one task from dominating the combined loss. In addition, this allows model selection based on the average loss.”

Reviewer: Point of Discussion:

Without validation sets, it may be difficult to understand whether catastrophic forgetting is the main culprit for decreasing performance with increasing dataset size or if the authors are overfitting to the downstream task, with the number of optimization steps increasing along with the fraction of the downstream dataset used for fine-tuning. Inevitably, some forgetting is happening when adapting to a specific downstream dataset. However, a somewhat related issue could be overfitting with 100 epochs of fine-tuning. I understand the challenge here where the authors want to be able to make the claim that truly 1% of the dataset was used for training, vs a larger fraction of training dataset size + validation dataset size. It may be necessary to have a larger validation set to get a clear signal about model performance. I commend the authors for truly using 1% of data for training, versus using a small training dataset but then a much larger validation dataset. A nice to have ablation to include in this work or future work would be investigating performance if you use full validation sets and modulate only the training dataset size. If using a validation set actually causes model performance to increase with training dataset size, then you can be confident that the performance decrease is only coming from suboptimal checkpoint selection. This would add additional support for the efficacy of the method.

Reply: We appreciate the idea and investigated if the decreasing performance happens due to checkpoint selection as suggested. We sampled a validation set from the training data and used the same test set, making the large training splits smaller than before. We selected one in-domain and one out-of-domain target task where we observed the decrease.

We integrated the experiment with the previous experiments regarding this topic on page 50, lines 1327-1351:

“Inverse relationship between performance and dataset size

Our evaluation within the clinical benchmark revealed an unexpected trend in some datasets: increasing the dataset size for fine-tuning sometimes led to a decrease in model performance.

To investigate the potential influence of catastrophic forgetting [37] or overfitting during fine-tuning, we first evaluated this phenomenon using four MedMNIST tasks that had shown improved performance with multi-task learning compared to single-task learning. We first measured the test accuracy of these tasks after multi-task learning, followed by further individualised training with the full dataset of each task, and assessed the test accuracy again. The results varied between datasets, suggesting that whether datasets are affected by forgetting the well generalizing state from multi-task learning is inconsistent and may be task-dependent:

- SynapseMNIST3D: $83.81 \pm 0.31\% \rightarrow 82.90 \pm 0.66\%$ (decrease)
- VesselMNIST3D: $93.66 \pm 0.31\% \rightarrow 93.77 \pm 0.87\%$ (no decrease)
- BreastMNIST: $86.92 \pm 1.04\% \rightarrow 85.90 \pm 0.91\%$ (decrease)
- PneumoniaMNIST: $91.54 \pm 0.48\% \rightarrow 91.70 \pm 0.49\%$ (no decrease)

For our in-domain and out-of-domain target tasks, we always used 100 epochs. Consequently, larger datasets used more optimization steps and could overfit more easily. We investigated by keeping large validation sets (30% of the full training data) in one in-domain and one out-of-domain task where the phenomenon occurred and performed model selection using the validation set. Extended Data Figure 5 shows that for one task the model selection with the validation set was better, for the other task it was worse.”

a

b

Extended Data Fig. 5 Model selection with and without validation sets. For the target tasks in our clinical benchmark, we did not use validation sets to really only use the given percentage of training data (UMedPT). This could lead to overfitting on the training data, which is usually solved by using a validation set, as done with UMedPT-Val. We investigated this using a representative out-of-domain data set, Tuber-CXR (a), and an in-domain target task, Pneumo-CXR (b).

We reference the key findings in the main text on page 6, lines 175-179:

“We further investigated whether this could be due to catastrophic forgetting of the well generalizing pretrained features or overfitting to the training data and found that the phenomenon is dataset specific, as detailed in Extended Data Section S5.”

Additionally, when applying UMedPT to MedMNIST we followed the standard protocol that was set out by the authors of the MedMNIST database. As opposed to the original experiments with UMedPT, these new results included the use of a validation set. The results of the model selection with validation set, together with the selection of the final model, are presented in the new Extended Data Figure 3 of our paper. This figure shows that the model selected by the top F1 score on the validation set performs slightly better than the final model, but at the cost of acquiring additional validation data:

Extended Data Fig. 3 UMedPT’s application to MedMNIST. First, UMedPT was applied to MedMNIST [15] with the shared encoder frozen and a randomly initialized linear head (linear probing) and evaluated on the test set using area under the curve (AUC, left) and accuracy (ACC, right). The whole model was then fine-tuned independently for each task. Blue and green lines represent the test performances when the model was selected using the validation set provided by the authors. Red and orange lines represent the test performance when the last model state was selected (validation data not used). Horizontal lines represent the theoretically best performance when the best reference method is selected for each task and metric independently (red) or when the best method is selected for all tasks (grey). We evaluated UMedPT with 1%, 10% and 100% of the training data. Details are given in Extended Data Table 6.

From these four findings, we see that there is no clear winning training strategy across all target tasks. The need for further in-depth evaluation is now motivated in the discussion section on page 12, lines 356-372:

“In some cases we observed that the performance of UMedPT decreased as the size of the training dataset increased. We investigated both, catastrophic forgetting [35] of the well generalizing features learned during pretraining and overfitting to the training set due to using all data for training instead of a validation set for model selection in Extended Data Section S6. The inconsistency of the results raises questions about the best practices for

using foundational models in tasks with varying data sizes and varying degrees of similarity to the pretraining database. There were tasks that performed best with model selection using a validation set, and tasks that performed best with all the data used for training. Similarly, some tasks performed best with the frozen training setting, and others with fine-tuning of all pretrained parameters of UMedPT. Recently, more sophisticated fine-tuning strategies have been proposed for foundational models in natural language processing, such as BitFit [36], where only the bias-terms are fine-tuned, or LoRA [37], where a small number of additional parameters are fine-tuned. A training configuration targeted specifically to foundational vision models could combine the strengths of the different training configurations."

Reviewer: Extended Data Table 4: the authors should specify in the caption what "/" means. In, for example, the third column of the CRC-WSI row, where 100,000/7,000 is written. Is this train/test data?

Reply: We clarified this in the caption of Extended Data Table 4:

"For the dataset splits provided by the respective publishers, we marked them as train/test and included only the training set in the pretraining."

Reviewer: Fig. 2 - In the caption: is Tuber-CXR the same dataset as Pneumo-CXR in the plot titles? BC-Bach-WSI referenced in caption, as opposed to CRC-WSI.

Reply: Thank you for pointing this out. The caption now describes the results of the in-domain datasets instead of describing the wrong plot:

"Fig. 2 Results for in-domain tasks. a In the diagnosis of pneumonia (Pneumo-CXR), UMedPT matched the full fine-tuned performance of ImageNet, even with a frozen encoder and a reduced dataset size [...]"

Reviewer: Page 1, line 29 - maybe consider updating "required not more than 50%" to "required only 50% of the original training data".

Reply: We appreciate the suggestion and implemented it in the abstract.

Reviewer: Page 1, line 19 - I'm not sure that I would consider medical dataset to be more heterogenous than natural domain datasets that can comprise any scene/object. Medical images generally look similar globally with differences coming from finer grain features.

Would the authors mind justifying this description or revising it?

Reply: We agree and revised it in the abstract:

"However, training these models typically requires large, comprehensive datasets, which contrasts with the smaller and more specialized datasets common in biomedical imaging."

Reviewer: Page 2, line 48 - for clarity, I would consider updating "increasingly large pretrainings" to "increasingly large pretraining datasets"

Reply: We agree and changes it in the Introduction section.

Reviewer: Page 2, line 53 - May want to add LAION, in addition to ImageNet.

Reply: We appreciate the suggestion and added LAION, in addition to ImageNet, to the introduction in line 38: "such as ImageNet-1K [1] or LAION [2]", and in line 53: "in the biomedical domain, there is no single pretraining dataset comparable to ImageNet or LAION."

Reviewer: Page 9, Table 1. Should there be a citation for CNS-MRI?

Reply: Thank you for pointing out that issue. We added the citation.

Reviewer: Page 23, line 696 - what does "synthetic dataset" mean here? Usually it refers to data which isn't real, but generated.

Reply: Yes, we used generated data to avoid over-fitting the hyperparameters to any of the real data sets. We briefly clarified that on page 23, lines 718-721:

"We developed the downstream training schedule and tuned the hyperparameters using a simple synthetic dataset based on simple, two-dimensional, geometric shapes for all label types. We then performed [...]"

Reviewer: Page 23, line 708 - What does subsequently refer to here? "After frozen and fine-tuning for 100 epochs, subsequently fine-tuning stage enabled training of shared blocks". Is the order implied by the word subsequently significant here?

Reply: The fine-tuning stage was initialized by the task head from the frozen stage. This ensured that all evaluations were completed in time for a fair comparison. Furthermore, in the fine-tuning phase, this may have helped to ensure that the pre-trained features are not destroyed from either UMedPT or ImageNet if the first steps generate large gradients due to a randomly initialized head. However, we did not quantify the benefits of this evaluation scheme and ran it only once. We added a clarification on page 24, lines 733-736:

"Subsequently, the fine-tuning stage enabled the training of the shared blocks such that the parameters learned during the frozen training setting were used to initialize the task-specific head."

Reviewer: Extended Data Table 5 extends beyond the page width

Reply: We appreciate the comment and decreased the font size of the table.

Reviewer: Would encourage the authors to remove commented code from their codebase and also add comments within most functions/classes that describe their purpose, along with descriptions of arguments, their types, and any outputs.

- Would encourage the authors to remove commented code from their codebase.
- Also within these readmes, I would use code blocks to demonstrate to the user how to run the code, as opposed to "use universal_pretraining.py".
- Would encourage the authors to add comments within most functions/classes that describe their purpose, along with descriptions of arguments, their types, and any outputs.
- In the readmes within each subdirectory, I would include a description of the organization of the code. What are each of the "neural", "optimization", "trainer", "logging", "interactive", and "data_loading" folders for?
- The specific wheels in the requirements.txt file are not supported by certain systems. Instead of including the wheel links, I would instead specify versions. I had to remove these to install the code.

Reply: We appreciate your feedback on the code. The submission only included preliminary code that will be uploaded to Zenodo for the purpose of reproducing the study. There, we have removed the commented code and improved the associated readme files as suggested.

In preparation for the open source release associated with this paper, we have added comments to most functions/classes describing their purpose, described the structure of the repo in the readme and improved the cleanliness. The dependency management has been reworked to use a pyproject.toml file and optional dependency groups to simplify the installation process. We also improved the code for training the latest version of UMedPT. Future incremental improvements will be made to the open-source repositories. We have added the current progress as a third directory in the code folder.

Reviewer: Overall the code seems modular and it seems as though lots of effort was put into making the code robust.

Reply: Thank you. We appreciate the positive feedback on the code.

Reviewer: There are currently two subfolders in the top level directory and no readme. I would put a readme in the top level directory so that users know what the two subfolders are for. This readme should describe the code within each subfolder on a high level. It should tell the user why code is split into two subfolders and what is different about each code base.

- I would change the name of the "code" subdirectory to be more descriptive.

Reply: We added a readme to the top-level directory explaining to reviewers that these are two versions of the same code base, one used for UMedPT training and one used for the MedMNIST benchmark. We have added the current state of the code intended for publication as a third folder and named each folder after its purpose.

Reviewer: If possible, I would add links to all datasets that the user needs to download to the readme. This would make it significantly easier for others to collect the datasets and reproduce the results in this paper.

Reply: Thank you for the valuable suggestion, we duplicated and extended the list from the respective paper section into the Readme of the UMedPT code repository.

Decision Letter, second revision:

Date: 6th May 24 13:13:49
Last Sent: 6th May 24 13:13:49
Triggered By: Ananya Rastogi
From: ananya.rastogi@nature.com
To: fkiessling@ukaachen.de
CC: computacionalscience@nature.com
BCC: ananya.rastogi@nature.com
Subject: AIP Decision on Manuscript NATCOMPUTSCI-23-1256B
Message: Our ref: NATCOMPUTSCI-23-1256B

6th May 2024

Dear Dr. Kiessling,

Thank you for submitting your revised manuscript "Overcoming Data Scarcity in Biomedical Imaging with a Foundational Multi-Task Model" (NATCOMPUTSCI-23-1256B). It has now been seen by the original referees and their comments are below. The reviewers find that the paper has improved in revision, and therefore we'll be happy in principle to publish it in Nature Computational Science, pending minor revisions to satisfy the referees' final requests and to comply with our editorial and formatting guidelines.

TRANSPARENT PEER REVIEW

Nature Computational Science offers a transparent peer review option for original research manuscripts. We encourage increased transparency in peer review by publishing the reviewer comments, author rebuttal letters and editorial decision letters if the authors agree. Such peer review material is made available as a supplementary peer review file. **Please remember to choose, using the manuscript system, whether or not you want to participate in transparent peer review.**

Please note: we allow redactions to authors' rebuttal and reviewer comments in the interest of confidentiality. If you are concerned about the release of confidential data, please let us know specifically what information you would like to have removed. Please note that we cannot incorporate redactions for any other reasons. Reviewer names will be published in the peer review files if the reviewer signed the comments to authors, or if reviewers explicitly agree to release their name. For more information, please refer to our FAQ page.

Thank you again for your interest in Nature Computational Science. Please do not hesitate to contact me if you have any questions.

Sincerely,

Ananya Rastogi, PhD
Senior Editor
Nature Computational Science

ORCID

Reviewer #1 (Remarks to the Author):

I have reviewed the revisions to the manuscript and am satisfied with the changes made. I recommend acceptance.

Reviewer #2 (Remarks to the Author):

I have no additional comments - the authors have done an excellent job in addressing the queries.

Reviewer #3 (Remarks to the Author):

I appreciate the authors' thorough responses to the my comments and the comments of the other reviewers and believe that the authors have addressed all of the points.

Final Decision Letter:

Date: 17th June 24 10:52:43

Last Sent: 17th June 24 10:52:43

Triggered By: Ananya Rastogi

From: ananya.rastogi@nature.com

To: fkiessling@ukaachen.de

BCC: ananya.rastogi@nature.com,fernando.chirigati@us.nature.com,computationalscience@nature.com,rjsproduction@springernature.com

Subject: Decision on Nature Computational Science manuscript NATCOMPUTSCI-23-1256C

Message Dear Professor Kiessling,

:

We are pleased to inform you that your Article "Overcoming Data Scarcity in Biomedical Imaging with a Foundational Multi-Task Model" has now been accepted for publication in *Nature Computational Science*.

Once your manuscript is typeset, you will receive an email with a link to choose the appropriate publishing options for your paper and our Author Services team will be in touch regarding any additional information that may be required.

Please note that *Nature Computational Science* is a Transformative Journal (TJ). Authors may publish their research with us through the traditional subscription access route or make their paper immediately open access through payment of an article-processing charge (APC). Authors will not be required to make a final decision about access to their article until it has been accepted. Find out more about Transformative Journals

Acceptance of your manuscript is conditional on all authors' agreement with our publication policies (see <https://www.nature.com/natcomputsci/for-authors>). In particular your manuscript must not be published elsewhere and there must be no announcement of the work to any media outlet until the publication date (the day on which it is uploaded onto our web site).

Before your manuscript is typeset, we will edit the text to ensure it is intelligible to our wide readership and conforms to house style. We look particularly carefully at the titles of all papers to ensure that they are relatively brief and understandable.

Once your manuscript is typeset, you will receive a link to your electronic proof via email with a request to make any corrections within 48 hours. If, when you receive your proof, you cannot meet this deadline, please inform us at rjsproduction@springernature.com immediately.

If you have queries at any point during the production process then please contact the production team at rjsproduction@springernature.com.

You may wish to make your media relations office aware of your accepted publication, in case they consider it appropriate to organize some internal or external publicity. Once your paper has been scheduled you will receive an email confirming the publication details. This is normally 3-4 working days in advance of publication. If you need additional notice of the date and time of publication, please let the production team know when you

receive the proof of your article to ensure there is sufficient time to coordinate. Further information on our embargo policies can be found here: <https://www.nature.com/authors/policies/embargo.html>

We welcome the submission of potential cover material (including a short caption of around 40 words) related to your manuscript; suggestions should be sent to Nature Computational Science as electronic files (the image should be 300 dpi at 210 x 297 mm in either TIFF or JPEG format). We also welcome suggestions for the Hero Image, which appears at the top of our home page; these should be 72 dpi at 1400 x 400 pixels in JPEG format. Please note that such pictures should be selected more for their aesthetic appeal than for their scientific content, and that colour images work better than black and white or grayscale images. Please do not try to design a cover with the Nature Computational Science logo etc., and please do not submit composites of images related to your work. I am sure you will understand that we cannot make any promise as to whether any of your suggestions might be selected for the cover of the journal.

Best regards,

Ananya Rastogi, PhD
Senior Editor
Nature Computational Science

P.S. Click on the following link if you would like to recommend Nature Computational Science to your librarian: <https://www.springernature.com/gp/librarians/recommend-to-your-library>

** Visit the Springer Nature Editorial and Publishing website at www.springernature.com/editorial-and-publishing-jobs for more information about our career opportunities. If you have any questions please click here.**